# VisScience: An Extensive Benchmark for Evaluating K12 Educational Multi-modal Scientific Reasoning

## Abstract

Multi-modal large language models (MLLMs) have shown promise in integrating textual and visual information to handle complex visual understanding tasks. However, most benchmarks evaluating MLLMs focus mainly on mathematics or general visual understanding, revealing a significant gap in assessing capabilities across other critical scientific disciplines like physics and chemistry. To bridge this gap, we meticulously construct a comprehensive benchmark, **VisScience**, to evaluate multi-modal scientific reasoning across mathematics, physics, and chemistry. This benchmark comprises 3,000 questions drawn from K12 education, from elementary to high school levels, evenly distributed with 1,000 questions per discipline. VisScience encompasses 21 distinct subjects, classified into five difficulty levels to cover a wide range of topics within each discipline. We utilize VisScience to conduct a detailed evaluation of 25 representative MLLMs in scientific reasoning. The experimental results show that closed-source MLLMs generally surpass open-source models, with standout performances including a 53.4% accuracy in mathematics by Claude3.5-Sonnet, 38.2% in physics by GPT-4o, and 47.0% in chemistry by Gemini-1.5-Pro. These results underscore the strengths and limitations of MLLMs, suggesting areas for future improvement and highlighting the importance of developing models that can effectively handle the diverse demands of multi-modal scientific reasoning.

## 1 Introduction

Recently, large language models (LLMs) OpenAI (2022); Achiam et al. (2023); GLM et al. (2024); Touvron et al. (2023a;b); Bai et al. (2023a); Brown et al. (2020); Chowdhery et al. (2023); Anil et al. (2023) have demonstrated remarkable capabilities across a wide range of tasks, including natural language understanding, text generation, and complex problem solving. The success of LLMs facilitates the development of multi-modal large language models (MLLMs) OpenAI (2023); Team et al. (2023); Anthropic (2024); Liu et al. (2024b;a); Ye et al. (2023; 2024), which extends these capabilities by integrating the ability to process and analyze both textual and visual information. Evaluation is a significant component in assessing the ability of these MLLMs across various tasks, which has attracted widespread attention and developed rapidly in recent years. For instance, several benchmark datasets are proposed to evaluate the ability of MLLMs in general visual understanding, including MME Fu et al. (2023), MMMU Yue et al. (2024), MMBench Liu et al. (2023), MMStar Chen et al. (2024a), and SEED-Bench Li et al. (2023a).

As a primary evaluation domain, mathematical reasoning presents specific challenges, requiring models to handle complex mathematical problems accompanied by visual information. Previous works Chen et al. (2021; 2022); Cao & Xiao (2022) focus on geometric problems, resulting in the emergence of various evaluation datasets such as GeoQA Chen et al. (2021), Geometry3K Lu et al. (2021), and UniGeo Chen et al. (2022). Subsequently, several benchmark datasets Lu et al. (2023); Zhang et al. (2024b); Wang et al. (2024) extend the scope of mathematical reasoning beyond geometry to encompass various branches such as arithmetic, algebraic, statistics, logic, and functions. Notably, MathVista also contains a portion of scientific datasets such as TQA Kembhavi et al. (2017), SciBench Wang et al. (2023b), and ScienceQA Lu et al. (2022). However, despite these advancements, there remains some issues:

- *Existing benchmarks often focus narrowly on specific mathematics, neglecting other crucial scientific disciplines like physics and chemistry.*

- *Existing benchmarks are often collected from limited sources, resulting in a lack of natural difficulty levels and leading to an incomplete evaluation of models' capabilities.*

- *Current benchmarks are predominantly available in a single language, limiting the evaluation of MLLMs' multilingual capabilities.*

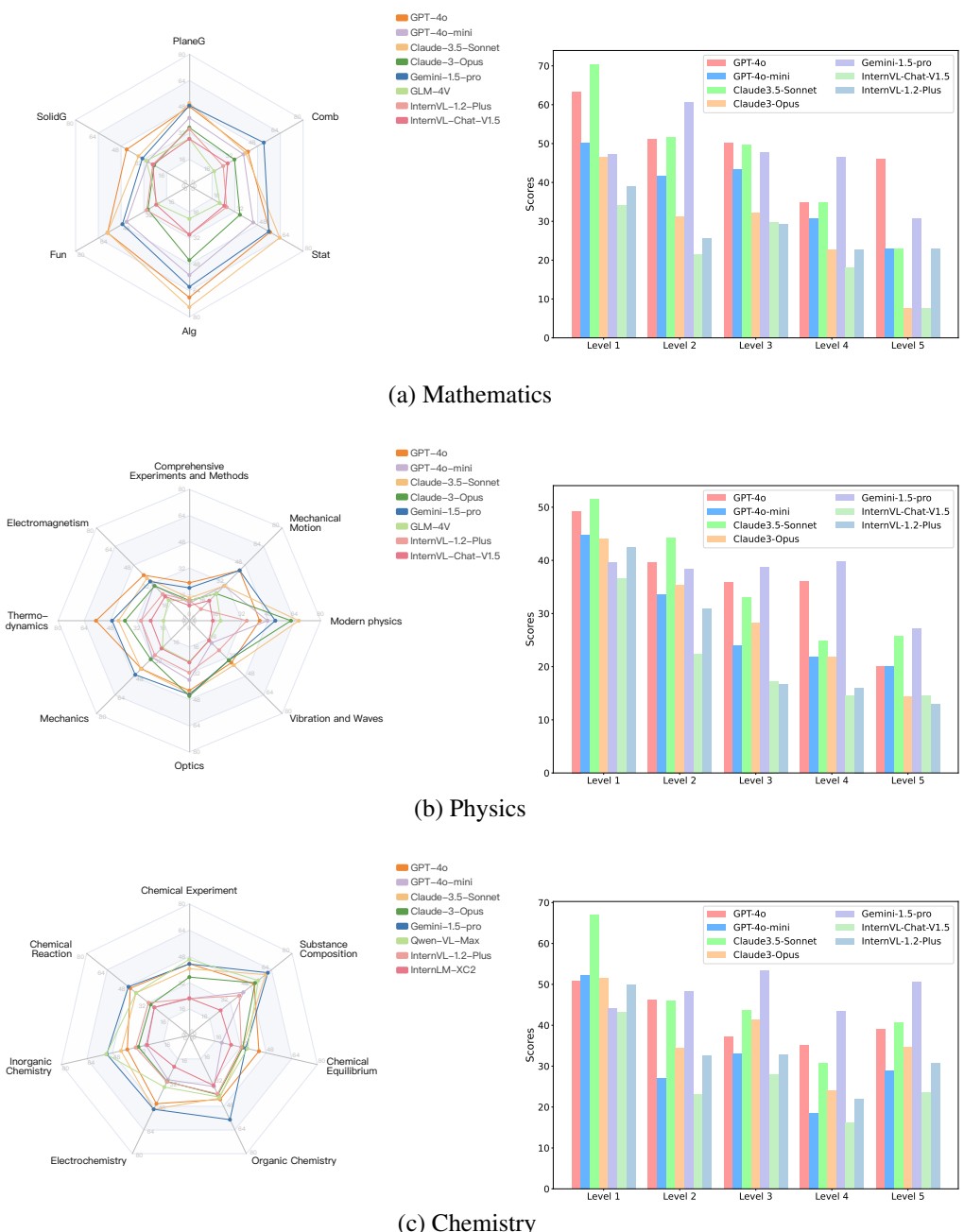

(a) Mathematics

(b) Physics

(c) Chemistry

Figure 1: The accuracies of representative MLLMs on VisScience across different subjects and difficulty levels. (Left) The accuracies on different subjects. (Right) The accuracies on various difficulty levels.

To address the limitations of existing benchmarks and provide a more comprehensive evaluation benchmark, we introduce a more expansive evaluation benchmark, named **VisScience**, integrating

both textual and visual information. This benchmark is designed to assess the performance of MLLMs in multi-modal scientific reasoning tasks across disciplines like physics and chemistry alongside mathematics. To construct this benchmark, we gather a total of 450,000 questions from K12 education and meticulously select 3,000 questions as the final dataset, with each discipline containing 1,000 questions. This benchmark spans a comprehensive range of knowledge points across different chapters, with difficulty levels ranging from 1 to 5, ensuring that models are assessed on both basic and challenging problems.

In order to better understand MLLMs' performance on more detailed subjects within three disciplines, we categorize VisScience into several subjects across each discipline. Specifically, we divide the mathematical part of VisScience into six subjects such as *plane geometry, solid geometry, functions and equations, algebraic operations, probability and statistics, and combinatorial mathematics*. For physics, the dataset is categorized as eight subjects, including *mechanics, thermodynamics, comprehensive experiments and methods, mechanical motion, vibration and waves, optics, electromagnetism, and modern physics*. The chemistry section includes seven topics such as *chemical experiments, organic chemistry, material composition, electrochemistry, chemical reactions, inorganic chemistry, and chemical equilibrium*. In summary, VisScience contains 21 subjects across the three disciplines of mathematics, physics, and chemistry.

We conduct extensive experiments on VisScience to evaluate the scientific reasoning abilities of 25 representative MLLMs. These models include close-source LLMs, close-source and open-source MLLMs, offering a comprehensive analysis of their performance across various disciplines. As illustrated in Figure 1, the best performance is observed in close-source MLLMs, with distinct models excelling in different disciplines. In specific, Claude3.5-Sonnet achieves an accuracy of 53.4% in mathematics, GPT-4o reaches a 38.2% accuracy in physics, and Gemini-1.5-Pro records an accuracy of 47.0% in chemistry. Among open-source models, InternVL-1.2-Plus performs best with accuracies of 30.1% in mathematics, 24.8% in physics, and 31.2% in chemistry. Lastly, we systematically analyze the errors made by advanced models like GPT-4o on VisScience, which providing valuable insights into the specific domains where these models excel and where they struggle.

## 2 VISSCIENCE BENCHMARK

In this section, we first illustrate the overview of our specially curated VisScience benchmark, designed to assess the capabilities of MLLMs in multi-modal scientific reasoning. Next, we introduce data generation process, which encompasses three core scientific disciplines: mathematics, physics, and chemistry. Lastly, we perform a comprehensive data analysis on the VisScience benchmark, including subject distributions and difficulty levels.

### 2.1 OVERVIEW

We introduce the VisScience benchmark, a meticulously curated collection aimed at evaluating the capabilities of multi-modal large language models (MLLMs) in multi-modal scientific reasoning, with a particular focus on bilingual tasks involving both English and Chinese. This dataset incorporates textual and visual contexts as inputs and spans three scientific disciplines, including mathematics, physics, and chemistry. Each discipline comprises 1,000 questions, meticulously gathered from different chapters to ensure comprehensive coverage of topics and concepts. The core statistics of the VisScience benchmark are presented in Table 1. The distributions of question length in VisScience are provided in Appendix A.1. A detailed introduction of each subjects within the three disciplines is available in Appendix A.2.

### 2.2 DATA GENERATION

The goal of the VisScience benchmark is to establish a comprehensive, bilingual (Chinese and English) benchmark for evaluating the capabilities of MLLMs in processing and understanding complex, scientifically-oriented tasks across various disciplines. In order to achieve this goal, we present a two-stage data generation pipeline to meticulously construct a benchmark dataset comprising 3,000 questions, evenly distributed with 1,000 questions each in the fields of mathematics, physics, and chemistry. More cases in VisScience are provided in Appendix B.

| Statistic | Number |
|---|---|
| Total questions | 3000 |
| - multiple-choice questions | 2,053 (68.4%) |
| - Free-form questions | 947 (31.6%) |
| Number of categories of math questions | 6 |
| Number of categories of physics questions | 8 |
| Number of categories of chemistry questions | 7 |
| Number of difficulty levels | 5 |
| Unique number of images | 3,000 |
| Unique number of questions | 3,000 |
| Unique number of answers | 1,427 |
| *Statistics with Chinese Language* | |
| Maximum question length | 1297 |
| Maximum answer length | 112 |
| Maximum choice number | 5 |
| Average question length | 162.85 |
| Average answer length | 20.93 |
| *Statistics with English Language* | |
| Maximum question length | 418 |
| Maximum answer length | 92 |
| Maximum choice number | 5 |
| Average question length | 80.93 |
| Average answer length | 12.30 |

Table 1: Key statistics of VISSCIENCE.

**Data Collection.**  We gather a total of 450,000 questions from the disciplines of mathematics, physics, and chemistry, each enriched with visual information sourced from K12 education. This collection spans a comprehensive range of knowledge points across different chapters, with the difficulty levels scaled based on education grade. Consequently, we cluster 150,000 questions per discipline and carefully select 1,000 representative questions. These questions exemplify a range of difficulty levels and a variety of subjects, guided by the following principles: *(1) Guaranteeing every knowledge point is included in VisScience benchmark. (2) Prioritizing the selection of questions from high-frequency knowledge points. (3) Ensuring a mixture of questions across various difficulty levels.*

In the end, the VisScience benchmark is constructed with 3,000 questions, with each of the three disciplines – mathematics, physics, and chemistry – contributing 1,000 questions. This approach ensures that the benchmark comprehensively covers a wide array of topics within each discipline, reflecting the breadth and depth required for a thorough assessment of MLLMs' capabilities.

**Data Annotation.**  To improve the quality of the VisScience benchmark, we conduct multiple checks using both manual reviews and LLM assessments to confirm the completeness of each question. For textual content, we check for accuracy, coherence and relevance, ensuring that each question aligns with the corresponding scientific discipline and is free of ambiguities. For associated visual content, we rigorously screen out images that are incorrect, unclear, or lacking in detail, retaining only those that are clear and richly informative. To maintain the volume of the VisScience benchmark, we compensate for questions removed due to incomplete information by selecting new questions on identical topics from the original dataset. This approach ensures that the overall number of questions and the breadth of content coverage are consistently maintained. This verification process guarantees that both the textual and visual components of the VisScience benchmark is a reliable and effective tool for evaluating the capabilities of MLLMs in scientific reasoning.

## 2.3 DATA ANALYSIS

We utilize statistical analysis to assess subject distributions and difficulty levels within the VisScience benchmark. Figure 2 presents a visual representation of the categorization of question within the

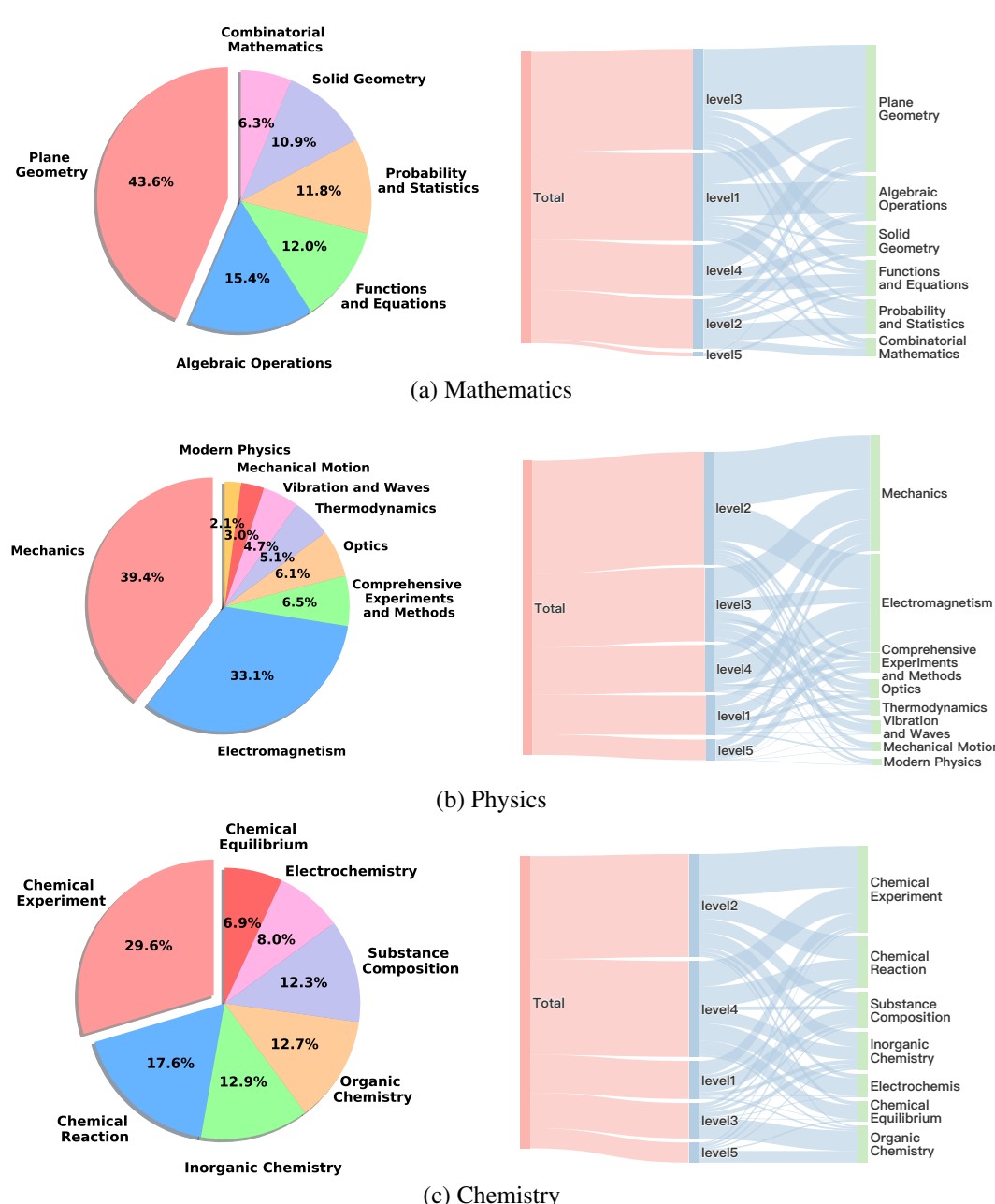

Figure 2: The distribution of detailed subjects and difficulty levels in the each discipline within the VisScience benchmark. (Left) The distributions of various subjects. (Right) The distributions of difficulty levels.

VisScience benchmark. This illustration shows the distribution of questions dedicated to each subject area – mathematics, physics, and chemistry – and details the distribution across various difficulty levels, ranging from 1 to 5. Besides, we discuss comparison among VisScience and other benchmarks in Appendix C.

**Subject Distributions.** To categorize each discipline into more detailed subjects, we first utilize LLM to segment the overall discipline into specific topics based on knowledge points and terminologies presented in the questions. Subsequently, we conduct a manual review of these categories to confirm its rationality and appropriateness, ensuring that each question is accurately categorized. As shown in Figure 2, the mathematical part of the VisScience benchmark is divided into six subjects, i.e., plane geometry (43.6%), algebraic operations (15.4%), functions and equations (12%), probability and

statistics (11.8%), solid geometry (10.9%), and combinatorial mathematics (6.3%). Furthermore, the distributions for physics and chemistry disciplines are presented in the figure, providing a comprehensive overview of the scope of the VisScience benchmark within these scientific fields.

**Difficulty Levels.** To classify the questions into distinct difficulty levels, we first utilize LLM for the initial sorting, and then conduct a manual verification. The questions within each discipline are stratified into five difficulty levels ranging from 1 to 5, defined as follows: Basic, Easy, Intermediate, Advanced, and Expert. Figure 2 shows the distribution of difficulty levels, providing a visual representation of the distribution of questions across different difficulty levels. Each discipline demonstrates a unique profile of topic distribution across the difficulty levels. For instance, in the field of mathematics, *plane geometry* is classified at the intermediate level, *algebraic operations* are positioned at the basic level, and *functions and equations* appears at the highest difficulty level, reflecting their various placement within educational curricula. In physics, *mechanics* dominates the introductory level, which belongs to a fundamental concept in physics education. *Electromagnet* is positioned at the highest difficulty level, demanding the application of various advanced knowledge points. In the discipline of chemistry, *organic chemistry* and *chemical equilibrium* represent the pinnacle of K12 chemical education, requiring deep conceptual understanding and the ability to apply knowledge to complex scenarios.

## 3 EXPERIMENTS

In this section, we conduct experiments to evaluate a variety of MLLMs using the VisScience benchmark. Besides, we provide a detailed error analysis of the advanced model GPT-4o.

### 3.1 EXPERIMENTAL SETUP

**Models.** We conduct our evaluation across a diverse array of models, including close-source text-only LLMs, close-source MLLMs, and open-source MLLMs. This comprehensive assessment covers 25 models, which are listed below. The sources of models is reported in Appendix D.1.

- Close-source text-only LLMs: ChatGPT OpenAI (2022), GPT-4 Achiam et al. (2023), Claude2 Anthropic (2023a).
- Close-source MLLMs: Gemini-1.0-Pro Team et al. (2023), Gemini-1.5-Pro Team et al. (2023), GPT-4o OpenAI (2024), Qwen-VL-Max Bai et al. (2023b), Qwen-VL-Plus Bai et al. (2023b), Claude3.5-Sonnet Anthropic (2023b), Claude3-Opus Anthropic (2024), GLM-4V AI (2023), and Step-1V StepFun (2024).
- Open-source MLLMs: mPLUG-Owl Ye et al. (2024), LLaMA-Adapter-V2 Gao et al. (2023), MiniCPM-Llama3-V2.5 Hu et al. (2024), LLaVA-1.5 Liu et al., DeepSeek-VL Lu et al. (2024), ShareGPT4V Chen et al. (2023a), SPHINX-Plus Gao et al. (2024), InternLM-XC2 Dong et al. (2024), InternVL-1.2-Plus Chen et al. (2023b), InternVL-Chat-V1.5 Chen et al. (2024c), CogVLM Wang et al. (2023a), CogVLM2 Wang et al. (2023a), and GLM-4V-9B GLM et al. (2024).

**Evaluation Details.** The evaluation process is conducted through two steps: generation and judgment. During the generation phase, the models are tasked with producing responses based on a set of questions. For zero-shot setting, we directly prompt the models with these questions without any examples. For 2-shot Chain-of-Thought (CoT) setting, we provide the models with two relevant examples before they are prompted with the questions. For MLLMs, we supply the models with the textual questions and the corresponding image to obtain their responses. During the judgment phase, we utilize GPT-4o to evaluate the models' responses by comparing them with the standard answers to assess consistency. This phase involves calculating the accuracy across different subjects and levels. The prompts used in two phases is defined in Appendix D.2.

### 3.2 EXPERIMENTAL RESULTS

**Overall Results.** Table 2 demonstrates the performance of several models on VisScience within the version of the Chinese language. Experimental results show that the close-source models achieves

best performance on VisScience. Specifically, Claude3.5-Sonnet achieves an accuracy of 53.4% in mathematics, GPT-4o attains a 38.2% accuracy in physics, and Gemini-1.5-Pro accomplishes an accuracy of 47.0% in chemistry. Among open-source models, InternVL-1.2-Plus stands out, demonstrating robust capabilities across various scientific disciplines with accuracies of 30.1% in mathematics, 24.8% in physics, and 31.2% in chemistry. Despite this, there is a notable disparity in performance between close-source and open-source models, with close-source models generally exhibiting superior performance. The performance of InternVL-1.2-Plus, although trailing behind the advanced close-source models such as GPT-4o, Claude3.5-Sonnet, and Gemini-1.5-Pro, showing significant potential for improvement. Notably, the performance in physics underscores unique challenges that necessitate targeted improvements in model training. This discipline often involves the interpretation of conceptual and numerical data, challenging the reasoning and computational abilities of MLLMs. As evidenced in Table 2, even advanced models like GPT-4o achieve relatively lower accuracies in physics compared to other disciplines. Results on VisScience with the version of the English language are provided in Appendix E.1.

**Results on Mathematics Across Different Subjects.** The mathematical part of VisScience encompasses a wide range of subjects, including plane geometry, solid geometry, functions and equations, algebraic operations, probability and statistics, and combinatorial mathematics. Table 3 reports the comprehensive results across different mathematical subjects. It is evident that models like Claude3.5-Sonnet and GPT-4o in close-source MLLMs excel across multiple subjects, particularly in *functions and equations*, *probability and statistics*, and *algebraic operations*. Conversely, open-source models show a more varied performance with notable strengths in certain areas but generally lower scores compared to close-source models. For instance, InternVL-1.2-Plus and InternVL-Chat-V1.5 perform relatively well in *plane geometry*, and *functions and equations*. These detailed performance on different subjects provide valuable insights into the specific strengths and weaknesses of various MLLMs. Additionally, results on physics and chemistry across different subjects are presented in Appendix E.2 and Appendix E.3, respectively. Case studies illustrating correct responses by MLLMs can be found in Appendix F.

## 3.3 ERROR ANALYSIS

To analyze the causes of errors in model responses, we meticulously review incorrect answers to identify common patterns. We specifically focus on GPT-4o to illustrate specific instances of errors and their distributions across the disciplines of mathematics, physics, and chemistry. Figure 3 demonstrates the distributions of these errors, categorizing them into several types such as reasoning error, knowledge error, calculation error, vision recognition error, and question misunderstood error. Notably, across all disciplines, reasoning errors are the most prevalent, indicating a challenge in model's ability to solve scientific problems that involve visual information. Specifically, reasoning errors account for 56.5% of the total errors in mathematics, 50.1% in physics, and 40.6% in chemistry, respectively. This is followed by knowledge error, which is particularly significant in chemistry, constituting 33.2% of the errors in that discipline. Similarly, knowledge error also represent the second most common error type in physics. However, knowledge error in mathematics is less prevalent, making up only 8.8% of the total errors. This indicates that while the model struggle with conceptual and fundamental principles in chemistry and physics, it demonstrate a better grasp of mathematical concepts. Vision recognition error is another significant type of error, accounting for 18.8% of the errors in mathematics, making it the second most prevalent error type in this discipline. This error category is also significant in physics and chemistry, where it constitutes 17.8% and 15.3% of the errors, respectively. This type of error highlights the challenges faced by the model in processing and understanding visual information. Furthermore, calculation error accounts for a small portion of the errors, especially in chemistry, indicating that the model excels particularly in handling numerical computations. More detailed examples of these errors can be found in Appendix G.

## 4 RELATED WORKS

**Multi-modal Reasoning Benchmarks.** Recently, the evaluation of multi-modal large language models (MLLMs) OpenAI (2023); Team et al. (2023); Anthropic (2024); Bai et al. (2023b); Wang et al. (2023a); Liu et al. (2024a;b) in various reasoning tasks has become increasingly crucial. So many benchmark datasets for these tasks span several categories are proposed like MME Fu et al.

| Model | LLM | Input | Mathematics | Physics | Chemistry |
|---|---|---|---|---|---|
| *Close Source Models* (APIs) | | | | | |
| *Text-only LLMs* | | | | | |
| Zero-shot ChatGPT | - | Q | 22.4 | 22.7 | 18.6 |
| Zero-shot GPT-4 | - | Q | 25.9 | 30.4 | 33.1 |
| Zero-shot Claude-2 | - | Q | 27.3 | 22.0 | 24.4 |
| Zero-shot Claude3-Opus | - | Q | 29.3 | 30.8 | 32.5 |
| Zero-shot Claude3.5-Sonnet | - | Q | 29.7 | 35.3 | 36.9 |
| Zero-shot GPT-4o | - | Q | 31.1 | 38.0 | 39.6 |
| 2-shot CoT Claude2 | - | Q | 27.8 | 21.7 | 23.9 |
| 2-shot CoT ChatGPT | - | Q | 20.2 | 18.6 | 21.3 |
| 2-shot CoT GPT-4 | - | Q | 32.1 | 31.5 | 32.4 |
| *Multi-modal LLMs* | | | | | |
| Gemini-1.0-Pro | - | Q, I | 26.6 | 23.70 | 27.8 |
| Gemini-1.5-Pro | - | Q, I | 49.4 | 38.1 | 47.0 |
| GPT-4o | - | Q, I | 51.7 | 38.2 | 41.6 |
| GPT-4o-mini | - | Q, I | 42.6 | 29.8 | 28.4 |
| Qwen-VL-Max | - | Q, I | 35.5 | 30.70 | 42.5 |
| Qwen-VL-Plus | - | Q, I | 27.6 | 26.5 | 37.7 |
| Claude3.5-Sonnet | - | Q, I | 53.4 | 38.0 | 43.1 |
| Claude-3 opus | - | Q, I | 34.4 | 31.1 | 34.1 |
| GLM-4V | - | Q, I | 24.2 | 19.2 | 25.0 |
| Step-1V | - | Q, I | 28.1 | 23.5 | 25.0 |
| *Open Source Models* | | | | | |
| *General Multi-modal LLMs* | | | | | |
| mPLUG-Owl | LLaMA-7B | Q, I | 7.6 | 8.3 | 9.5 |
| LLaMA-Adapter-V2 | LLaMA-7B | Q, I | 9.6 | 10.3 | 10.8 |
| MiniCPM-Llama3-V2.5 | LLaMA3-8B | Q, I | 15.4 | 17.9 | 19.5 |
| LLaVA-1.5 | Vicuna-13B | Q, I | 15.5 | 15.2 | 18.8 |
| LLaVA-1.5 | Vicuna-7B | Q, I | 13.0 | 13.5 | 16.0 |
| DeepSeek-VL | DeepSeek-LLM-7B | Q, I | 8.3 | 16.8 | 21.0 |
| ShareGPT4V | Vicuna-7B | Q, I | 15.7 | 14.0 | 19.0 |
| ShareGPT4V | Vicuna-13B | Q, I | 16.4 | 14.9 | 18.4 |
| SPHINX-Plus | LLaMA2-13B | Q, I | 17.0 | 15.3 | 20.4 |
| InternLM-XC2 | InternLM2-7B | Q, I | 24.9 | 18.3 | 25.6 |
| InternVL-1.2-Plus | Nous-Hermes-2-Yi-34B | Q, I | 30.1 | 24.8 | 31.2 |
| InternVL-Chat-V1.5 | Mixtral 8*7B | Q, I | 26.9 | 20.8 | 23.7 |
| CogVLM | Vicuna-7B | Q, I | 16.7 | 14.5 | 17.0 |
| CogVLM2 | LLaMA-3-8B | Q, I | 23.2 | 14.4 | 21.0 |
| GLM-4V-9B | GLM-4-9B | Q, I | 24.7 | 19.3 | 22.5 |

Table 2: **Results on VisScience within the version of the Chinese language across the disciplines of mathematics, physics, and chemistry.** For input, $Q$ represents for question, $I$ represents for image. The highest scores among close-source and open-source models are highlighted in red and blue, respectively.

(2023), MMMU Yue et al. (2024), MMBench Liu et al. (2023), MMStar Chen et al. (2024a), SEED-Bench Li et al. (2023a), and CMMMU Zhang et al. (2024a), which evaluate models' capabilities to apply logic and inference; mathematical reasoning; scientific reasoning, and agent-based reasoning. These benchmark datasets provide comprehensive measurements of MLLMs' capabilities in applying specialized knowledge and decision-making in simulated environments. For instance, MMMU covers university-level questions from six domains, which is utilized to assess MLLMs' advanced perception and reasoning abilities. CMMMU Zhang et al. (2024a) evaluates models' reasoning abilities across various disciplines through bilingual multi-modal questions in Chinese and English. Existing benchmark like ScienceQA Lu et al. (2022) is a specialized dataset designed to evaluate the capabilities of MLLMs, particularly in the domain of scientific reasoning. Furthermore, several

| Model | Mathematics | | | | | | |
| --- | --- | --- | --- | --- | --- | --- | --- |
| | ALL | PlaneG | SolidG | Fun | Alg | Stat | Comb |
| *Close Source Models* (APIs) | | | | | | | |
| *Text-only LLMs* | | | | | | | |
| Zero-shot ChatGPT | 22.40 | 20.18 | 11.93 | 18.33 | 13.63 | 15.25 | 26.98 |
| Zero-shot GPT-4 | 25.90 | 30.73 | 18.35 | 28.33 | 17.53 | 24.58 | 33.33 |
| Zero-shot Claude-2 | 27.30 | 27.06 | 25.69 | 25.83 | 31.17 | 31.36 | 25.40 |
| Zero-shot Claude3-Opus | 29.30 | 30.28 | 21.10 | 32.50 | 27.27 | 34.75 | 31.75 |
| Zero-shot Claude3.5-Sonnet | 29.70 | 33.94 | 15.60 | 33.33 | 27.27 | 27.12 | 34.92 |
| Zero-shot GPT-4o | 31.10 | 36.24 | 24.77 | 35.83 | 25.32 | 24.58 | 31.75 |
| 2-shot CoT Claude2 | 27.80 | 30.05 | 26.61 | 25.00 | 28.57 | 27.97 | 26.98 |
| 2-shot CoT ChatGPT | 20.20 | 23.17 | 20.18 | 19.17 | 17.53 | 22.88 | 14.29 |
| 2-shot CoT GPT-4 | 32.10 | 37.16 | 31.19 | 28.33 | 22.08 | 30.51 | 38.10 |
| *Multi-modal LLMs* | | | | | | | |
| Gemini-1.0-Pro | 26.60 | 24.08 | 22.02 | 23.73 | 35.71 | 29.66 | 34.92 |
| Gemini-1.5-Pro | 49.40 | 48.74 | 33.03 | 47.06 | 61.69 | 55.93 | 52.38 |
| GPT-4o | 51.70 | 48.17 | 44.04 | 57.50 | 68.18 | 56.78 | 41.27 |
| GPT-4o-mini | 42.60 | 41.28 | 29.36 | 44.17 | 54.55 | 44.92 | 38.10 |
| Qwen-VL-Max | 35.50 | 34.86 | 27.52 | 35.83 | 50.00 | 33.05 | 26.98 |
| Qwen-VL-Plus | 27.60 | 27.98 | 18.35 | 29.17 | 31.17 | 35.59 | 20.63 |
| Claude3.5-Sonnet | 53.4 | 50.23 | 35.78 | 57.50 | 74.03 | 63.56 | 39.68 |
| Claude3-Opus | 34.40 | 35.31 | 24.77 | 29.17 | 45.45 | 35.59 | 31.75 |
| GLM-4V | 24.20 | 28.57 | 30.28 | 22.50 | 20.26 | 21.37 | 17.46 |
| Step-1V | 28.10 | 31.68 | 24.71 | 23.15 | 48.85 | 40.57 | 22.64 |
| *Open Source Models* | | | | | | | |
| *General Multi-modal LLMs* | | | | | | | |
| mPLUG-Owl | 7.60 | 6.19 | 10.09 | 5.00 | 12.34 | 7.63 | 7.94 |
| LLaMA-Adapter-V2 | 9.60 | 10.78 | 10.09 | 7.50 | 9.09 | 13.56 | 4.76 |
| MiniCPM-Llama3-V2.5 | 15.40 | 23.62 | 19.27 | 15.83 | 26.62 | 26.27 | 15.87 |
| LLaVA-1.5-13B | 15.50 | 15.83 | 15.60 | 12.50 | 18.83 | 14.41 | 14.29 |
| LLaVA-1.5-7B | 13.00 | 12.84 | 12.84 | 15.83 | 14.29 | 11.86 | 11.11 |
| DeepSeek-VL | 8.30 | 13.99 | 8.26 | 10.00 | 11.04 | 10.17 | 7.94 |
| ShareGPT4V-7B | 15.70 | 16.06 | 16.51 | 13.33 | 14.29 | 17.80 | 17.46 |
| ShareGPT4V-13B | 16.40 | 15.60 | 11.93 | 19.17 | 17.53 | 22.03 | 14.29 |
| SPHINX-Plus | 17.00 | 21.79 | 19.27 | 15.83 | 20.13 | 22.88 | 7.94 |
| InternLM-XC2 | 24.90 | 25.92 | 22.02 | 22.50 | 27.92 | 27.97 | 20.63 |
| InternVL-1.2-Plus | 30.10 | 34.40 | 25.69 | 30.00 | 29.87 | 26.27 | 23.81 |
| InternVL-Chat-V1.5 | 26.90 | 28.44 | 25.69 | 23.33 | 29.87 | 24.58 | 26.98 |
| CogVLM | 16.70 | 16.06 | 23.85 | 17.50 | 17.53 | 13.56 | 19.05 |
| CogVLM2 | 23.20 | 21.56 | 22.02 | 29.17 | 22.73 | 26.27 | 20.63 |
| GLM-4V-9B | 14.70 | 25.23 | 20.18 | 19.17 | 27.27 | 33.05 | 19.05 |

Table 3: **Results on the mathematical part of VisScience across different subjects.** Subjects: PlaneG: plane geometry, SolidG: solid geometry, Fun: functions and equations, Alg: algebraic operations, Stat: probability and statistics, Comb: combinatorial mathematics. The highest scores among close-source and open-source models are highlighted in red and blue, respectively.

benchmarks such as MathVista Lu et al. (2023), MathVerse Zhang et al. (2024b), and MATH-Vision (MATH-V) Wang et al. (2024) are specially designed to evaluate the mathematical reasoning capabilities of MLLMs. While these benchmarks are valuable, they present limitations such as an overemphasis on mathematics and a broad array of topics that often lack depth in science-related questions and exhibit uneven difficulty levels. Our dataset addresses these shortcomings by

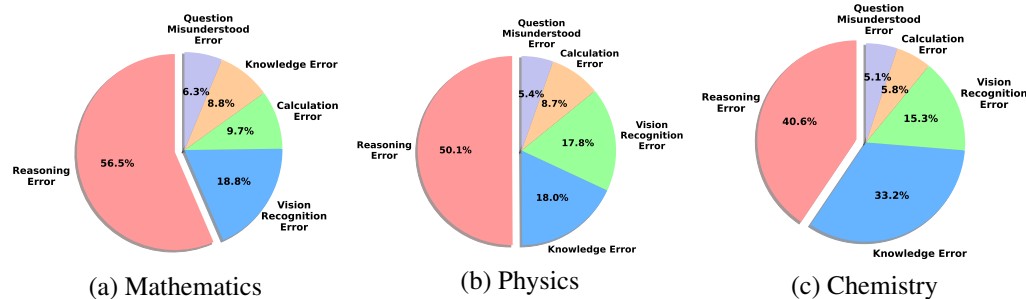

(a) Mathematics        (b) Physics        (c) Chemistry

Figure 3: Error distributions of GPT-4o on VisScience across the disciplines of mathematics, physics, and chemistry.

providing 3,000 scientific reasoning questions across mathematics, physics, and chemistry, which is collected from K12 education. Additionally, it includes bilingual questions in Chinese and English, enriching the knowledge base and offering a more extensive range of difficulty levels to create a more comprehensive evaluation platform.

**Multi-modal Large Language Models.** Recently, the success of large language models (LLMs) Du et al. (2021); Zeng et al. (2022); Achiam et al. (2023); Gao et al. (2023); GLM et al. (2024); Bai et al. (2023a) has spurred the ongoing development of multi-modal large language models (MLLMs). These MLLMs Liu et al. (2024b); Liu et al.; Wang et al. (2023a); Li et al. (2023b); Dai et al. (2024); Bai et al. (2023a) expand upon traditional LLM capabilities by integrating the ability to process and analyze both text and images. For instance, models like miniGPT Zhu et al. (2023) and InstructBLIP Dai et al. (2024) attempt to utilize a trainable Q-Former or a linear layer to connect a frozen pretrained vision encoder and language model. Subsequently, LLaVA Liu et al. (2024b); Liu et al. presents visual instruction tuning, which achieves a end-to-end fine-tuning on a large multi-modal model (LMM) comprising visual encoder and language model. Currently, close-source MLLMs like Gemini Team et al. (2023), GPT-4v OpenAI (2023), Qwen-VL Bai et al. (2023b), and Claude3 Anthropic (2024) demonstrate impressive capabilities in general image understanding and scientific reasoning. Besides, the development of open-source multi-modal large language models (MLLMs) continues to expand, providing an important complement to their closed-source models. These open-source MLLMs, such as mPLUG-Owl Ye et al. (2023; 2024), LLaMA-Adapter-V2 Gao et al. (2023), MiniCPM Hu et al. (2024), LLaVA-1.5 Liu et al. (2024a), LLaVA-NeXT Liu et al., DeepSeek-VL Lu et al. (2024), ShareGPT4V Chen et al. (2023a), SPHINX Gao et al. (2024), InternVL Chen et al. (2023b), InternVL 1.5 Chen et al. (2024b), InternLM-XComposer2 Dong et al. (2024), and CogVLM Wang et al. (2023a), also achieves advance performance, further enriching the landscape of MLLM domain. Here, we utilize our specially curated benchmark VisScience to evaluate these MLLMs across tasks in mathematics, physics, and chemistry. This comprehensive evaluation aims to assess their capabilities in scientific reasoning.

## 5 CONCLUSION

In this paper, we introduce a comprehensive benchmark, VisScience, designed to evaluate the capabilities of multi-modal large language models (MLLMs) in scientific reasoning across mathematics, physics, and chemistry. VisScience consists of 3,000 questions, evenly distributed across these three disciplines, spanning 21 subjects and categorized into five difficulty levels. We conduct evaluations using VisScience on 25 prominent models, including both closed-source and open-source variants. The experimental results reveal that closed-source MLLMs generally excel over open-source models, particularly in complex problem-solving and analytical reasoning. Notable performances include Claude3.5-Sonnet with a 53.4% accuracy in mathematics, GPT-4o achieving 38.2% in physics, and Gemini-1.5-Pro securing 30.1% in chemistry. Despite the performance gap between open-source models and closed-source models, some open-source models, such as InternVL-1.2-Plus, demonstrate competitive strengths, outperforming others like Gemini-1.0-Pro in all three disciplines. By offering a challenging set of questions across varied scientific fields, VisScience provides a robust benchmark for assessing the scientific reasoning abilities of MLLMs.

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

## A  DATASET DETAILS

### A.1  QUESTION LENGTH DISTRIBUTION

We provide both Chinese and English versions of the VisScience benchmark. The Chinese version features an average of 162.85 words per question, with the longest question comprising 1,297 words. Answers in this version average 20.93 words, with the longest reaching 112 words. Conversely, the English version shows an average of 80.93 words per question, with the longest question spanning 418 words. Answers here average 12.3 words, with the most detailed answer containing 92 words. Figure 4 depicts the distribution of word counts, highlighting the diversity and complexity of questions.

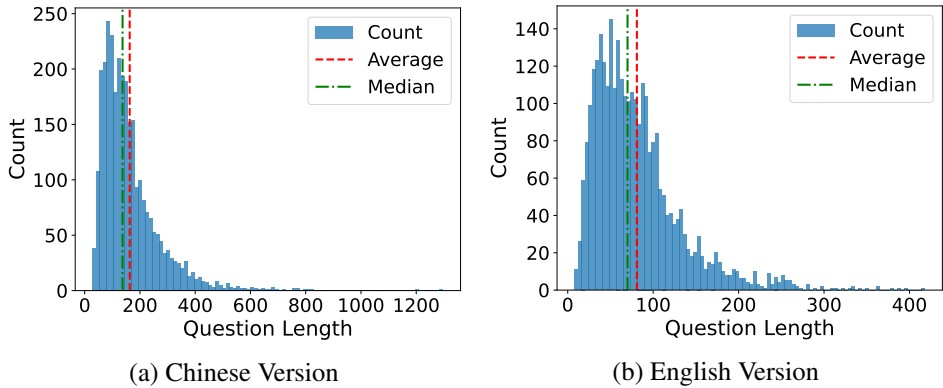

(a) Chinese Version          (b) English Version

Figure 4: The distributions of word counts per question in the Chinese and English versions of VisScience.

### A.2  DETAILED DESCRIPTION OF SUBJECTS

VisScience consists of three disciplines: mathematics, physics, and chemistry. The mathematics section includes six subjects: *algebraic operations, combinatorial mathematics, functions and equations, probability and statistics, plane geometry, and solid geometry*. The physics section is composed of eight subjects: *mechanics, optics, modern physics, mechanical motion, electromagnetism, vibrations and waves, comprehensive experiments and methods, and thermodynamics*. The chemistry section includes seven subjects: *chemical experiments, chemical reactions, inorganic chemistry, organic chemistry, electrochemistry, substance composition, and chemical equilibrium*. A more detailed introduction of the above subjects is presented as follows:

#### A.2.1  MATHEMATICS

**Algebraic Operations.**  Algebraic operations include the manipulation of algebraic expressions, such as addition, subtraction, multiplication, and division. They are fundamental for solving algebraic equations and inequalities and are widely applied across various fields of mathematics.

**Combinatorial Mathematics.**  Combinatorial mathematics studies the counting, arrangement, and combination of discrete structures, involving graph theory, number theory, and coding theory. It has significant applications in computer science, optimization, and probability theory.

**Functions and Equations.**  Functions and equations are core parts of mathematics, dealing with relationships between variables and their representations. Functions are mappings between inputs and outputs, while equations are equalities concerning these mappings. Mastering knowledge of functions and equations is fundamental for solving many practical problems and is widely applied in engineering, physics, and economics.

**Probability and Statistics.**  Probability and statistics study the laws of random events and methods of data analysis, including probability distributions, statistical inference, and data analysis techniques. They have broad applications in scientific research, engineering, and economics.

**Plane Geometry.** Plane geometry studies the shapes and figures in two-dimensional space, including points, lines, angles, and polygons. It is a fundamental part of mathematics education.

**Solid Geometry.** Solid geometry involves the study of geometric shapes in three-dimensional space, including points, lines, surfaces, and polyhedra. It examines the properties, volumes, and surface areas of these geometric bodies and is foundational for architecture, physics, and engineering.

### A.2.2 PHYSICS

**Mechanics.** Mechanics studies the motion of objects and the forces acting upon them, including classical mechanics, quantum mechanics, and relativistic mechanics. It is the foundation of physics and is widely applied in engineering, astronomy, and materials science.

**Optics.** Optics studies the properties of light and its interactions with matter, including reflection, refraction, interference, and diffraction. Optical technologies have broad applications in imaging, communication, and laser technology.

**Modern Physics.** Modern physics includes theories developed since the 20th century, such as quantum mechanics, relativity, and particle physics. These theories have expanded our understanding of the fundamental laws of nature.

**Mechanical Motion.** Mechanical motion studies the movement of objects under the influence of forces, including linear motion, rotational motion, and vibration. Understanding mechanical motion is fundamental for the design and analysis of mechanical systems.

**Electromagnetism.** Electromagnetism studies the interactions between electric and magnetic fields, including electrostatics, magnetic fields, and electromagnetic waves. It is the basis of modern physics and electrical engineering.

**Vibration and Waves.** Vibration and waves study vibrating systems and wave phenomena, including sound waves, light waves, and electromagnetic waves. They have broad applications in communication, acoustics, and optical technologies.

**Comprehensive Experiments and Methods.** Comprehensive experiments and methods involve using various experimental techniques and methods in physics teaching and research. They include designing and conducting experiments to observe and analyze the effects of specific variables on outcomes. Through comprehensive experiments, students can grasp the complexities of scientific research, cultivate scientific reasoning abilities, and understand the meticulousness and uncertainties of experimental work.

**Thermodynamics.** Thermodynamics studies the processes of energy transformation and transfer, including the laws of thermodynamics, thermodynamic systems, phase transitions, and heat engines. Thermodynamics is a fundamental aspect of both physics and engineering, with broad applications in energy, environmental science, and materials science. By investigating the relationship between internal and external energy of objects, thermodynamics reveals the basic principles of energy conversion and transfer in nature, providing theoretical support for the development of modern industrial technology.

### A.2.3 CHEMISTRY

**Chemical Experiment.** Chemical experiments involve studying the properties and changes of substances through experimental methods. Students learn to design experiments, observe chemical reactions, collect and analyze data, and draw conclusions in chemical experiments. Chemical experiments play a crucial role in understanding chemical theories and applying chemical knowledge.

**Chemical Reaction.** Chemical reactions study the chemical changes between substances, including reaction types, mechanisms, and rates. Understanding chemical reactions is essential for predicting

and controlling chemical processes, which have wide applications in pharmaceutical manufacturing, materials science, and environmental engineering.

**Inorganic Chemistry.** Inorganic chemistry studies the properties and reactions of non-carbon elements and their compounds. It covers a wide range of topics from metals and non-metals to transition metals and coordination compounds and is key to understanding the periodic table of elements and chemical reaction mechanisms.

**Organic Chemistry.** Organic chemistry studies the structure, properties, and reactions of carbon-containing compounds. It has significant applications in pharmaceutical chemistry, materials science, and biochemistry.

**Electrochemistry.** Electrochemistry studies the interconversion between electrical and chemical energy, including processes such as batteries, electrolysis, and electroplating. Electrochemistry has important applications in energy storage, corrosion control, and electrochemical sensors.

**Substance Composition.** Substance composition studies the chemical composition and structure of substances, including the arrangement of molecules, atoms, and ions. It has important applications in chemistry, materials science, and biology.

**Chemical Equilibrium.** Chemical equilibrium studies the behavior of chemical reactions when they reach a dynamic equilibrium state, including equilibrium constants, Le Chatelier's principle, and solubility equilibrium. Understanding chemical equilibrium is essential for predicting reaction directions and optimizing chemical processes.

## B  DATASET CASE

The VisScience dataset consists of 3,000 carefully selected high-quality questions, evenly distributed across three disciplines: mathematics, physics, and chemistry, with each comprising 1,000 questions. Each discipline within VisScience encompasses several subjects: mathematics includes six subjects, physics contains eight subjects, and chemistry comprises seven subjects. To illustrate the diversity and depth of VisScience, we provide more examples sampled from each discipline. In mathematics, six subjects include algebraic operations, combinatorial mathematics, functions and equations, probability and statistics, plane geometry, and solid geometry are illustrated in Figure 5 to Figure 10. Figure 11 to Figure 18 demonstrate eight subjects within the physics section of VisScience, comprising mechanics, optics, modern physics, mechanical motion, electromagnetism, vibrations and waves, comprehensive experiments and methods, and thermodynamics. The chemistry section includes seven subjects: chemical experiments, chemical reactions, inorganic chemistry, organic chemistry, electrochemistry, substance composition, and chemical equilibrium, which are illustrated in Figure 19 to Figure 25.

## C  COMPARISON WITH OTHER BENCHMARKS

We compare the VisScience benchmark with 5 existing benchmarks, including MathVista Lu et al. (2023), Math-Vision Wang et al. (2024), CMMMU Zhang et al. (2024a), ScienceQA Lu et al. (2022), and SciBench Wang et al. (2023b).

**VisScience vs MathVista.** MathVista is a comprehensive multi-modal benchmark for mathematical reasoning, comprising data from 28 existing datasets and 3 newly collected datasets. In MathVista, the majority of questions are annotated after collecting images, which results in a certain homogeneity within the data. In contrast, VisScience directly collects its questions from K12 education, featuring an average question length of 80.93 words. Such questions provide more contextual information, which facilitate a more thorough evaluation of the models' reasoning capabilities. Unlike MathVista that encompasses only seven subjects within mathematics, VisScience offers a far broader scope, including 22 distinct subjects across mathematics, physics, and chemistry. Furthermore, VisScience distinguishes itself by being a bilingual benchmark, including both Chinese and English versions of questions. This feature is particularly advantageous as it assesses MLLMs' capabilities in scientific reasoning across different languages.

**VisScience vs Math-Vision.** Math-Vision is a mathematics benchmark derived from 19 competitions, covering 16 topics across 5 levels of difficulty. Different from Math-Vision that collected from competitions, VisScience spans a broader educational spectrum, incorporating a natural gradient of difficulty from elementary school to high school. Furthermore, VisScience extends beyond mathematics to include questions from physics and chemistry, significantly broadening its scope and applicability. While Math-Vision primarily focuses on the unique challenges of competitive environments, VisScience is grounded in real-world educational settings.

**VisScience vs CMMMU.** CMMMU comprises 12,000 manually collected multi-modal questions from university exams, quizzes, and textbooks, which covers 6 core subjects and 30 specific fields. Similar to VisScience, CMMMU is a bilingual benchmark, offering questions in both Chinese and English. Within this dataset, only 1,601 questions are dedicated to the disciplines of mathematics, physics, and chemistry, accounting for only 13.34% of the total dataset. VisScience features a total of 3,000 questions, significantly outnumbering those in CMMMU dedicated to the same subjects. The questions in CMMMU are set at the university level, characterized by high difficulty, demanding that the model possesses substantial professional domain knowledge and expert-level reasoning abilities. In contrast, VisScience comes from K12 education, with a broader range of difficulty. This range allows VisScience to more comprehensively evaluate MLLMs' capabilities across different educational stages.

**VisScience vs ScienceQA.** ScienceQA is a newly developed benchmark featuring approximately 21,000 multimodal multiple-choice questions across a variety of science topics. In the ScienceQA dataset, 30.8% of questions incorporate both image and text contexts, providing a multimodal benchmark to test MLLMs in scientific reasoning. The questions in ScienceQA have an average length of only 12.11 words. In contrast, VisScience also serves as a benchmark for evaluating the scientific reasoning abilities of MLLMs, but it typically features longer and more textually detailed questions. Specifically, the Chinese version of VisScience has an average question length of 162.85 words, providing a more comprehensive and intricate testing ground for evaluating the depth of detailed reasoning in MLLMs. Additionally, VisScience contains mathematical problems, further enriching the benchmark's scope by testing MLLMs on their mathematical problem solving capabilities alongside their scientific reasoning.

**VisScience vs SciBench.** SciBench is a benchmark developed to evaluate the reasoning capabilities of LLMs in solving collegiate-level scientific problems within the domains of mathematics, chemistry, and physics. The majority of the data in SciBench focuses on assessing the scientific reasoning of LLMs, it only includes 177 problems that incorporate visual elements to evaluate the performance of MLLMs. In contrast, VisScience is primarily focused on multimodal scientific reasoning, covering similar subjects such as mathematics, chemistry, and physics. VisScience differentiates itself by offering a more comprehensive range of difficulty levels and subjects, making it a broader benchmark for assessing the capabilities of MLLMs in scientific reasoning.

## D    Evaluation Details

### D.1    The Sources of Models

In Table 4, we present the sources of the models tested on VisScience.

### D.2    Prompts

We introduce the prompts used to guide models in generating responses in Chain-of-Thought (CoT) settings and judging the LLMs' answers. The specific prompts can be found in Table 5.

## E    More Experimental Results

### E.1    Results on VisScience in English Version

Table 6 reports a comprehensive comparison of various models on the VisScience benchmark in the English version. The benchmark evaluates performance across three disciplines: mathematics,

| Model | Input | LLM Size | Source |
|---|---|---|---|
| *Closed Source Models* | | | |
| *Text-only LLMs* | | | |
| ChatGPT | *Q* | - | gpt-3.5-turbo |
| GPT-4 | *Q* | - | gpt-4 |
| Claude-2 | *Q* | - | claude-2 |
| *Multi-modal LLMs* | | | |
| Gemini-1.0-Pro | *Q, I* | - | gemini-pro |
| Gemini-1.5-Pro | *Q, I* | - | gemini-1.5-pro |
| GPT-4o | *Q, I* | - | gpt-4o |
| Claude3-Opus | *Q, I* | - | claude-3-opus-20240229 |
| Claude3.5-Sonnet | *Q, I* | - | claude-3-5-sonnet-2024620 |
| Qwen-VL-Plus | *Q, I* | - | qwen-vl-plus |
| Qwen-VL-Max | *Q, I* | - | qwen-vl-max |
| GLM-4V | *Q, I* | - | glm-4v |
| Step-1V | *Q, I* | - | step-1v |
| *Open Source Models* | | | |
| *General Multi-modal LLMs* | | | |
| mPLUG-Owl | *Q, I* | 7B | mPLUG-Owl |
| DeepSeek-VL | *Q, I* | 7B | deepseek-vl-7b-base |
| LLaMA-Adapter-V2 | *Q, I* | 7B | LLaMA-Adapter V2 |
| LLaVA-1.5 | *Q, I* | 7B | LLaVA-v1.5-7B |
| LLaVA-1.5 | *Q, I* | 13B | LLaVA-v1.5-13B |
| ShareGPT-4V | *Q, I* | 7B | ShareGPT4V-7B |
| ShareGPT-4V | *Q, I* | 13B | ShareGPT4V-13B |
| GLM-4v-9B | *Q, I* | 7B | GLM-4v-9B |
| SPHINX-Plus | *Q, I* | 13B | SPHINX-Plus |
| InternVL-Chat-V1.5 | *Q, I* | 20B | InternVL 1.5 |
| InternVL-1.2-Plus | *Q, I* | 34B | InternVL-Chat-V1-2-Plus |
| InternLM-XC2 | *Q, I* | 7B | InternLM-XComposer2-VL-7B |
| CogVLM | *Q, I* | 17B | CogVLM-17B |
| CogVLM2 | *Q, I* | 19B | cogvlm2-llama3-chat-19B |
| MiniCPM-Llama3-V-2_5 | *Q, I* | 19B | MiniCPM-Llama3-V 2.5 |

Table 4: The source of the models used in the evaluation.

| Task | Prompt |
|---|---|
| Response Generation | You are an exceptionally talented mathematics (physics/chemistry) instructor. Kindly furnish an elaborate, step-by-step solution to the question. |
| Answer Judgment | You are a highly skilled mathematics (physics/chemistry) teacher. I will provide you with a mathematics (physics/chemistry) problem, along with its ground answer and the model response from the model. Please determine whether the ground answer and the model response are consistent. Note that you do not need to judge the correctness of either answer, only whether they are consistent. If it is a multiple-choice question, both answers must choose the exact same option to be considered consistent. If it is a calculation problem, the relative error between the model response and the ground answer must be less than 0.05 to be considered consistent. If the problem has multiple sub-questions, each sub-question's answer must be identical for consistency. If you find them consistent, please add [Consistent] at the end of your response. If you find them inconsistent, please add [Inconsistent] at the end of your response. |

Table 5: Prompts for response generation and answer judgment.

physics, and chemistry. Among close-source models, GPT-4o demonstrates the highest performance across two disciplines, achieving an accuracy of 53.6% in mathematics and 42.7% in physics. However, Claude3.5-Sonnet surpasses GPT-4o in chemistry with a higher accuracy of 43.6%. Open-source models generally show lower performance compared to close-source counterparts. Notably, InternVL-1.2-Plus displays competitive performance, reaching up to 26.0% in mathematics, 23.6% in physics, and 27.8% in chemistry. The English version of VisScience is designed to facilitate the evaluation of MLLMs that specialize in English, assessing their capabilities in scientific reasoning.

### E.2 RESULTS ON PHYSICS ACROSS DIFFERENT SUBJECTS

Table 7 presents a detailed analysis of various models on VisScience across different subjects within the physics section, which includes mechanics, electromagnetism, thermodynamics, comprehensive experiments and methods, optics, vibration and waves, modern physics, and mechanical motion. The table highlights that while GPT-4o exhibits the top performance on the entire physics discipline, the best performance in individual subjects varies notably. For instances, Claude3.5-Sonnet excels specifically in modern physics with an accuracy of 66.67%, significantly surpassing other close-source models in this area. This variation in performance by subject underscores the specialized capabilities of different models. Moreover, this detailed analysis provides more insights, emphasizing the need for targeted improvements to achieve balanced performance across all physics subjects.

### E.3 RESULTS ON CHEMISTRY ACROSS DIFFERENT SUBJECTS

Table 8 presents a nuanced view of the performance of various models across different subjects within the chemistry discipline of the VisScience benchmark. The chemistry discipline includes chemical experiment, chemical reaction, inorganic chemistry, electrochemistry, organic chemistry, chemical equilibrium, and substance composition. Notably, Gemini-1.5-Pro stands out among close-source models, excelling across the entire chemistry discipline. It demonstrates particular prowess in organic chemistry and substance composition, achieving impressive accuracies of 57.02% and 61.16%, respectively. Additionally, Qwen-VL-Max leads in chemical experiment and inorganic chemistry, achieving the highest accuracies of 46.28% and 51.94%, respectively. Open-source models demonstrate a range of performances, with InternVL-1.2-Plus leading this group. It achieves the highest open-source accuracy in nearly all subjects. This comprehensive review of model performances within the chemistry section of the VisScience benchmark highlights the need to enhance MLLMs' capabilities in scientific domains, ensuring models are both accurate and adaptable across various disciplines.

| Model | LLM | Input | Mathematics | Physics | Chemistry |
|---|---|---|---|---|---|
| *Close Source Models* (APIs) | | | | | |
| *Text-only LLMs* | | | | | |
| Zero-shot ChatGPT | - | *Q* | 17.4 | 20.7 | 25.2 |
| Zero-shot GPT-4 | - | *Q* | 29.9 | 37.7 | 38.7 |
| Zero-shot Claude-2 | - | *Q* | 24.6 | 22.7 | 25.6 |
| Zero-shot Claude3-Opus | - | *Q* | 21.7 | 15.8 | 29.4 |
| Zero-shot Claude3.5-Sonnet | - | *Q* | 27.2 | 35.7 | 35.2 |
| Zero-shot GPT-4o | - | *Q* | 35.2 | 40.3 | 42.5 |
| 2-shot CoT Claude2 | - | *Q* | 25.7 | 21.9 | 24.1 |
| 2-shot CoT ChatGPT | - | *Q* | 24.4 | 20.1 | 22.1 |
| 2-shot CoT GPT-4 | - | *Q* | 36.5 | 39.0 | 38.1 |
| *Multi-modal LLMs* | | | | | |
| Gemini-1.0-Pro | - | *Q, I* | 26.4 | 39.1 | 27.9 |
| Gemini-1.5-Pro | - | *Q, I* | 47.8 | 35.1 | 39.1 |
| GPT-4o | - | *Q, I* | 53.6 | 42.7 | 43.3 |
| GPT-4o-mini | - | *Q, I* | 43.2 | 33.7 | 34.9 |
| Qwen-VL-Max | - | *Q, I* | 30.7 | 26.4 | 36.3 |
| Qwen-VL-Plus | - | *Q, I* | 21.9 | 20.9 | 29.7 |
| Claude3.5-Sonnet | - | *Q, I* | 50.8 | 36.6 | 43.6 |
| Claude3-Opus | - | *Q, I* | 34.4 | 29.4 | 34.7 |
| GLM-4V | - | *Q, I* | 23.1 | 18.5 | 23.4 |
| Step-1V | 7B | *Q, I* | 32.0 | 19.5 | 27.6 |
| *Open Source Models* | | | | | |
| *General Multi-modal LLMs* | | | | | |
| mPLUG-Owl | LLaMA-7B | *Q, I* | 7.4 | 12.3 | 12.3 |
| LLaMA-Adapter-V2 | LLaMA-7B | *Q, I* | 12.6 | 11.4 | 16.2 |
| MiniCPM-Llama3-V2.5 | LLaMA3-8B | *Q, I* | 24.4 | 20.6 | 24.4 |
| LLaVA-1.5 | Vicuna-13B | *Q, I* | 15.0 | 17.4 | 21.1 |
| LLaVA-1.5 | Vicuna-7B | *Q, I* | 17.4 | 16.6 | 18.9 |
| DeepSeek-VL | DeepSeek-LLM-7B | *Q, I* | 16.0 | 16.9 | 17.8 |
| ShareGPT-4V | Vicuna-7B | *Q, I* | 14.7 | 17.7 | 21.3 |
| ShareGPT-4V | Vicuna-13B | *Q, I* | 14.5 | 16.0 | 20.2 |
| SPHINX-Plus | LLaMA2-13B | *Q, I* | 17.9 | 15.7 | 22.4 |
| InternLM-XC2 | InternLM2-7B | *Q, I* | 20.7 | 20.5 | 25.0 |
| InternVL-1.2-Plus | Nous-Hermes-2-Yi-34B | *Q, I* | 26.0 | 23.6 | 27.8 |
| InternVL-Chat-V1.5 | Mixtral 8*7B | *Q, I* | 24.9 | 23.0 | 25.9 |
| CogVLM | Vicuna-7B | *Q, I* | 18.5 | 15.9 | 23.1 |
| CogVLM2 | LLaMA-3-8B | *Q, I* | 24.2 | 16.6 | 24.9 |
| GLM-4V-9B | GLM-4-9B | *Q, I* | 24.7 | 19.2 | 23.9 |

Table 6: **Results on VisScience within the version of the English language across the disciplines of mathematics, physics, and chemistry.** The highest scores among close-source and open-source models are highlighted in red and blue, respectively.

| Model | ALL | Mech | Ele | Therm | Comp | Physics Opt | Vib & Waves | Mod Phys | Mech Motion |
|---|---|---|---|---|---|---|---|---|---|
| *Close Source Models* (APIs) | | | | | | | | | |
| *Text-only LLMs* | | | | | | | | | |
| Zero-shot ChatGPT | 22.70 | 22.08 | 19.94 | 23.53 | 4.62 | 40.98 | 29.79 | 19.05 | 23.33 |
| Zero-shot GPT-4 | 30.40 | 34.26 | 30.21 | 33.33 | 15.38 | 40.98 | 34.04 | 42.86 | 20.00 |
| Zero-shot Claude-2 | 22.00 | 24.62 | 23.56 | 25.49 | 12.31 | 27.87 | 21.28 | 28.57 | 23.33 |
| Zero-shot Claude3-Opus | 30.80 | 34.26 | 32.02 | 33.33 | 10.77 | 39.34 | 31.91 | 42.86 | 10.00 |
| Zero-shot Claude3.5-Sonnet | 35.30 | 40.36 | 35.95 | 35.29 | 15.38 | 40.98 | 34.04 | 47.62 | 26.67 |
| Zero-shot GPT-4o | 38.00 | 43.91 | 38.67 | 45.10 | 9.23 | 49.18 | 38.30 | 52.38 | 23.33 |
| 2-shot CoT Claude2 | 21.70 | 24.87 | 22.96 | 22.96 | 10.77 | 18.03 | 23.40 | 28.57 | 10.00 |
| 2-shot CoT ChatGPT | 18.60 | 20.30 | 20.54 | 13.73 | 12.31 | 22.95 | 23.40 | 23.81 | 13.33 |
| 2-shot CoT GPT-4 | 31.50 | 35.03 | 32.02 | 37.25 | 12.31 | 44.26 | 29.79 | 47.62 | 23.33 |
| *Multi-modal LLMs* | | | | | | | | | |
| Gemini-1.0-Pro | 23.70 | 26.97 | 23.03 | 17.65 | 6.15 | 31.15 | 34.04 | 19.05 | 10.00 |
| Gemini-1.5-Pro | 38.10 | 46.56 | 33.74 | 47.06 | 20.00 | 45.00 | 34.04 | 52.38 | 43.33 |
| GPT-4o | 38.20 | 41.37 | 39.27 | 56.86 | 23.08 | 42.62 | 36.17 | 42.86 | 43.33 |
| GPT-4o-mini | 29.80 | 31.73 | 30.51 | 29.41 | 10.70 | 36.07 | 19.15 | 47.62 | 30.00 |
| Qwen-VL-Max | 30.70 | 36.13 | 26.59 | 39.22 | 9.23 | 34.43 | 31.91 | 28.57 | 30.00 |
| Qwen-VL-Plus | 26.50 | 31.04 | 24.77 | 33.33 | 6.15 | 36.07 | 36.17 | 23.81 | 16.67 |
| Claude3.5-Sonnet | 38.00 | 41.62 | 36.56 | 43.14 | 13.85 | 44.26 | 38.30 | 66.67 | 30.00 |
| Claude3-Opus | 31.10 | 33.25 | 29.91 | 39.22 | 12.31 | 45.90 | 34.04 | 61.90 | 23.33 |
| GLM-4V | 19.20 | 23.16 | 17.82 | 15.69 | 12.31 | 25.00 | 17.02 | 19.05 | 23.33 |
| Step-1V | 23.50 | 21.55 | 24.35 | 28.57 | 7.84 | 12.82 | 25.00 | 31.25 | 39.13 |
| *Open Source Models* | | | | | | | | | |
| *General Multi-modal LLMs* | | | | | | | | | |
| mPLUG-Owl | 8.30 | 11.93 | 8.46 | 1.96 | 4.62 | 8.20 | 10.64 | 4.76 | 10.00 |
| LLaMA-Adapter-V2 | 10.30 | 10.41 | 10.88 | 8.00 | 4.84 | 13.11 | 25.53 | 14.29 | 3.33 |
| MiniCPM-Llama3-V2.5 | 17.90 | 21.57 | 19.64 | 15.69 | 6.15 | 26.23 | 19.15 | 9.52 | 23.33 |
| LLaVA-1.5-13B | 15.20 | 17.26 | 14.80 | 7.84 | 7.69 | 21.31 | 17.02 | 9.52 | 16.67 |
| LLaVA-1.5-7B | 13.50 | 15.28 | 15.12 | 11.76 | 3.12 | 15.25 | 15.56 | 5.26 | 17.24 |
| DeepSeek-VL | 16.80 | 18.77 | 19.33 | 13.73 | 7.69 | 16.67 | 13.04 | 19.05 | 3.45 |
| ShareGPT4V-7B | 14.00 | 13.71 | 15.41 | 9.80 | 3.08 | 19.67 | 19.15 | 28.57 | 6.67 |
| ShareGPT4V-13B | 14.90 | 15.23 | 16.92 | 9.80 | 6.15 | 14.75 | 19.15 | 19.05 | 16.67 |
| SPHINX-Plus | 15.30 | 16.50 | 18.43 | 17.65 | 4.62 | 11.48 | 12.77 | 19.05 | 13.33 |
| InternLM-XC2 | 18.30 | 20.81 | 17.82 | 13.73 | 10.77 | 26.23 | 21.28 | 14.29 | 6.67 |
| InternVL-1.2-Plus | 24.80 | 29.69 | 22.94 | 29.41 | 12.31 | 31.67 | 25.53 | 35.00 | 10.00 |
| InternVL-Chat-V1.5 | 20.80 | 23.97 | 20.87 | 23.53 | 9.23 | 25.42 | 17.02 | 14.29 | 17.24 |
| CogVLM | 14.50 | 18.02 | 13.29 | 7.84 | 6.15 | 14.75 | 19.15 | 19.05 | 6.67 |
| CogVLM2 | 14.40 | 16.75 | 16.00 | 12.00 | 6.15 | 13.11 | 19.15 | 4.76 | 10.00 |
| GLM-4V-9B | 19.30 | 21.78 | 21.12 | 24.00 | 4.62 | 25.42 | 15.91 | 15.00 | 13.33 |

Table 7: **Results on the physics part of VisScience across different subjects.** Subjects: Mech: mechanics, Ele: electromagnetism, Threm: thermodynamics, Comp: comprehensive experiments and methods, Opt: optics, Vib & Waves: vibration and waves, Mod Phys: modern physics, Mech Motion: mechanical motion. The highest scores among close-source and open-source models are highlighted in red and blue, respectively.

## F  CASE STUDY

The VisScience dataset includes three disciplines: mathematics, physics, and chemistry. The mathematical section comprises 6 subjects, the chemistry section contains 7 subjects, and the physics section includes 8 subjects, culminating in a total of 21 distinct subjects across the VisScience benchmark. Here, we present one question from each subject, along with its standard answer and the correct response provided by GPT-4o. Figure 26 to Figure 28 demonstrate cases of the mathematical part of VisScience. Figure 29 to Figure 32 illustrate sampled questions from the physics section of the VisScience benchmark. Each figure provides insight into the diverse range of topics covered, showing GPT-4o's capabilities to handle complex physical principles and calculations. Figure 33 to Figure 36 display examples from the chemistry section. These examples not only demonstrate the diversity of the VisScience benchmark within chemistry but also illustrate how effectively GPT-4o can generate accurate responses across different scientific subjects.

| Model | ALL | Chemistry | | | | | | |
| | | Chem Exp | Chem React | Inorg Chem | Electrochem | Org Chem | Chem Equil | Sub Comp |
|---|---|---|---|---|---|---|---|---|
| *Close Source Models* (APIs) | | | | | | | | |
| *Text-only LLMs* | | | | | | | | |
| Zero-shot ChatGPT | 18.60 | 26.35 | 23.86 | 23.26 | 23.75 | 35.43 | 24.64 | 30.89 |
| Zero-shot GPT-4 | 33.10 | 40.54 | 30.68 | 38.76 | 32.50 | 36.22 | 30.43 | 31.71 |
| Zero-shot Claude-2 | 24.40 | 24.32 | 26.36 | 31.71 | 23.86 | 20.29 | 0.30 | 25.98 |
| Zero-shot Claude3-Opus | 32.50 | 37.16 | 30.68 | 31.78 | 31.25 | 36.22 | 30.43 | 39.84 |
| Zero-shot Claude3.5-Sonnet | 36.90 | 34.80 | 36.93 | 39.53 | 46.25 | 45.67 | 23.19 | 47.15 |
| Zero-shot GPT-4o | 39.60 | 42.57 | 40.34 | 44.96 | 35.00 | 41.73 | 26.09 | 54.47 |
| 2-shot CoT Claude2 | 23.90 | 23.99 | 26.70 | 22.48 | 30.00 | 26.77 | 27.54 | 27.64 |
| 2-shot CoT ChatGPT | 21.30 | 19.93 | 23.30 | 20.93 | 22.50 | 22.83 | 26.09 | 30.08 |
| 2-shot CoT GPT-4 | 32.40 | 29.05 | 32.39 | 32.56 | 32.50 | 42.52 | 28.99 | 53.66 |
| *Multi-modal LLMs* | | | | | | | | |
| Gemini-1.0-Pro | 27.80 | 24.03 | 26.70 | 26.36 | 31.25 | 35.54 | 31.82 | 37.19 |
| Gemini-1.5-Pro | 47.00 | 43.46 | 47.43 | 51.59 | 50.00 | 57.02 | 35.29 | 61.16 |
| GPT-4o | 41.60 | 43.58 | 46.02 | 38.76 | 46.25 | 43.31 | 43.48 | 50.41 |
| GPT-4o-mini | 28.40 | 22.30 | 27.27 | 27.13 | 30.00 | 34.65 | 20.29 | 42.09 |
| Qwen-VL-Max | 42.50 | 46.28 | 41.48 | 51.94 | 35.00 | 41.73 | 36.23 | 53.66 |
| Qwen-VL-Plus | 37.70 | 33.78 | 40.34 | 44.19 | 41.25 | 48.03 | 33.33 | 41.80 |
| Claude3.5-Sonnet | 43.10 | 40.54 | 41.48 | 42.64 | 50.00 | 42.52 | 33.33 | 59.35 |
| Claude3-Opus | 34.10 | 35.47 | 30.11 | 31.78 | 31.25 | 40.16 | 33.33 | 51.22 |
| GLM-4V | 25.00 | 23.65 | 25.86 | 21.71 | 28.75 | 27.78 | 31.88 | 32.52 |
| Step-1V | 25.00 | 32.51 | 27.48 | 25.26 | 25.45 | 17.72 | 13.33 | 21.95 |
| *Open Source Models* | | | | | | | | |
| *General Multi-modal LLMs* | | | | | | | | |
| mPLUG-Owl | 9.50 | 7.77 | 11.36 | 7.75 | 12.50 | 12.60 | 13.04 | 9.76 |
| LLaMA-Adapter-V2 | 10.80 | 7.77 | 13.64 | 8.53 | 12.66 | 12.80 | 17.65 | 17.07 |
| MiniCPM-Llama3-V2.5 | 19.50 | 20.96 | 26.29 | 26.61 | 18.18 | 24.00 | 28.79 | 30.83 |
| LLaVA-1.5-13B | 18.80 | 15.54 | 16.48 | 24.03 | 20.00 | 22.05 | 23.19 | 19.51 |
| LLaVA-1.5-7B | 16.00 | 13.49 | 17.14 | 19.20 | 16.25 | 20.49 | 26.09 | 10.74 |
| DeepSeek-VL | 21.00 | 18.84 | 20.57 | 20.16 | 21.25 | 23.62 | 36.76 | 20.66 |
| ShareGPT4V-7B | 19.00 | 13.85 | 19.32 | 26.36 | 18.75 | 23.62 | 28.99 | 15.45 |
| ShareGPT4V-13B | 18.40 | 13.51 | 21.02 | 19.38 | 23.75 | 22.83 | 13.04 | 19.51 |
| SPHINX-Plus | 20.40 | 20.27 | 21.02 | 24.03 | 22.50 | 22.83 | 27.54 | 21.95 |
| InternLM-XC2 | 25.60 | 22.64 | 27.27 | 26.36 | 21.25 | 33.86 | 26.09 | 24.39 |
| InternVL-1.2-Plus | 31.20 | 22.29 | 31.82 | 33.58 | 31.46 | 39.57 | 32.47 | 38.84 |
| InternVL-Chat-V1.5 | 23.70 | 20.07 | 25.00 | 25.20 | 22.37 | 28.80 | 25.00 | 28.46 |
| CogVLM | 17.00 | 15.54 | 20.45 | 10.85 | 16.25 | 22.05 | 20.29 | 17.07 |
| CogVLM2 | 21.00 | 13.10 | 21.39 | 25.78 | 20.51 | 31.45 | 22.73 | 30.17 |
| GLM-4V-9B | 22.50 | 21.00 | 25.44 | 26.23 | 23.08 | 26.83 | 17.39 | 25.83 |

Table 8: **Results on the chemistry part of VisScience across different subjects.** Subjects: Chem Exp: chemical experiment, Chem React: chemical reaction, Inorg Chem: inorganic chemistry, Electrochem: Electrochemistry, Org Chem: organic chemistry, Chem Equil: chemical equilibrium, and Sub Comp: substance composition. The highest scores among close-source and open-source models are highlighted in red and blue, respectively.

# G    ERROR CASE

We conduct rigorous tests on a series of open-source and close-source models on VisScience and perform a detailed analysis of the models' responses. These errors in the models' answers can be classified into five categories: reasoning error, vision recognition error, knowledge error, calculation error, and question misunderstood error. We present examples of these five error types across the disciplines of mathematics, physics, and chemistry, with a specific focus on errors made by GPT-4o. Additionally, we demonstrate error examples from other representative close-source models such as GLM-4V, Qwen-VL-max, and Claude 3.5, as well as open-source models like LLAVA-1.5 , GLM-4V-9B and InternVL-Chat-1.5. Notably, it should be noted that the types of errors made by these models in response to the same questions can differ from those made by GPT-4o. This analysis helps to underline the varied challenges faced by different models in processing complex scientific questions, providing insight into their respective strengths and limitations. Figure 37 to Figure 46 demonstrate cases of errors from representative models in the mathematical part of VisScience. Figure 47 to Figure 59 show the incorrect answers in the physics section. Figure 60 to Figure 68 demonstrate the errors in the chemistry section.

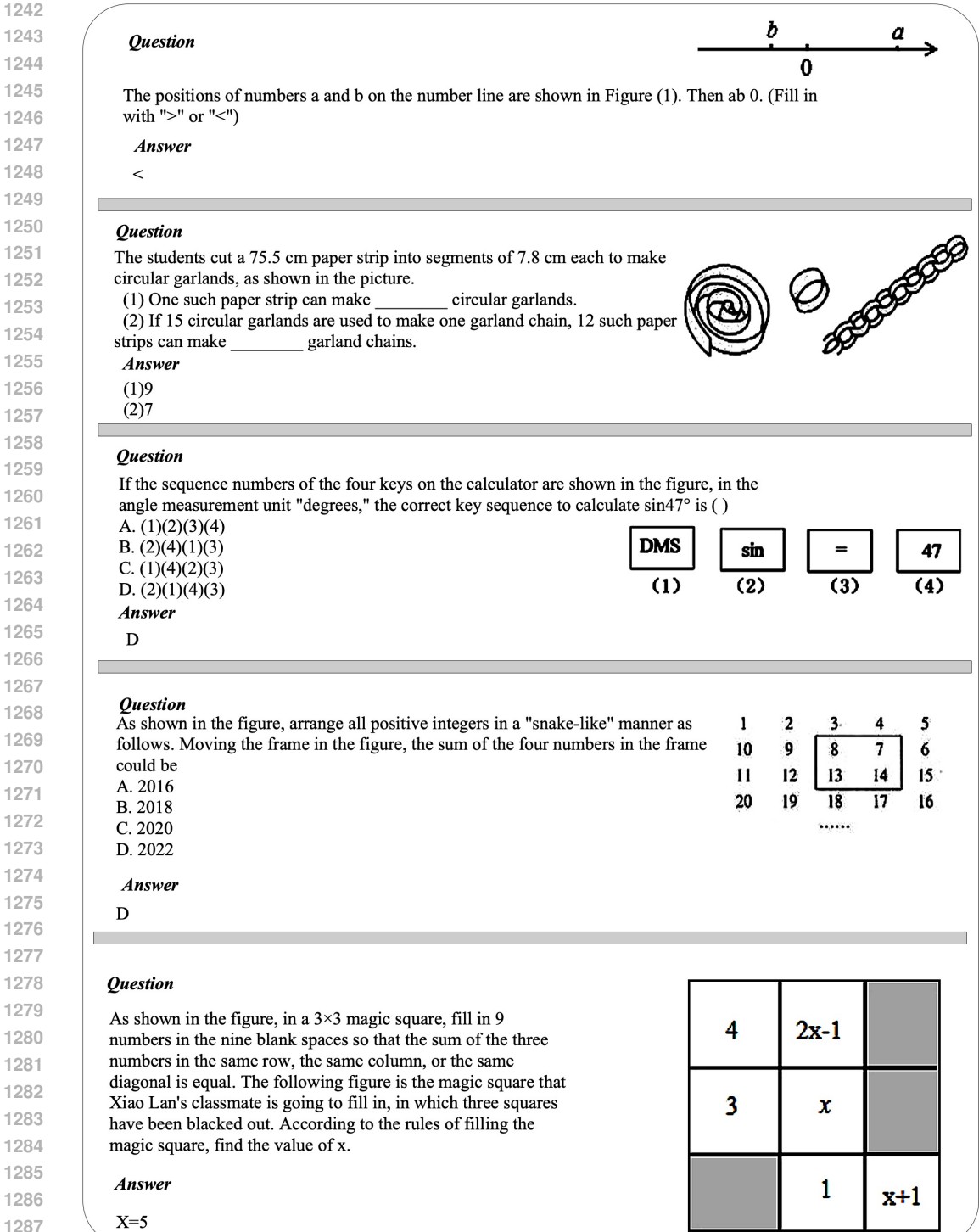

**Question**

The positions of numbers a and b on the number line are shown in Figure (1). Then ab 0. (Fill in with ">" or "<")

**Answer**

<

**Question**

The students cut a 75.5 cm paper strip into segments of 7.8 cm each to make circular garlands, as shown in the picture.
(1) One such paper strip can make _________ circular garlands.
(2) If 15 circular garlands are used to make one garland chain, 12 such paper strips can make _________ garland chains.

**Answer**

(1)9
(2)7

**Question**

If the sequence numbers of the four keys on the calculator are shown in the figure, in the angle measurement unit "degrees," the correct key sequence to calculate sin47° is ( )
A. (1)(2)(3)(4)
B. (2)(4)(1)(3)
C. (1)(4)(2)(3)
D. (2)(1)(4)(3)

**Answer**

D

**Question**

As shown in the figure, arrange all positive integers in a "snake-like" manner as follows. Moving the frame in the figure, the sum of the four numbers in the frame could be
A. 2016
B. 2018
C. 2020
D. 2022

**Answer**

D

**Question**

As shown in the figure, in a 3×3 magic square, fill in 9 numbers in the nine blank spaces so that the sum of the three numbers in the same row, the same column, or the diagonal is equal. The following figure is the magic square that Xiao Lan's classmate is going to fill in, in which three squares have been blacked out. According to the rules of filling the magic square, find the value of x.

**Answer**

X=5

Figure 5: Cases of *algebraic operations* in mathematical part of VisScience.

**Question**

A certain city is building a flowerbed in the central square. The flowerbed is divided into 6 parts (as shown in the figure). Now, 4 different colors of flowers need to be planted, one type in each part, and no adjacent parts can be planted with the same color. How many different planting methods are there (answer with a number).

**Answer**

120

**Question**

As shown in the picture, A drives from Longgang to Nanshan and must pass through Buji. It is known that there are three roads to choose from Longgang to Buji and two roads to choose from Buji to Nanshan. A has a total of ( ) routes to choose from.
A. 5  B. 6  C. 4  D. 9

龙岗 ————— 布吉 ————— 南山

**Answer**

B

**Question**

刷牙洗脸：6分钟    读书：10分钟    煮面条：10分钟    吃面条：8分钟

As shown in the figure, after Xiao Yi gets up in the morning, she has to do the following tasks. To minimize the time required, the minimum time needed is ( ______ ) minutes.

**Answer**

24

**Question**

As shown in the figure: Color the six points ABCDEF in the figure with four different colors, requiring each point to be colored with one color, and the two endpoints of each line segment in the figure to be colored differently. The number of different coloring methods is . (Answer with a number)

**Answer**

264

**Question**

As shown in the figure, small circles represent network nodes, the lines between the nodes indicate they have network connections, and the numbers on the lines show the maximum information flow through that segment of the network per unit time. If information is transmitted from node A to node B and can be split to travel along different routes simultaneously, the maximum amount of information that can be transmitted per unit time is (  )
A. 26
B. 24
C. 20
D. 19

**Answer**

D

Figure 6: Cases of *combinatorial mathematics* in mathematical part of VisScience.

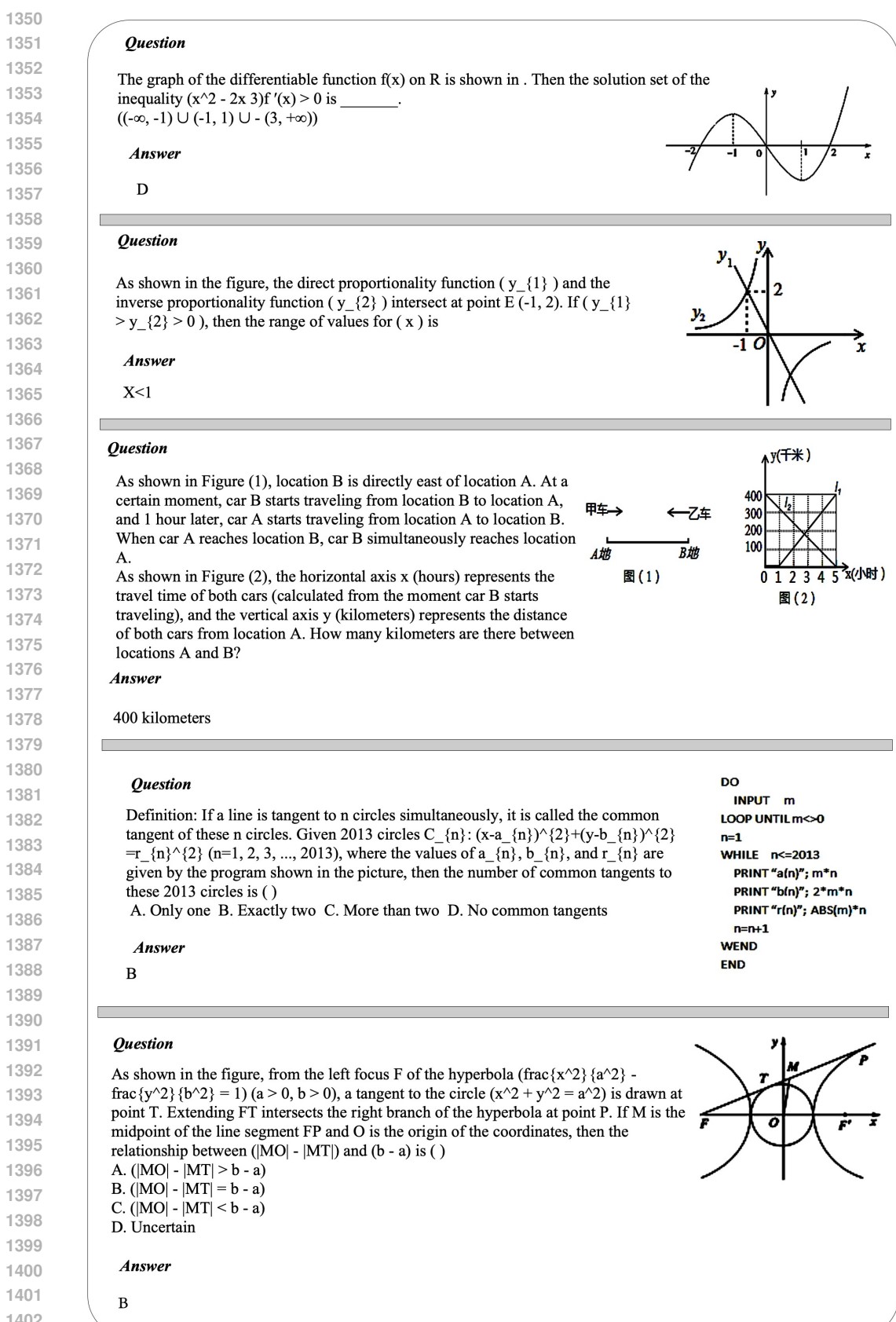

**Question**

The graph of the differentiable function f(x) on R is shown in . Then the solution set of the inequality (x^2 - 2x 3)f '(x) > 0 is ________.
((-∞, -1) ∪ (-1, 1) ∪ - (3, +∞))

**Answer**

D

**Question**

As shown in the figure, the direct proportionality function ( y_{1} ) and the inverse proportionality function ( y_{2} ) intersect at point E (-1, 2). If ( y_{1} > y_{2} > 0 ), then the range of values for ( x ) is

**Answer**

X<1

**Question**

As shown in Figure (1), location B is directly east of location A. At a certain moment, car B starts traveling from location B to location A, and 1 hour later, car A starts traveling from location A to location B. When car A reaches location B, car B simultaneously reaches location A.
As shown in Figure (2), the horizontal axis x (hours) represents the travel time of both cars (calculated from the moment car B starts traveling), and the vertical axis y (kilometers) represents the distance of both cars from location A. How many kilometers are there between locations A and B?

**Answer**

400 kilometers

**Question**

Definition: If a line is tangent to n circles simultaneously, it is called the common tangent of these n circles. Given 2013 circles C_{n}: (x-a_{n})^{2}+(y-b_{n})^{2}=r_{n}^{2} (n=1, 2, 3, ..., 2013), where the values of a_{n}, b_{n}, and r_{n} are given by the program shown in the picture, then the number of common tangents to these 2013 circles is ( )
 A. Only one  B. Exactly two  C. More than two  D. No common tangents

**Answer**

B

**Question**

As shown in the figure, from the left focus F of the hyperbola (frac{x^2}{a^2} - frac{y^2}{b^2} = 1) (a > 0, b > 0), a tangent to the circle (x^2 + y^2 = a^2) is drawn at point T. Extending FT intersects the right branch of the hyperbola at point P. If M is the midpoint of the line segment FP and O is the origin of the coordinates, then the relationship between (|MO| - |MT|) and (b - a) is ( )
A. (|MO| - |MT| > b - a)
B. (|MO| - |MT| = b - a)
C. (|MO| - |MT| < b - a)
D. Uncertain

**Answer**

B

Figure 7: Cases of *functions and equations* in mathematical part of VisScience.

**Question**

As shown in Figure ①, there is an irregular pattern (the shaded part in the figure) on a flat ground. Xiaoming wants to know the area of this pattern, so he took the following approach: he surrounded the irregular pattern with a rectangle of length 5m and width 4m, and then randomly threw small balls at the rectangular area and recorded the number of times the balls landed on the irregular pattern (balls landing on the boundary line or outside the rectangular area were not counted). He plotted the results of several valid trials into a line graph as shown in Figure ②, from which he estimated the area of the irregular pattern to be approximately (   )

A. 6m²
B. 7m²
C. 8m²
D. 9m²

**Answer**

X<1

**Question**

In a vegetable plot, green peppers, cucumbers, loofahs, and eggplants are planted. As shown in the image, it represents the planting area percentage of each vegetable relative to the total area.

(1) The planting area of eggplants accounts for ( ______ ) % of the total vegetable plot area.

(2) If the planting area of loofahs is 30m², the total area of this vegetable plot is ( ______ ) m².

(3) ( ______ ) has the largest planting area, which is greater than the planting area of loofahs by ( ______ ) % of the total vegetable plot area.

**Answer**

15 120 cucumber 15

**Question**

Image recognition:
Please carefully observe the highest and lowest temperatures of a certain city for the coming week as given in the picture, and directly answer the following questions:

(1) What are the highest and lowest temperatures in the city this week?
(2) On which day of the week is the temperature difference the largest, and what is the difference?

**Answer**

The highest temperature is 9°C, the lowest temperature is -4°C, and the temperature difference on Thursday this week is the largest, with a maximum difference of 8°C.

**Question**

The Municipal Traffic Police Brigade conducted traffic safety knowledge publicity for the students of a certain school. They randomly surveyed the instances of jaywalking among these students anonymously and created a statistical chart as shown in the figure. Please answer the following questions based on the information in the chart:
(1) How many students were surveyed in this survey?
(2) If the total number of students is 1500, estimate how many of them frequently jaywalk.
(3) Share your insights based on the information reflected in the chart. (No more than 30 words).

**Answer**
(1) 100
(2) 225
(3) You can simply share your own understanding based on the actual situation; there isn't a single correct answer.

Figure 8: Cases of *probability and statistics* in mathematical part of VisScience.

**Question**

As shown in the figure, rectangle ABCD, R is the midpoint of CD, point M moves along the side BC, E and F are the midpoints of AM and MR respectively. Then the length of EF with the movement of point M(  )

A.Becomes shorterB. Becomes longerC. Remains the sameD.Cannot be determined

**Answer**

C

**Question**

As shown in the figure, AB is a chord of the circle O. Point C is a moving point on the circle O, and ∠ACB=30°. Points E and F are the midpoints of AC and BC, respectively. The line EF intersects the circle O at points G and H. If the radius of the circle O is 7, then the maximum value of GE+FH is

**Answer**

10.5

**Question**

Given as shown in the image, in right triangles ABC and DAE, ∠BAC = 90°, ∠ADE = 90°, ∠B = 60°, and ∠E = 45°, with AE ∥ BC. Side AC intersects side DE at point F. Find the degree of ∠AFD

**Answer**

75°

**Question**

Exploration and Discovery:

The shape shown in Figure 1 resembles a common learning tool—a compass. Let us call this shape a "compass diagram." What kind of mathematical knowledge is hidden in this simple diagram? Now, use your intelligence and solve the following problems:

(1) Observe the "compass diagram" and explore the relationship among ∠BDC and ∠A, ∠B, ∠C, and explain the reasoning;

(2) Directly use the above conclusion to solve the following three problems:

① As shown in Figure 2, place a set square XYZ on △ABC such that the two right-angle edges XY, XZ just pass through points B and C. If ∠A=50°, then ∠ABX+∠ACX= °;

② As shown in Figure 3, DC bisects ∠ADB, and EC bisects ∠AEB. If ∠DAE=50° and ∠DBE=130°, find the measure of ∠DCE;

③ As shown in Figure 4, the 10 division lines of ∠ABD and ∠ACD intersect at points G_{1}, G_{2}…G_{9}. If ∠BDC=140° and ∠BG_{1}C=77°, find the measure of ∠A.

**Answer**

(1) ∠BDC=∠A+∠B+∠C; (2) ①40°; ②90°; ③70°

**Question**

In a square grid where the side length of each small square is 1 unit, the positions of the three vertices of △ABC are as shown in the figure. △ABC is now translated so that point A is moved to point D, and points E and F correspond to points B and C, respectively.

(1) Please draw the translated △DEF and find the area of △DEF;

(2) If AD and CF are connected, the relationship between these two segments is .

**Answer**

(1) The area of △DEF = 7;

(2) Parallel and equal.

Figure 9: Cases of *plane geometry* in mathematical part of VisScience.

**Question**

As shown in the figure, if the container is turned upside down, what is the height of the water surface in centimeters?

**Answer**

11cm

---

**Question**

As shown in the figure, point P is outside the plane of square ABCD. PD is perpendicular to plan ABCD, and PD equals AD. The angle between PA and BD is

**Answer**

60°

---

**Question**

As shown in the figure, in the cube ABCD-EFGH, the following statements are incorrect:
A. BD // Face FHA
B. EC ⊥ BD
C. EC ⊥ Face FHA
D. The angle between the skew lines BC and AH is 60°

**Answer**

D

---

**Question**

As shown in the figure, the remaining part after a cone is removed from a cylinder is to be calculated. Please calculate its volume.

**Answer**

1884 cm3

---

**Question**

As shown in the figure, in the rhombus ABCD with side length 4 and ∠DAB = 60°, points E and F are on sides CD and CB respectively, and point E does not coincide with points C and D. EF is perpendicular to AC, and EF intersects AC at O. By folding △CEF along EF to the position of △PEF, the plane PEF is perpendicular to the plane ABFED.Let the volume of the triangular pyramid P-ABD be $V_{1}$, and the volume of the quadrilateral pyramid P-BDEF be $V_{2}$. Find the value of $V_{1}:V_{2}$ when PB is minimized.

**Answer**

4:3

Figure 10: Cases of *solid geometry* in mathematical part of VisScience.

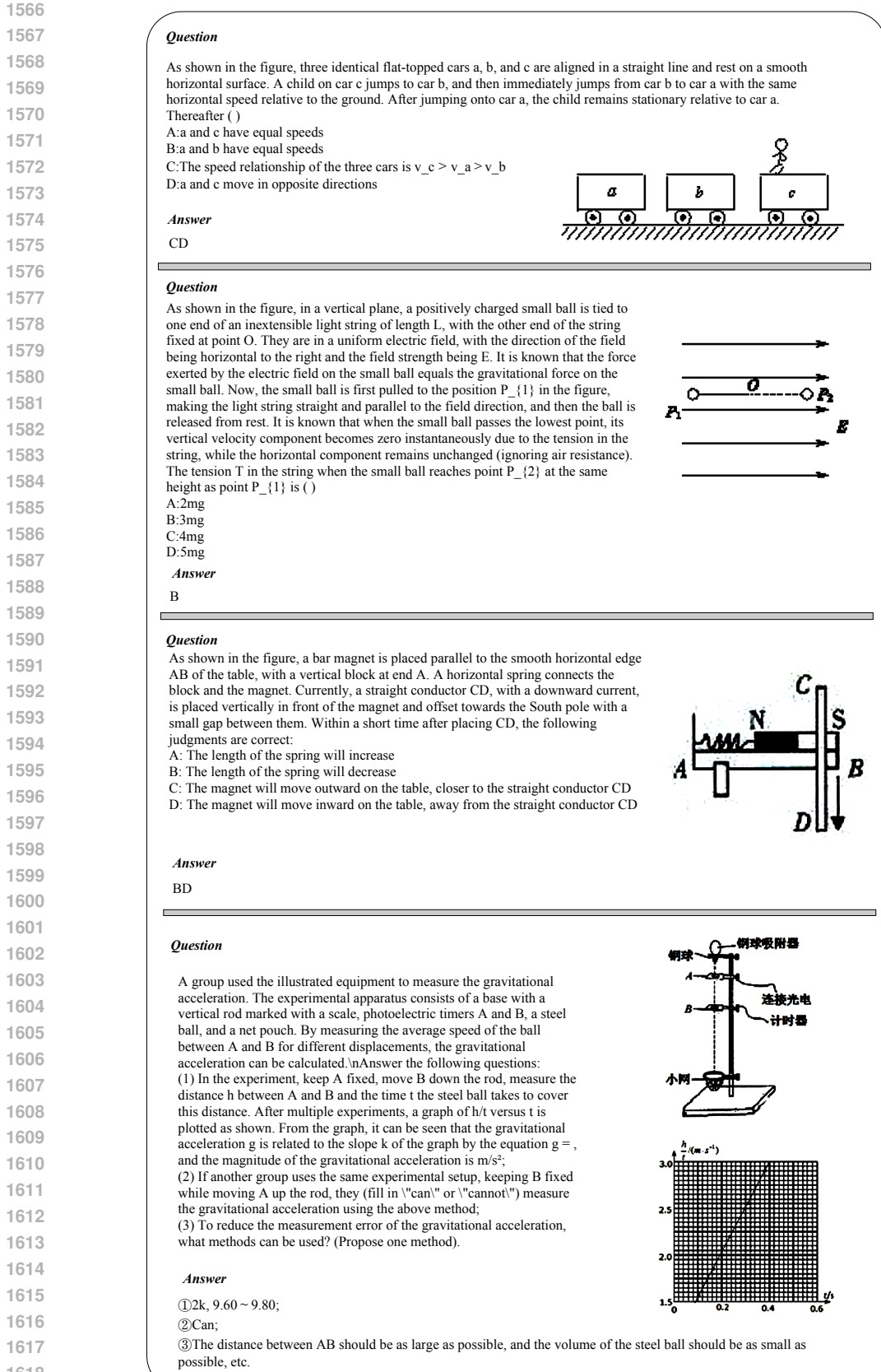

**Question**

As shown in the figure, three identical flat-topped cars a, b, and c are aligned in a straight line and rest on a smooth horizontal surface. A child on car c jumps to car b, and then immediately jumps from car b to car a with the same horizontal speed relative to the ground. After jumping onto car a, the child remains stationary relative to car a. Thereafter ( )
A:a and c have equal speeds
B:a and b have equal speeds
C:The speed relationship of the three cars is $v\_c > v\_a > v\_b$
D:a and c move in opposite directions

**Answer**

CD

**Question**

As shown in the figure, in a vertical plane, a positively charged small ball is tied to one end of an inextensible light string of length L, with the other end of the string fixed at point O. They are in a uniform electric field, with the direction of the field being horizontal to the right and the field strength being E. It is known that the force exerted by the electric field on the small ball equals the gravitational force on the small ball. Now, the small ball is first pulled to the position $P\_{1}$ in the figure, making the light string straight and parallel to the field direction, and then the ball is released from rest. It is known that when the small ball passes the lowest point, its vertical velocity component becomes zero instantaneously due to the tension in the string, while the horizontal component remains unchanged (ignoring air resistance). The tension T in the string when the small ball reaches point $P\_{2}$ at the same height as point $P\_{1}$ is ( )
A:2mg
B:3mg
C:4mg
D:5mg

**Answer**

B

**Question**

As shown in the figure, a bar magnet is placed parallel to the smooth horizontal edge AB of the table, with a vertical block at end A. A horizontal spring connects the block and the magnet. Currently, a straight conductor CD, with a downward current, is placed vertically in front of the magnet and offset towards the South pole with a small gap between them. Within a short time after placing CD, the following judgments are correct:
A: The length of the spring will increase
B: The length of the spring will decrease
C: The magnet will move outward on the table, closer to the straight conductor CD
D: The magnet will move inward on the table, away from the straight conductor CD

**Answer**

BD

**Question**

A group used the illustrated equipment to measure the gravitational acceleration. The experimental apparatus consists of a base with a vertical rod marked with a scale, photoelectric timers A and B, a steel ball, and a net pouch. By measuring the average speed of the ball between A and B for different displacements, the gravitational acceleration can be calculated.\nAnswer the following questions:
(1) In the experiment, keep A fixed, move B down the rod, measure the distance h between A and B and the time t the steel ball takes to cover this distance. After multiple experiments, a graph of h/t versus t is plotted as shown. From the graph, it can be seen that the gravitational acceleration g is related to the slope k of the graph by the equation g = , and the magnitude of the gravitational acceleration is m/s²;
(2) If another group uses the same experimental setup, keeping B fixed while moving A up the rod, they (fill in \"can\" or \"cannot\") measure the gravitational acceleration using the above method;
(3) To reduce the measurement error of the gravitational acceleration, what methods can be used? (Propose one method).

**Answer**

①2k, 9.60 ~ 9.80;

②Can;

③The distance between AB should be as large as possible, and the volume of the steel ball should be as small as possible, etc.

Figure 11: Cases of *mechanics* in physics part of VisScience.

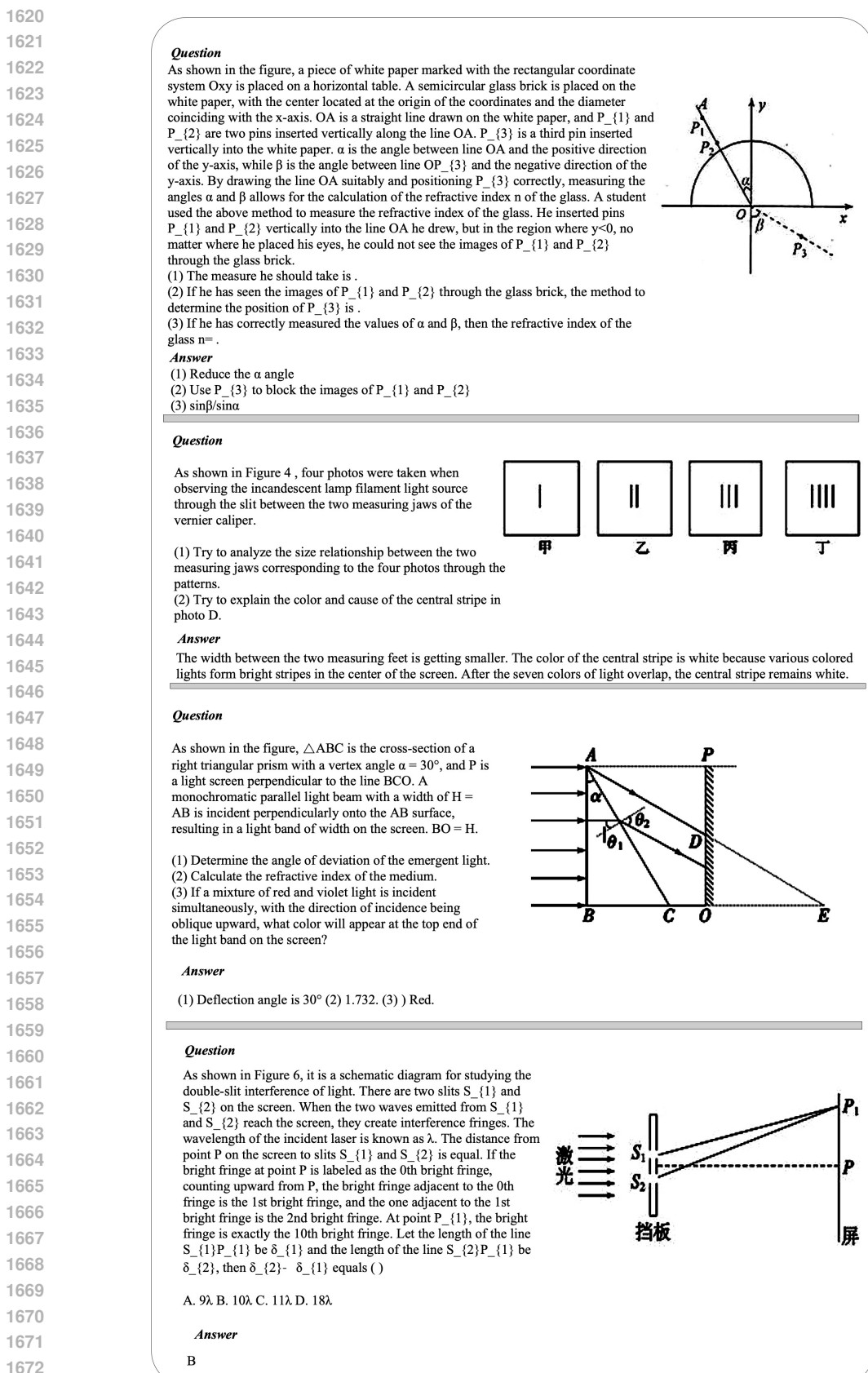

*Question*

As shown in the figure, a piece of white paper marked with the rectangular coordinate system Oxy is placed on a horizontal table. A semicircular glass brick is placed on the white paper, with the center located at the origin of the coordinates and the diameter coinciding with the x-axis. OA is a straight line drawn on the white paper, and $P_{1}$ and $P_{2}$ are two pins inserted vertically along the line OA. $P_{3}$ is a third pin inserted vertically into the white paper. α is the angle between line OA and the positive direction of the y-axis, while β is the angle between line $OP_{3}$ and the negative direction of the y-axis. By drawing the line OA suitably and positioning $P_{3}$ correctly, measuring the angles α and β allows for the calculation of the refractive index n of the glass. A student used the above method to measure the refractive index of the glass. He inserted pins $P_{1}$ and $P_{2}$ vertically into the line OA he drew, but in the region where y<0, no matter where he placed his eyes, he could not see the images of $P_{1}$ and $P_{2}$ through the glass brick.

(1) The measure he should take is .
(2) If he has seen the images of $P_{1}$ and $P_{2}$ through the glass brick, the method to determine the position of $P_{3}$ is .
(3) If he has correctly measured the values of α and β, then the refractive index of the glass n= .

*Answer*

(1) Reduce the α angle
(2) Use $P_{3}$ to block the images of $P_{1}$ and $P_{2}$
(3) sinβ/sinα

*Question*

As shown in Figure 4 , four photos were taken when observing the incandescent lamp filament light source through the slit between the two measuring jaws of the vernier caliper.

甲   乙   丙   丁

(1) Try to analyze the size relationship between the two measuring jaws corresponding to the four photos through the patterns.
(2) Try to explain the color and cause of the central stripe in photo D.

*Answer*

The width between the two measuring feet is getting smaller. The color of the central stripe is white because various colored lights form bright stripes in the center of the screen. After the seven colors of light overlap, the central stripe remains white.

*Question*

As shown in the figure, △ABC is the cross-section of a right triangular prism with a vertex angle α = 30°, and P is a light screen perpendicular to the line BCO. A monochromatic parallel light beam with a width of H = AB is incident perpendicularly onto the AB surface, resulting in a light band of width on the screen. BO = H.

(1) Determine the angle of deviation of the emergent light.
(2) Calculate the refractive index of the medium.
(3) If a mixture of red and violet light is incident simultaneously, with the direction of incidence being oblique upward, what color will appear at the top end of the light band on the screen?

*Answer*

(1) Deflection angle is 30° (2) 1.732. (3) ) Red.

*Question*

As shown in Figure 6, it is a schematic diagram for studying the double-slit interference of light. There are two slits $S_{1}$ and $S_{2}$ on the screen. When the two waves emitted from $S_{1}$ and $S_{2}$ reach the screen, they create interference fringes. The wavelength of the incident laser is known as λ. The distance from point P on the screen to slits $S_{1}$ and $S_{2}$ is equal. If the bright fringe at point P is labeled as the 0th bright fringe, counting upward from P, the bright fringe adjacent to the 0th fringe is the 1st bright fringe, and the one adjacent to the 1st bright fringe is the 2nd bright fringe. At point $P_{1}$, the bright fringe is exactly the 10th bright fringe. Let the length of the line $S_{1}P_{1}$ be $\delta_{1}$ and the length of the line $S_{2}P_{1}$ be $\delta_{2}$, then $\delta_{2}$- $\delta_{1}$ equals ( )

A. 9λ B. 10λ C. 11λ D. 18λ

*Answer*

B

Figure 12: Cases of *optics* in physics part of VisScience.

**Question**

According to the hydrogen atom energy level diagram (as shown):

A: The energy of the photons emitted when the hydrogen atom transitions is continuous

B: The smaller the orbital radius of the electron, the greater the energy of the hydrogen atom

C: The hydrogen atom in its ground state is the most stable

D: To excite the hydrogen atom in the ground state, 11eV photons can be used

$n=4$ ———— $-0.85$ eV
$n=3$ ———— $-1.51$ eV
$n=2$ ———— $-3.40$ eV
$n=1$ ———— $-13.6$ eV

**Answer**

C

**Question**

As shown in the figure, an electroscope is connected to a metal plate. When the metal plate is illuminated with a green light at point A, the electroscope's needle deflects at a certain angle. After turning off the light, the needle maintains its position at a certain angle.

(1) When a small metal sphere carrying a slight negative charge touches the metal plate, the electroscope's needle deflection angle will ______ (fill in "increase", "decrease", or "remain unchanged").
(2) If the electroscope needle is reset to zero and the metal plate is illuminated with yellow light from a sodium lamp of the same intensity, the electroscope needle does not deflect. If a more intense infrared lamp is used to illuminate the metal plate, the observation will show ______ (fill in "deflection" or "no deflection") of the electroscope needle. If a very faint purple light is used instead, the observation will show ______ (fill in "deflection" or "no deflection") of the electroscope needle.

验电器 金属板 绿色灯

**Answer**

(1) Reduce (2) No Yes

**Question**

As shown in the figure, the circuit is used to study the photoelectric effect. The cathode K and anode A are two electrodes sealed in a vacuum glass tube. When K is illuminated, it can emit photoelectrons. The anode A absorbs the photoelectrons emitted by the cathode K, forming a photocurrent in the circuit. When the cathode K is illuminated with monochromatic light a, the pointer of the ammeter deflects; when the photoelectric tube cathode K is illuminated with monochromatic light b, the pointer of the ammeter does not deflect. Which of the following statements is correct? (Fill in the correct answer letter)

A. The wavelength of light a must be less than the wavelength of light b
B. Increasing the intensity of light a alone may increase the current through the ammeter
C. Increasing the intensity of light a alone can increase the maximum initial kinetic energy of the ejected electrons
D. The work function of the cathode material is related to the frequency of the incident light

E. When the cathode K is illuminated with monochromatic light a, if the polarity of the power supply is reversed, the reading of the ammeter may drop to zero

单 色 W K A

**Answer**

ABE

**Question**

As shown in the figure, Rutherford's apparatus for bombarding gold foil with alpha particles is depicted. The correct description of the experiment is ()

A: The experiment of bombarding gold foil with α particles needs to be conducted under vacuum conditions
B: The α particle scattering experiment revealed the complex structure of the atomic nucleus
C: The experimental results show that the vast majority of α particles passed through the gold foil without scattering D: The α particles carried information about the internal structure of the atom after passing through the gold atoms

荧光屏 α粒子源 α金箔 α粒子束 α

**Answer**

ACD

Figure 13: Cases of *modern physics* in physics part of VisScience.

*Question*

In the four images shown in , the one that represents an object in uniform accelerated linear motion is ( )

*Answer*

C

*Question*

As shown in the figure is the v-t graph of an elevator moving upward, the maximum speed of the object's motion equals m/s, and the height of uniform ascent equals ( ).

*Answer*

From the graph, it can be seen that the elevator first accelerates uniformly upwards, then moves upwards at a constant speed, and finally decelerates uniformly upwards, with a maximum speed of 6m/s.
The height of the constant speed upward motion h = 6×4m = 24m.
Therefore, the answer is: 6, 24m.

*Question*

To ensure orderly and safe highway traffic, many traffic signs are placed along the roadside.
As shown in the picture, image A is a speed limit sign indicating the maximum allowed speed is 80 km/h; image B is a route indicator sign, showing the distance of 100 km to Hangzhou. The physical meanings of the two data mentioned are ( )
A: 80 km/h is average speed
B: 80 km/h is instantaneous speed
C: 100 km is displacement
D: 100 km is distance

*Answer*

Solution: The maximum allowable speed represents the speed at a certain position, which is the instantaneous speed, so 80 km/h refers to the instantaneous speed;
There are still 100 km to Hangzhou, and 100 km is the length of the trajectory, which is the distance. Therefore, BD is correct, and AC is incorrect.
The answer is: BD

*Question*

The value of the area enclosed by the velocity-time graph of a particle's straight-line motion and the coordinate axes is equal to the displacement of the particle during that period. This holds true not only for uniform linear motion and uniformly accelerated linear motion but also for any variable motion. The figure shows the velocity-time graph of a certain particle during the 0-4s interval of straight-line motion. According to the graph, the displacement of the particle during this period is closest to ( )
A:15m
B:13m
C:11m
D:7m

*Answer*

Solution: The area enclosed by the graph line and the time axis is approximately 15 squares, so the displacement is:
x=15×1×1m=15m.
Thus, the answer is: A.

Figure 14: Cases of *mechanical motion* in physics part of VisScience.

**Question**

As shown in the figure, when an electron beam passes through the two magnetic poles in the direction indicated by the arrow in the figure, its deflection direction is ( )
A: Upward
B: Downward
C: Toward the N pole
D: Toward the S pole

**Answer**

B

**Question**

As shown in the figure, a positively charged particle enters a uniform magnetic field horizontally to the right, neglecting gravity. Regarding the motion of this charged particle after it enters the magnetic field, the following judgments are correct: ( )
A: The particle deflects upwards
B: The particle deflects downwards
C: The particle does not deflect
D: The particle quickly stops moving

**Answer**

A

**Question**

As shown in the figure, there is a uniform magnetic field with a magnetic induction intensity B=1T. A wire with a length of L=0.1 m is placed perpendicularly to the direction of the magnetic field. When a horizontal current of I=0.5A flows to the right through the wire,
(1) Determine whether the direction of the Ampere force on the wire is vertically upward or downward;
(2) Calculate the magnitude of the Ampere force F on the wire.

**Answer**

Solution: (1)According to the left-hand rule, it can be determined that the direction of the Ampere force on the wire is vertically upward.
(2)Ampere force on the wire: F=BIL=1×0.5×0.1N=0.05N

**Question**

As shown in Figure A, a uniform magnetic field B=0.5T exists in space, directed vertically downward. MN and PQ are rough, parallel, long straight rails in the same horizontal plane, with a distance L=0.2m between them; R is a resistor connected at one end of the rails. The ab is a conducting rod with a mass of m=0.1kg bridged across the rails. From time zero, a small motor exerts a horizontal leftward traction force F on rod ab, causing it to start moving from rest and accelerating along the rails. During this process, the rod always remains perpendicular to and in good contact with the rails. Figure B shows the v-t graph of the rod, where segment OA is a straight line, and AC is a curve. The small motor reaches its rated power P=4.5W at the end of 12s, after which it maintains constant power. The resistance of other parts except R is negligible, and g=10m/s^{2}.
(1) Find the acceleration of the conductor rod ab during the 0-12s interval;
(2) Find the coefficient of kinetic friction between rod ab and the rails and the value of the resistor R;
(3) Draw the graph of the traction force's power versus time (P-t) on the answer sheet.

**Answer**

0.75m/s^{2} Dynamic friction coefficient is 0.2 Resistance value is 0.4 ohms\nThe graph is a parabola passing through the origin from 0 to 12 seconds (but the vertex of the parabola is not the origin). After 12 seconds, the power is constant.

Figure 15: Cases of *electromagnetism* in physics part of VisScience.

**Question**

M and N are two points in the direction of wave propagation in the medium, with a distance s =\"1.5\" m, and their vibration images are shown in the figure. The possible values for the wave speed are( ):

A:15 m/s

B:7.5 m/s

C:5 m/s

D:3 m/s

**Answer**

ACD

**Question**

As shown in the figure, the image represents a mechanical wave propagating along the x-axis at a certain moment. From the figure, it can be seen that the amplitude A and wavelength $\lambda$ of this wave are respectively ( )

A:A=2cm,$\lambda$=4m

B:A=2cm,$\lambda$=8m

C:A=4cm,$\lambda$=4m

D:A=4cm,$\lambda$=8m

**Answer**

According to the definition of amplitude: Amplitude A is the maximum distance the vibrating object moves away from the equilibrium position, read the maximum value of y as A=2cm. Based on the fact that wavelength equals the distance between two adjacent wave crests or troughs, read the wavelength $\lambda$=4m.

So choose B.

**Question**

Starlight Middle School has a 150m×70m playground. Broadcast speakers S_1 and S_2 are set up at both ends of the playground (as shown in the picture). Teacher Wu walked around the playground to test the sound and found that at points A, B, C, and D (which are the midpoints of each side), the sound was indistinct at two places. These two places are ( ).

A: B and D

B: A and B

C: C and D

D: A and C

**Answer**

Solution: From the diagram, it is evident that B and D are equidistant from both sound sources. The sound reaches them simultaneously without interference, resulting in a louder perception. In contrast, A and C are at noticeably different distances from the two sound sources, causing interference between the sounds, making them harder to hear clearly. Therefore, the answer is D.

**Question**

There are two vibrating sources A and B on the water surface, vibrating in exactly the same way. On the perpendicular bisector of the line connecting A and B, there are three points a, b, and c. At a certain moment, point a is the convergence point of the crests of the two waves, and point c is the nearest convergence point of the troughs of the two waves from point a. Point b is located on the line connecting points a and c, as shown in the figure. Which of the following statements is correct? ( )

A: Both a and c are points of constructive interference, and b is a point of destructive interference.

B: a is a point of constructive interference, and c is a point of destructive interference.

C: Both a and c are currently points of constructive interference, but after some time, they will become points of destructive interference.

D: a, b, and c are all points of constructive interference.

**Answer**

Solution: From the question, A and B are two sources of vibration with identical conditions. Points a, b, and c lie on the perpendicular bisector of the line AB, meaning the distances from a, b, and c to the two wave sources are equal, resulting in a path difference of zero. The vibrations at these three points all reinforce each other.\nTherefore, the answer is: D

Figure 16: Cases of *vibration and waves* in physics part of VisScience.

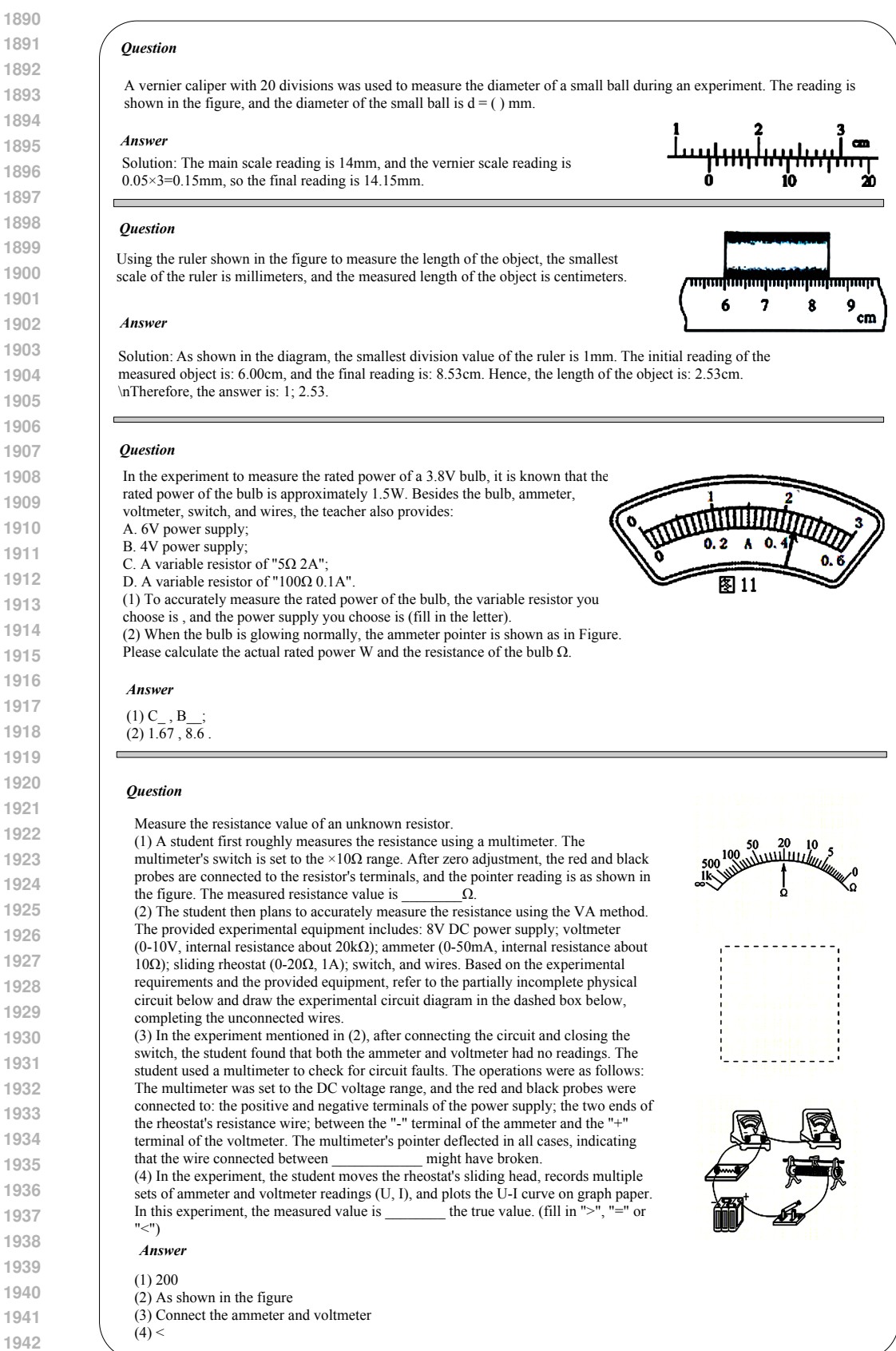

*Question*

A vernier caliper with 20 divisions was used to measure the diameter of a small ball during an experiment. The reading is shown in the figure, and the diameter of the small ball is d = ( ) mm.

*Answer*

Solution: The main scale reading is 14mm, and the vernier scale reading is 0.05×3=0.15mm, so the final reading is 14.15mm.

*Question*

Using the ruler shown in the figure to measure the length of the object, the smallest scale of the ruler is millimeters, and the measured length of the object is centimeters.

*Answer*

Solution: As shown in the diagram, the smallest division value of the ruler is 1mm. The initial reading of the measured object is: 6.00cm, and the final reading is: 8.53cm. Hence, the length of the object is: 2.53cm. \nTherefore, the answer is: 1; 2.53.

*Question*

In the experiment to measure the rated power of a 3.8V bulb, it is known that the rated power of the bulb is approximately 1.5W. Besides the bulb, ammeter, voltmeter, switch, and wires, the teacher also provides:
A. 6V power supply;
B. 4V power supply;
C. A variable resistor of "5Ω 2A";
D. A variable resistor of "100Ω 0.1A".
(1) To accurately measure the rated power of the bulb, the variable resistor you choose is , and the power supply you choose is (fill in the letter).
(2) When the bulb is glowing normally, the ammeter pointer is shown as in Figure. Please calculate the actual rated power W and the resistance of the bulb Ω.

*Answer*

(1) C_ , B__;
(2) 1.67 , 8.6 .

*Question*

Measure the resistance value of an unknown resistor.
(1) A student first roughly measures the resistance using a multimeter. The multimeter's switch is set to the ×10Ω range. After zero adjustment, the red and black probes are connected to the resistor's terminals, and the pointer reading is as shown in the figure. The measured resistance value is ________Ω.
(2) The student then plans to accurately measure the resistance using the VA method. The provided experimental equipment includes: 8V DC power supply; voltmeter (0-10V, internal resistance about 20kΩ); ammeter (0-50mA, internal resistance about 10Ω); sliding rheostat (0-20Ω, 1A); switch, and wires. Based on the experimental requirements and the provided equipment, refer to the partially incomplete physical circuit below and draw the experimental circuit diagram in the dashed box below, completing the unconnected wires.
(3) In the experiment mentioned in (2), after connecting the circuit and closing the switch, the student found that both the ammeter and voltmeter had no readings. The student used a multimeter to check for circuit faults. The operations were as follows: The multimeter was set to the DC voltage range, and the red and black probes were connected to: the positive and negative terminals of the power supply; the two ends of the rheostat's resistance wire; between the "-" terminal of the ammeter and the "+" terminal of the voltmeter. The multimeter's pointer deflected in all cases, indicating that the wire connected between ____________ might have broken.
(4) In the experiment, the student moves the rheostat's sliding head, records multiple sets of ammeter and voltmeter readings (U, I), and plots the U-I curve on graph paper. In this experiment, the measured value is ________ the true value. (fill in ">", "=" or "<")

*Answer*

(1) 200
(2) As shown in the figure
(3) Connect the ammeter and voltmeter
(4) <

Figure 17: Cases of *comprehensive experiments and methods* in physics part of VisScience.

**Question**

Using two identical electric heaters to heat substances A and water, both with a mass of 2kg, the relationship between their temperature and time is shown in the diagram. Based on this, determine the amount of heat absorbed by substance A in 10 minutes ( ).
A:5.04×10^5 J
B:4.2×10^5 J
C:2.52×10^5 J
D:Insufficient conditions, cannot be calculated

**Answer**

C

**Question**

In the experiment of exploring the "boiling law of water," Xiaoming recorded the temperature every 30s after the water temperature rose to 90°C, and then plotted the temperature versus time graph, as shown in the figure. If the thermometer used by Xiaoming is accurate, it can be concluded from the graph that the boiling temperature of water is °C, which indicates that the atmospheric pressure at that time is 1 standard atmosphere (choose "greater than," "less than," or "equal to").

**Answer**

From the figure, it can be seen that the boiling point of water is 98°C. The boiling point of water under standard atmospheric pressure is 100°C. The boiling point of a liquid decreases as the pressure decreases, therefore the atmospheric pressure at that time is lower than 1 standard atmospheric pressure.
So the answer is: 98; lower.

**Question**

For the experiment shown in the figure, the following statements are correct ():
A: In the first diagram, the copper tube's temperature increases when rubbed with a rubber band. This shows that the internal energy of the copper tube increases through work done on it.
B: In the second diagram, the steam pushes out the rubber stopper, indicating that the steam does work on the rubber stopper, causing the internal energy of the steam to increase.
C: In the third diagram, ink diffuses significantly more in hot water than in cold water, demonstrating that the intensity of molecular random motion is related to temperature.
D: In the fourth diagram, after removing the glass plate, the gases in the two bottles gradually mix evenly, which is a diffusion phenomenon.

**Answer**

ACD

**Question**

Some students in a group conducted an experiment to "compare the heat absorption capacity of different substances" using the setup shown in the figure.
(1) When designing the experimental plan, the following controlled variables need to be determined. Which one do you think is unnecessary?
A. Use the exact same heating method
B. Use the same amount of alcohol in the alcohol lamps
C. Use the same mass of water and another liquid
D. Use the same container to hold water and another liquid
(2) When heated to a certain point, the water starts boiling. The temperature at this moment is shown in Figure C. Therefore, the boiling point of water is °C, which indicates that the atmospheric pressure during the experiment is (choose "greater than", "less than", or "equal to") one standard atmosphere.
(3) The other liquid does not boil at the corresponding moment, but the reading of the thermometer is much higher than the water temperature. Based on this phenomenon, the preliminary conclusion of this experiment is: the heat absorption capacity of different substances is (choose "the same" or "different").

**Answer**

(1) B (2) 97, less than (3) different

Figure 18: Cases of *thermodynamics* in physics part of VisScience.

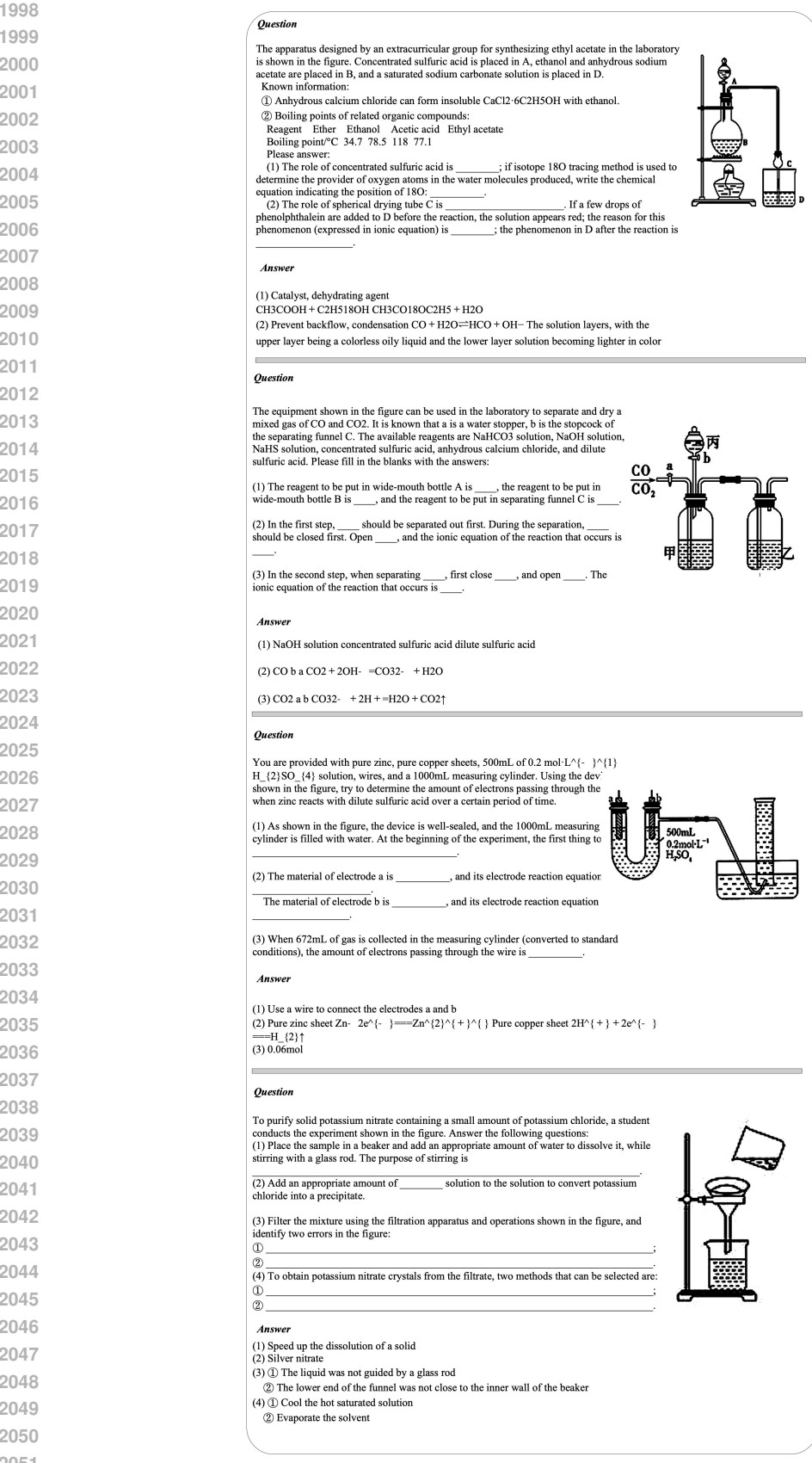

Figure 19: Cases of *chemical experiment* in chemistry part of VisScience.

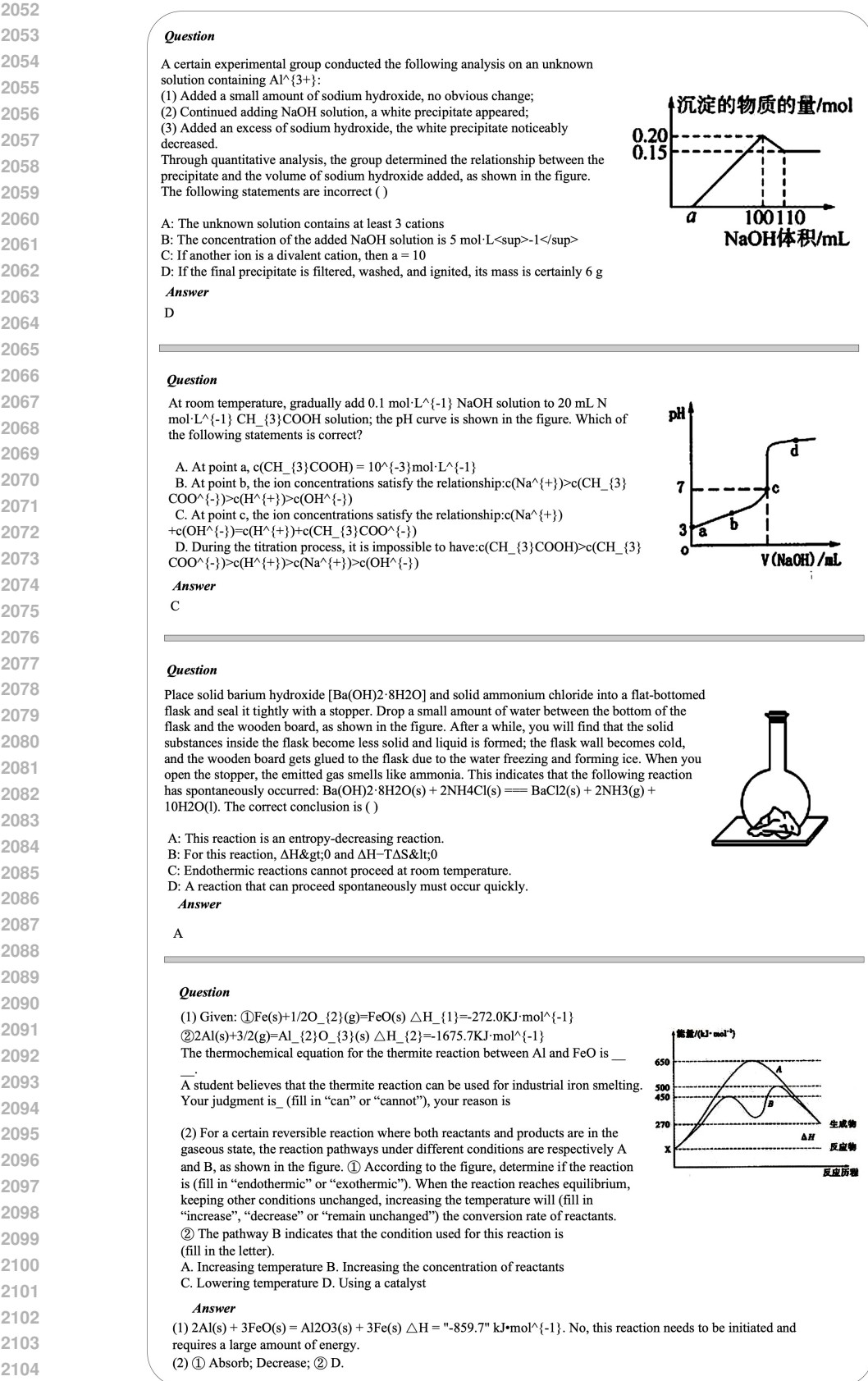

*Question*

A certain experimental group conducted the following analysis on an unknown solution containing $Al^{3+}$:
(1) Added a small amount of sodium hydroxide, no obvious change;
(2) Continued adding NaOH solution, a white precipitate appeared;
(3) Added an excess of sodium hydroxide, the white precipitate noticeably decreased.
Through quantitative analysis, the group determined the relationship between the precipitate and the volume of sodium hydroxide added, as shown in the figure. The following statements are incorrect ( )

A: The unknown solution contains at least 3 cations
B: The concentration of the added NaOH solution is 5 mol·L-1
C: If another ion is a divalent cation, then a = 10
D: If the final precipitate is filtered, washed, and ignited, its mass is certainly 6 g

*Answer*

D

*Question*

At room temperature, gradually add 0.1 mol·L$^{-1}$ NaOH solution to 20 mL N mol·L$^{-1}$ $CH_3COOH$ solution; the pH curve is shown in the figure. Which of the following statements is correct?

A. At point a, c($CH_3COOH$) = $10^{-3}$mol·L$^{-1}$
B. At point b, the ion concentrations satisfy the relationship:c($Na^+$)>c($CH_3COO^-$)>c($H^+$)>c($OH^-$)
C. At point c, the ion concentrations satisfy the relationship:c($Na^+$) +c($OH^-$)=c($H^+$)+c($CH_3COO^-$)
D. During the titration process, it is impossible to have:c($CH_3COOH$)>c($CH_3COO^-$)>c($H^+$)>c($Na^+$)>c($OH^-$)

*Answer*

C

*Question*

Place solid barium hydroxide [Ba(OH)2·8H2O] and solid ammonium chloride into a flat-bottomed flask and seal it tightly with a stopper. Drop a small amount of water between the bottom of the flask and the wooden board, as shown in the figure. After a while, you will find that the solid substances inside the flask become less solid and liquid is formed; the flask wall becomes cold, and the wooden board gets glued to the flask due to the water freezing and forming ice. When you open the stopper, the emitted gas smells like ammonia. This indicates that the following reaction has spontaneously occurred: Ba(OH)2·8H2O(s) + 2NH4Cl(s) === BaCl2(s) + 2NH3(g) + 10H2O(l). The correct conclusion is ( )

A: This reaction is an entropy-decreasing reaction.
B: For this reaction, ΔH>0 and ΔH−TΔS<0
C: Endothermic reactions cannot proceed at room temperature.
D: A reaction that can proceed spontaneously must occur quickly.

*Answer*

A

*Question*

(1) Given: ①Fe(s)+1/2O$_2$(g)=FeO(s) △H$_1$=-272.0KJ·mol$^{-1}$
②2Al(s)+3/2(g)=Al$_2$O$_3$(s) △H$_2$=-1675.7KJ·mol$^{-1}$
The thermochemical equation for the thermite reaction between Al and FeO is ___.

A student believes that the thermite reaction can be used for industrial iron smelting. Your judgment is_ (fill in "can" or "cannot"), your reason is

(2) For a certain reversible reaction where both reactants and products are in the gaseous state, the reaction pathways under different conditions are respectively A and B, as shown in the figure. ① According to the figure, determine if the reaction is (fill in "endothermic" or "exothermic"). When the reaction reaches equilibrium, keeping other conditions unchanged, increasing the temperature will (fill in "increase", "decrease" or "remain unchanged") the conversion rate of reactants.
② The pathway B indicates that the condition used for this reaction is (fill in the letter).
A. Increasing temperature B. Increasing the concentration of reactants
C. Lowering temperature D. Using a catalyst

*Answer*

(1) 2Al(s) + 3FeO(s) = Al2O3(s) + 3Fe(s) △H = "-859.7" kJ•mol$^{-1}$. No, this reaction needs to be initiated and requires a large amount of energy.
(2) ① Absorb; Decrease; ② D.

Figure 20: Cases of *chemical reaction* in chemistry part of VisScience.

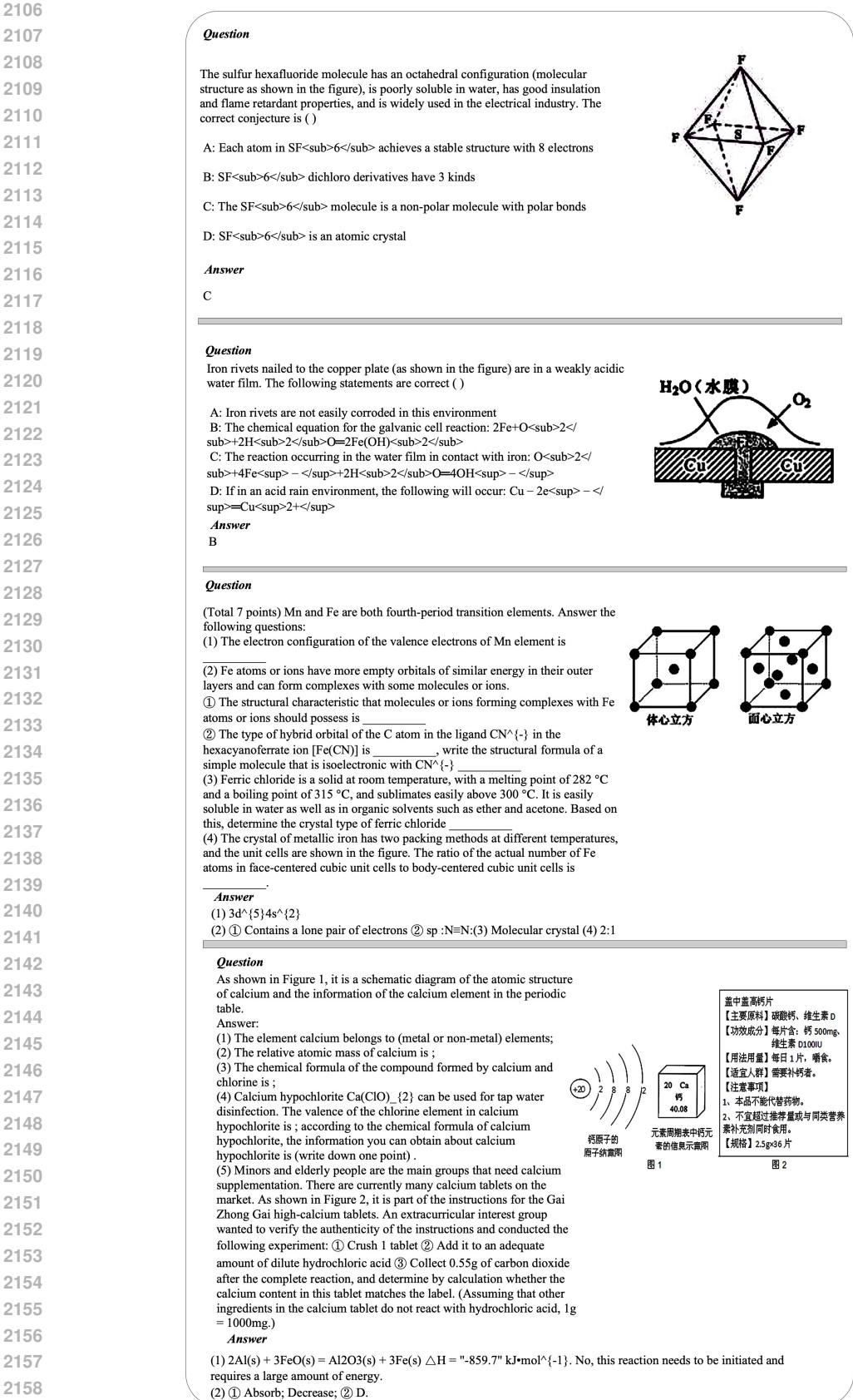

**Question**

The sulfur hexafluoride molecule has an octahedral configuration (molecular structure as shown in the figure), is poorly soluble in water, has good insulation and flame retardant properties, and is widely used in the electrical industry. The correct conjecture is ( )

A: Each atom in $SF_6$ achieves a stable structure with 8 electrons

B: $SF_6$ dichloro derivatives have 3 kinds

C: The $SF_6$ molecule is a non-polar molecule with polar bonds

D: $SF_6$ is an atomic crystal

**Answer**

C

**Question**

Iron rivets nailed to the copper plate (as shown in the figure) are in a weakly acidic water film. The following statements are correct ( )

A: Iron rivets are not easily corroded in this environment
B: The chemical equation for the galvanic cell reaction: $2Fe + O_2 + 2H_2O = 2Fe(OH)_2$
C: The reaction occurring in the water film in contact with iron: $O_2 + 4Fe^- + 2H_2O = 4OH^-$
D: If in an acid rain environment, the following will occur: $Cu - 2e^- = Cu^{2+}$

**Answer**

B

**Question**

(Total 7 points) Mn and Fe are both fourth-period transition elements. Answer the following questions:
(1) The electron configuration of the valence electrons of Mn element is __________
(2) Fe atoms or ions have more empty orbitals of similar energy in their outer layers and can form complexes with some molecules or ions.
① The structural characteristic that molecules or ions forming complexes with Fe atoms or ions should possess is __________
② The type of hybrid orbital of the C atom in the ligand $CN^{-}$ in the hexacyanoferrate ion [Fe(CN)] is __________, write the structural formula of a simple molecule that is isoelectronic with $CN^{-}$ __________
(3) Ferric chloride is a solid at room temperature, with a melting point of 282 °C and a boiling point of 315 °C, and sublimes easily above 300 °C. It is easily soluble in water as well as in organic solvents such as ether and acetone. Based on this, determine the crystal type of ferric chloride __________
(4) The crystal of metallic iron has two packing methods at different temperatures, and the unit cells are shown in the figure. The ratio of the actual number of Fe atoms in face-centered cubic unit cells to body-centered cubic unit cells is __________.

**Answer**

(1) $3d^{5}4s^{2}$
(2) ① Contains a lone pair of electrons ② sp :N≡N:(3) Molecular crystal (4) 2:1

**Question**

As shown in Figure 1, it is a schematic diagram of the atomic structure of calcium and the information of the calcium element in the periodic table.
Answer:
(1) The element calcium belongs to (metal or non-metal) elements;
(2) The relative atomic mass of calcium is ;
(3) The chemical formula of the compound formed by calcium and chlorine is ;
(4) Calcium hypochlorite $Ca(ClO)_{2}$ can be used for tap water disinfection. The valence of the chlorine element in calcium hypochlorite is ; according to the chemical formula of calcium hypochlorite, the information you can obtain about calcium hypochlorite is (write down one point) .
(5) Minors and elderly people are the main groups that need calcium supplementation. There are currently many calcium tablets on the market. As shown in Figure 2, it is part of the instructions for the Gai Zhong Gai high-calcium tablets. An extracurricular interest group wanted to verify the authenticity of the instructions and conducted the following experiment: ① Crush 1 tablet ② Add it to an adequate amount of dilute hydrochloric acid ③ Collect 0.55g of carbon dioxide after the complete reaction, and determine by calculation whether the calcium content in this tablet matches the label. (Assuming that other ingredients in the calcium tablet do not react with hydrochloric acid, 1g = 1000mg.)

**Answer**

(1) $2Al(s) + 3FeO(s) = Al2O3(s) + 3Fe(s)$ △H = "-859.7" kJ•mol^{-1}. No, this reaction needs to be initiated and requires a large amount of energy.
(2) ① Absorb; Decrease; ② D.

Figure 21: Cases of *inorganic chemistry* in chemistry part of VisScience.

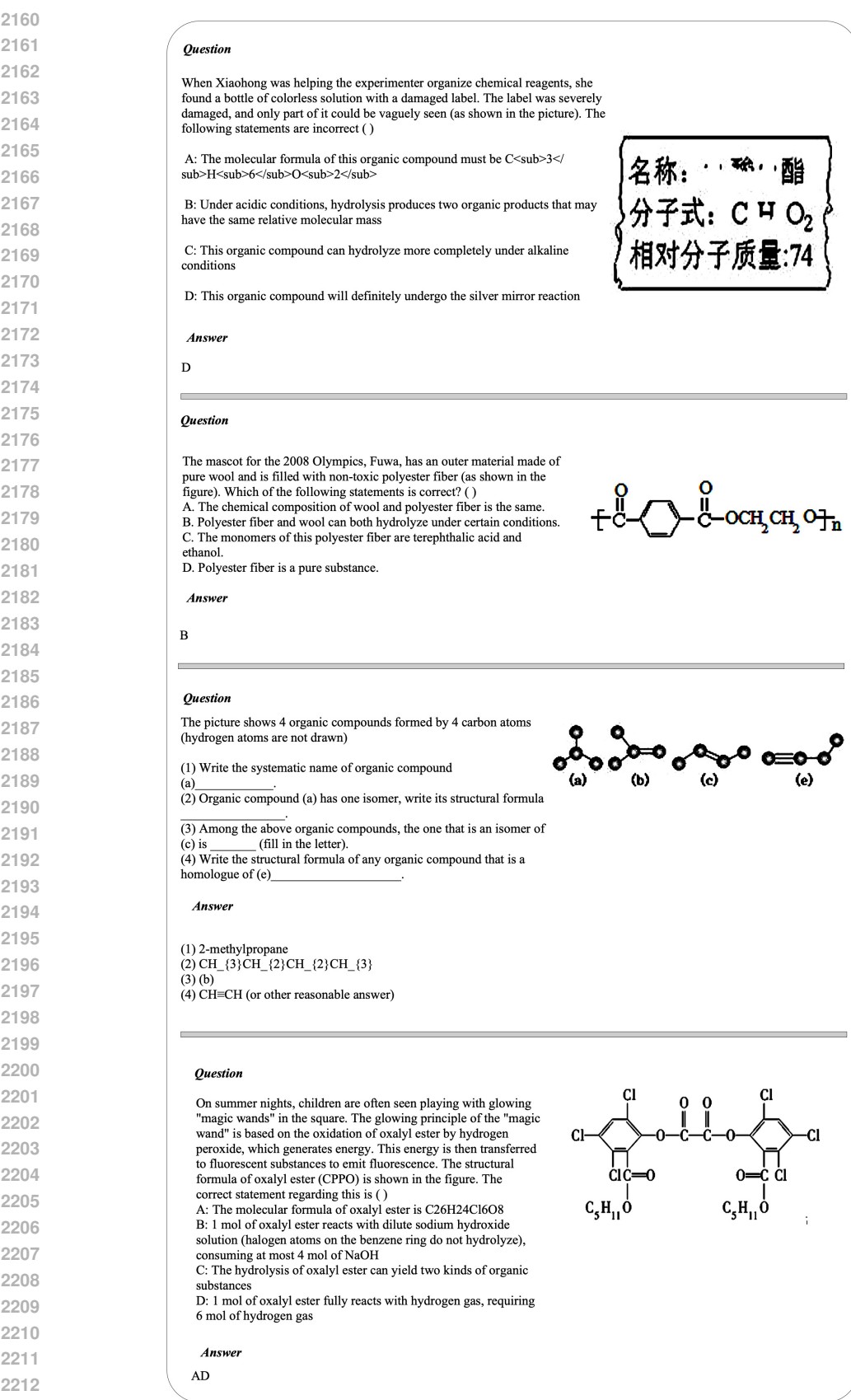

**Question**

When Xiaohong was helping the experimenter organize chemical reagents, she found a bottle of colorless solution with a damaged label. The label was severely damaged, and only part of it could be vaguely seen (as shown in the picture). The following statements are incorrect ( )

A: The molecular formula of this organic compound must be $C_3H_6O_2$

B: Under acidic conditions, hydrolysis produces two organic products that may have the same relative molecular mass

C: This organic compound can hydrolyze more completely under alkaline conditions

D: This organic compound will definitely undergo the silver mirror reaction

**Answer**

D

**Question**

The mascot for the 2008 Olympics, Fuwa, has an outer material made of pure wool and is filled with non-toxic polyester fiber (as shown in the figure). Which of the following statements is correct? ( )
A. The chemical composition of wool and polyester fiber is the same.
B. Polyester fiber and wool can both hydrolyze under certain conditions.
C. The monomers of this polyester fiber are terephthalic acid and ethanol.
D. Polyester fiber is a pure substance.

**Answer**

B

**Question**

The picture shows 4 organic compounds formed by 4 carbon atoms (hydrogen atoms are not drawn)

(1) Write the systematic name of organic compound (a)____________.
(2) Organic compound (a) has one isomer, write its structural formula ________________.
(3) Among the above organic compounds, the one that is an isomer of (c) is ________ (fill in the letter).
(4) Write the structural formula of any organic compound that is a homologue of (e)___________________.

**Answer**

(1) 2-methylpropane
(2) $CH_3CH_2CH_2CH_3$
(3) (b)
(4) CH≡CH (or other reasonable answer)

**Question**

On summer nights, children are often seen playing with glowing "magic wands" in the square. The glowing principle of the "magic wand" is based on the oxidation of oxalyl ester by hydrogen peroxide, which generates energy. This energy is then transferred to fluorescent substances to emit fluorescence. The structural formula of oxalyl ester (CPPO) is shown in the figure. The correct statement regarding this is ( )
A: The molecular formula of oxalyl ester is C26H24Cl6O8
B: 1 mol of oxalyl ester reacts with dilute sodium hydroxide solution (halogen atoms on the benzene ring do not hydrolyze), consuming at most 4 mol of NaOH
C: The hydrolysis of oxalyl ester can yield two kinds of organic substances
D: 1 mol of oxalyl ester fully reacts with hydrogen gas, requiring 6 mol of hydrogen gas

**Answer**

AD

Figure 22: Cases of *organic chemistry* in chemistry part of VisScience.

**Question**

The working principle of the new rechargeable sodium-ion battery represented by Prussian blue$${\rm Fe[Fe(CN)_{6}]}$$ during discharge is shown in the figure. The following statement is incorrect( )

A: During discharge, the positive electrode reaction is$$ {\rm Fe[Fe(CN)_{6}]} +2Na^{+}+2e^{-}=Na_{2}Fe[Fe(CN)_{6}]}$$
B: During charging, $${\rm Mo(}$$molybdenum$${\rm )}$$ foil is connected to the negative terminal of the power supply
C: During charging, $${\rm Na^{+}}$$ moves from the left chamber to the right chamber through the exchange membrane
D: When an amount of $${\rm 0.2mol}$$ electrons passes through the external circuit, the mass change of the negative electrode is $${\rm 2.4g}$$

**Answer**

B

**Question**

The electrolysis of water experiment reveals the composition of water. In the experiment shown in the figure, the substance obtained in test tube 1 is (fill in the chemical formula).

**Answer**

H_{2}

**Question**

At present, scientists have proposed the most economical and ideal cycle system for obtaining hydrogen energy (as shown in the figure). The following statements are incorrect:

A: In this cycle, the reaction water electrified  hydrogen gas + oxygen gas occurs
B: This hydrogen energy cycle system can achieve the conversion of solar energy into electrical energy
C: Fuel cells can convert the energy produced by chemical reactions into electrical energy
D: The urgent issue chemists need to solve is finding a suitable photocatalyst

**Answer**

A

**Question**

Utilizing the apparatus shown in the figure, the electrochemical protection of iron can be simulated.
To reduce iron corrosion:
① If the switch $${\rm K}$$ is positioned at $${\rm N}$$, then $${\rm X}$$ should be ________________, and this electrochemical protection method is ________.
② If the switch $${\rm K}$$ is positioned at $${\rm M}$$, then $${\rm X}$$ should be ________________, and this electrochemical protection method is ________.

**Answer**

1. Cathodic protection method with external current using inert electrodes such as carbon rods or Pt.

2. Cathodic protection method with sacrificial anodes using metals more active than iron, such as Zn.

Figure 23: Cases of *electrochemistry* in chemistry part of VisScience.

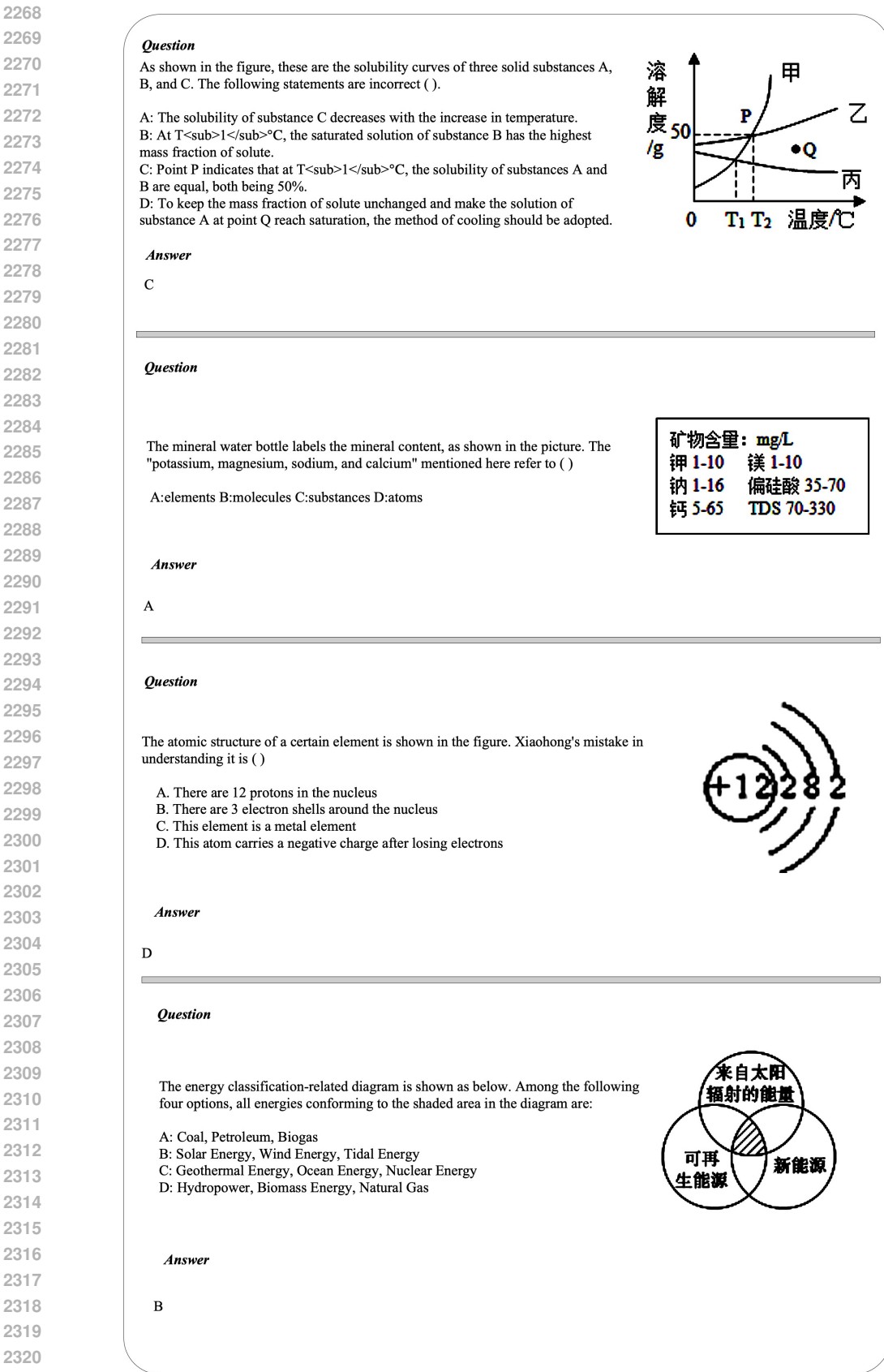

**Question**

As shown in the figure, these are the solubility curves of three solid substances A, B, and C. The following statements are incorrect ( ).

A: The solubility of substance C decreases with the increase in temperature.
B: At $T_1$°C, the saturated solution of substance B has the highest mass fraction of solute.
C: Point P indicates that at $T_1$°C, the solubility of substances A and B are equal, both being 50%.
D: To keep the mass fraction of solute unchanged and make the solution of substance A at point Q reach saturation, the method of cooling should be adopted.

*Answer*

C

**Question**

The mineral water bottle labels the mineral content, as shown in the picture. The "potassium, magnesium, sodium, and calcium" mentioned here refer to ( )

A:elements B:molecules C:substances D:atoms

*Answer*

A

**Question**

The atomic structure of a certain element is shown in the figure. Xiaohong's mistake in understanding it is ( )

A. There are 12 protons in the nucleus
B. There are 3 electron shells around the nucleus
C. This element is a metal element
D. This atom carries a negative charge after losing electrons

*Answer*

D

**Question**

The energy classification-related diagram is shown as below. Among the following four options, all energies conforming to the shaded area in the diagram are:

A: Coal, Petroleum, Biogas
B: Solar Energy, Wind Energy, Tidal Energy
C: Geothermal Energy, Ocean Energy, Nuclear Energy
D: Hydropower, Biomass Energy, Natural Gas

*Answer*

B

Figure 24: Cases of *substance composition* in chemistry part of VisScience.

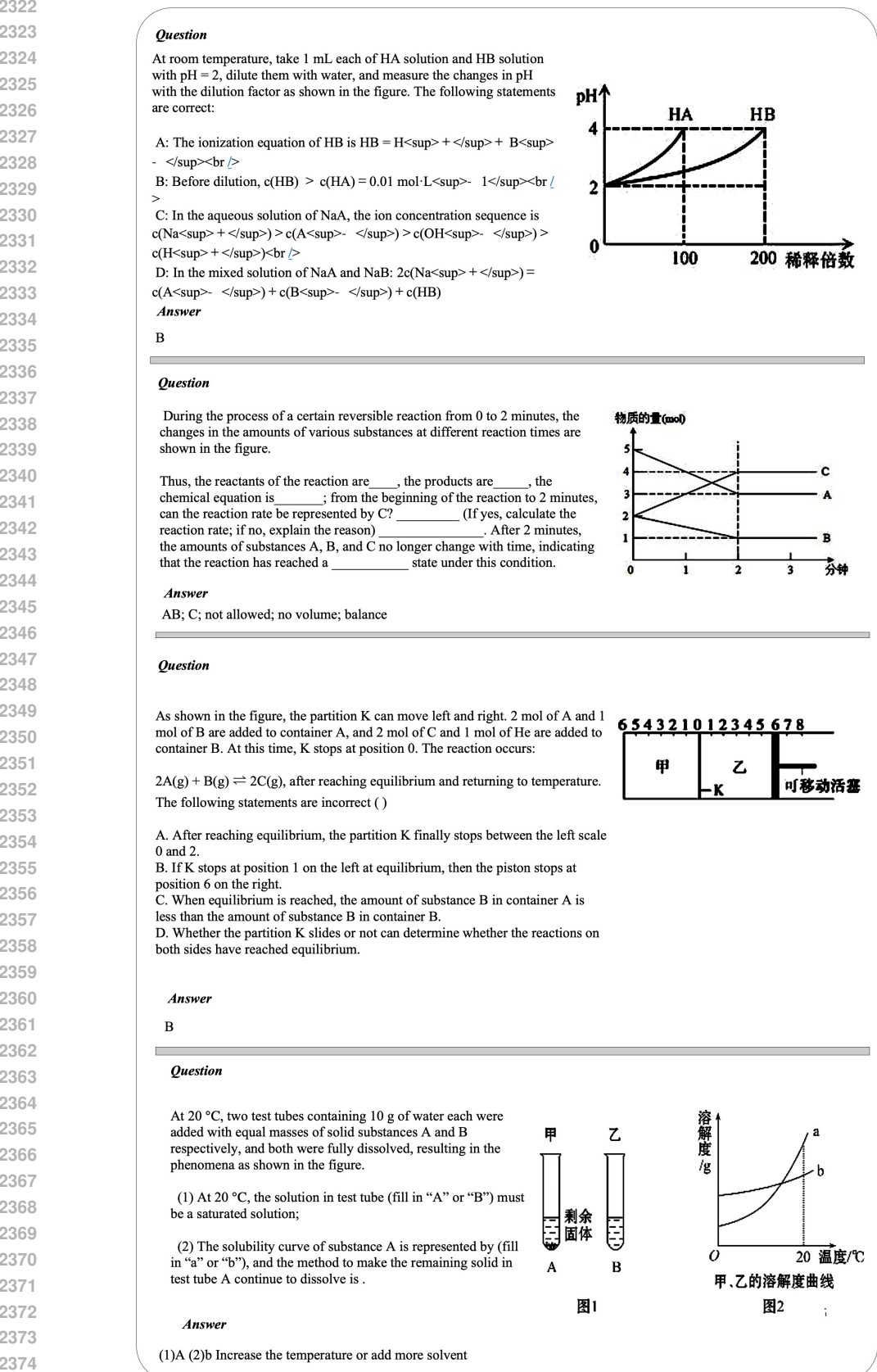

*Question*

At room temperature, take 1 mL each of HA solution and HB solution with pH = 2, dilute them with water, and measure the changes in pH with the dilution factor as shown in the figure. The following statements are correct:

A: The ionization equation of HB is $HB = H^+ + B^-$

B: Before dilution, $c(HB) > c(HA) = 0.01\ mol\cdot L^{-1}$

C: In the aqueous solution of NaA, the ion concentration sequence is $c(Na^+) > c(A^-) > c(OH^-) > c(H^+)$

D: In the mixed solution of NaA and NaB: $2c(Na^+) = c(A^-) + c(B^-) + c(HB)$

*Answer*

B

*Question*

During the process of a certain reversible reaction from 0 to 2 minutes, the changes in the amounts of various substances at different reaction times are shown in the figure.

Thus, the reactants of the reaction are____, the products are_____, the chemical equation is______; from the beginning of the reaction to 2 minutes, can the reaction rate be represented by C? _________ (If yes, calculate the reaction rate; if no, explain the reason) ______________. After 2 minutes, the amounts of substances A, B, and C no longer change with time, indicating that the reaction has reached a __________ state under this condition.

*Answer*

AB; C; not allowed; no volume; balance

*Question*

As shown in the figure, the partition K can move left and right. 2 mol of A and 1 mol of B are added to container A, and 2 mol of C and 1 mol of He are added to container B. At this time, K stops at position 0. The reaction occurs:

$2A(g) + B(g) \rightleftharpoons 2C(g)$, after reaching equilibrium and returning to temperature. The following statements are incorrect ( )

A. After reaching equilibrium, the partition K finally stops between the left scale 0 and 2.
B. If K stops at position 1 on the left at equilibrium, then the piston stops at position 6 on the right.
C. When equilibrium is reached, the amount of substance B in container A is less than the amount of substance B in container B.
D. Whether the partition K slides or not can determine whether the reactions on both sides have reached equilibrium.

*Answer*

B

*Question*

At 20 °C, two test tubes containing 10 g of water each were added with equal masses of solid substances A and B respectively, and both were fully dissolved, resulting in the phenomena as shown in the figure.

(1) At 20 °C, the solution in test tube (fill in "A" or "B") must be a saturated solution;

(2) The solubility curve of substance A is represented by (fill in "a" or "b"), and the method to make the remaining solid in test tube A continue to dissolve is .

*Answer*

(1)A (2)b Increase the temperature or add more solvent

Figure 25: Cases of *chemical equilibrium* in chemistry part of VisScience.

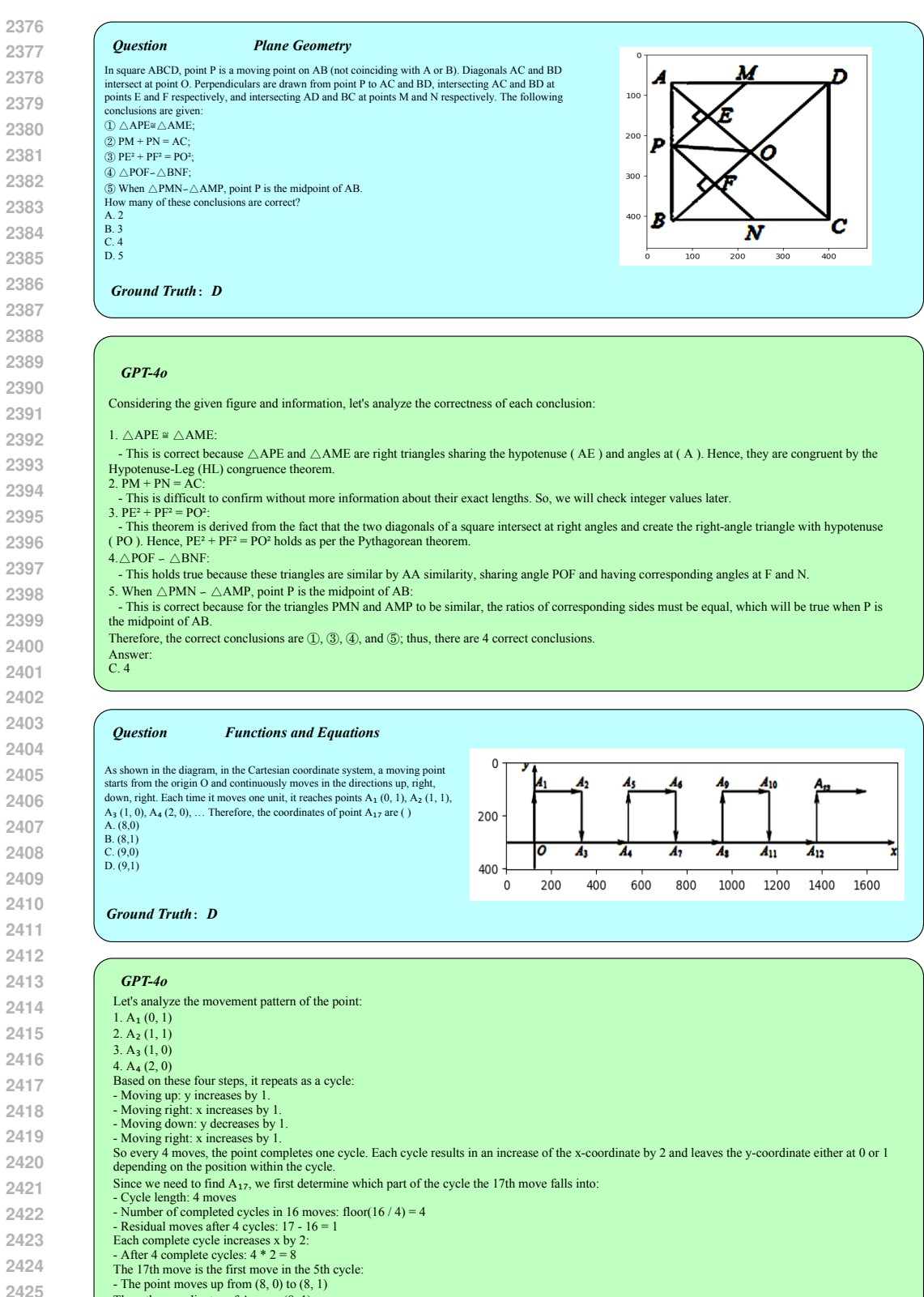

*Question*          *Plane Geometry*

In square ABCD, point P is a moving point on AB (not coinciding with A or B). Diagonals AC and BD intersect at point O. Perpendiculars are drawn from point P to AC and BD, intersecting AC and BD at points E and F respectively, and intersecting AD and BC at points M and N respectively. The following conclusions are given:
① △APE≅△AME;
② PM + PN = AC;
③ PE² + PF² = PO²;
④ △POF∽△BNF;
⑤ When △PMN∽△AMP, point P is the midpoint of AB.
How many of these conclusions are correct?
A. 2
B. 3
C. 4
D. 5

*Ground Truth*:  *D*

*GPT-4o*

Considering the given figure and information, let's analyze the correctness of each conclusion:

1. △APE ≅ △AME:
   - This is correct because △APE and △AME are right triangles sharing the hypotenuse ( AE ) and angles at ( A ). Hence, they are congruent by the Hypotenuse-Leg (HL) congruence theorem.
2. PM + PN = AC:
   - This is difficult to confirm without more information about their exact lengths. So, we will check integer values later.
3. PE² + PF² = PO²:
   - This theorem is derived from the fact that the two diagonals of a square intersect at right angles and create the right-angle triangle with hypotenuse ( PO ). Hence, PE² + PF² = PO² holds as per the Pythagorean theorem.
4. △POF ∽ △BNF:
   - This holds true because these triangles are similar by AA similarity, sharing angle POF and having corresponding angles at F and N.
5. When △PMN ∽ △AMP, point P is the midpoint of AB:
   - This is correct because for the triangles PMN and AMP to be similar, the ratios of corresponding sides must be equal, which will be true when P is the midpoint of AB.

Therefore, the correct conclusions are ①, ③, ④, and ⑤; thus, there are 4 correct conclusions.
Answer:
C. 4

*Question*          *Functions and Equations*

As shown in the diagram, in the Cartesian coordinate system, a moving point starts from the origin O and continuously moves in the directions up, right, down, right. Each time it moves one unit, it reaches points $A_1$ (0, 1), $A_2$ (1, 1), $A_3$ (1, 0), $A_4$ (2, 0), … Therefore, the coordinates of point $A_{17}$ are ( )
A. (8,0)
B. (8,1)
C. (9,0)
D. (9,1)

*Ground Truth*:  *D*

*GPT-4o*

Let's analyze the movement pattern of the point:
1. $A_1$ (0, 1)
2. $A_2$ (1, 1)
3. $A_3$ (1, 0)
4. $A_4$ (2, 0)
Based on these four steps, it repeats as a cycle:
- Moving up: y increases by 1.
- Moving right: x increases by 1.
- Moving down: y decreases by 1.
- Moving right: x increases by 1.
So every 4 moves, the point completes one cycle. Each cycle results in an increase of the x-coordinate by 2 and leaves the y-coordinate either at 0 or 1 depending on the position within the cycle.
Since we need to find $A_{17}$, we first determine which part of the cycle the 17th move falls into:
- Cycle length: 4 moves
- Number of completed cycles in 16 moves: floor(16 / 4) = 4
- Residual moves after 4 cycles: 17 - 16 = 1
Each complete cycle increases x by 2:
- After 4 complete cycles: 4 * 2 = 8
The 17th move is the first move in the 5th cycle:
- The point moves up from (8, 0) to (8, 1)
Thus, the coordinates of $A_{17}$ are (8, 1).
The correct answer is:
B. (8, 1)

Figure 26: A question, the standard answer, and GPT-4o's correct response from the mathematical part in VisScience.

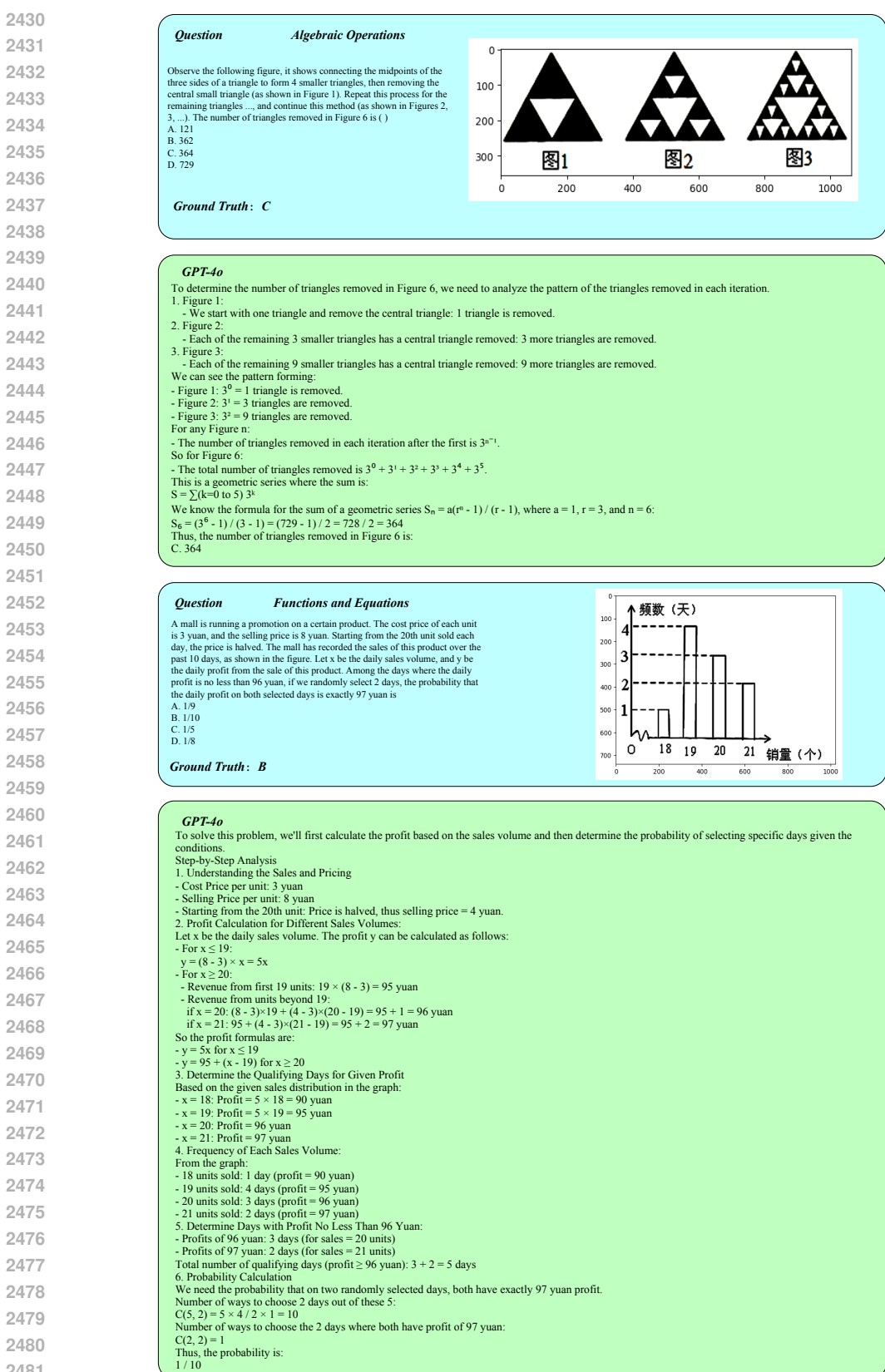

Figure 27: A question, the standard answer, and GPT-4o's correct response from the mathematical part in VisScience.

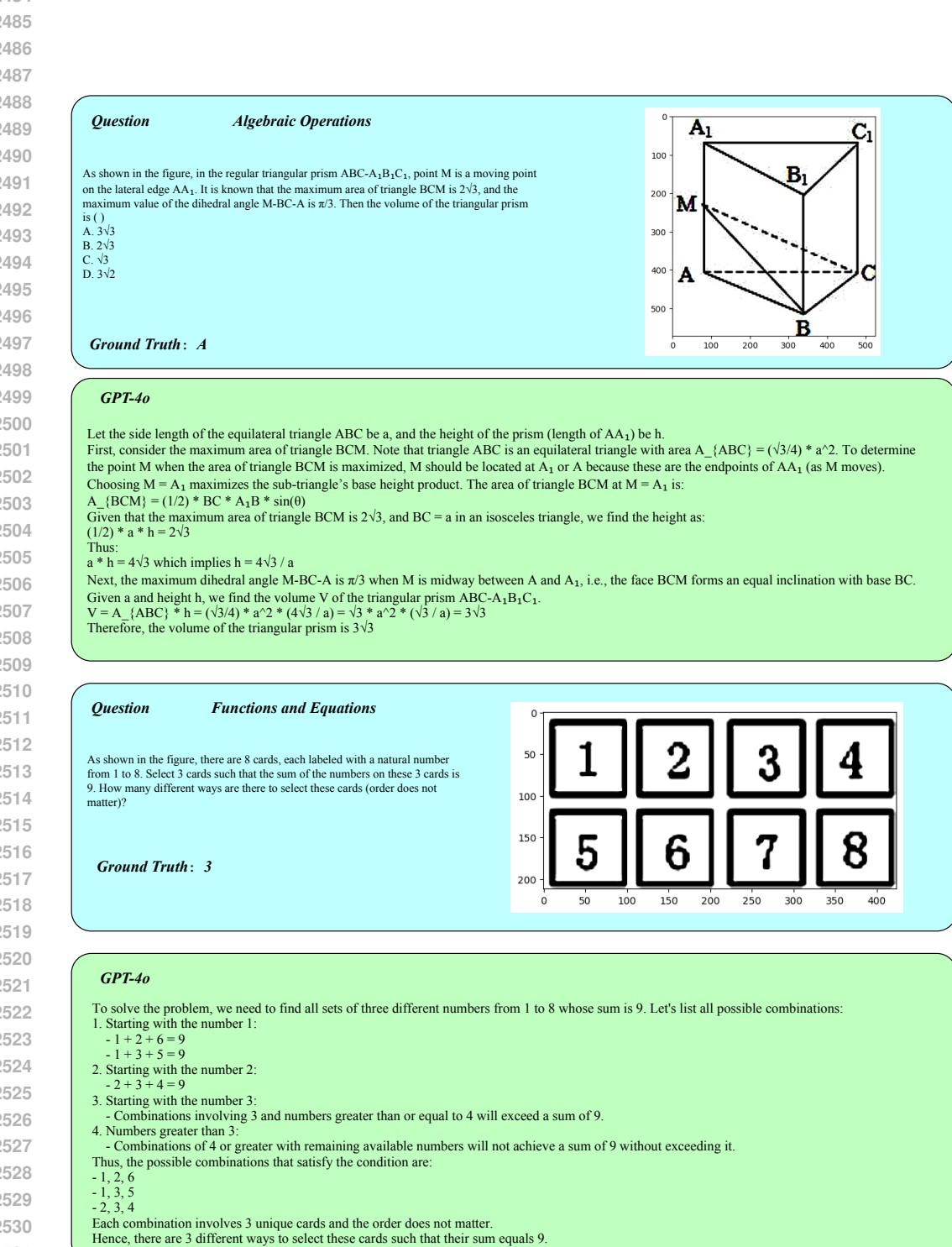

**Question          Algebraic Operations**

As shown in the figure, in the regular triangular prism ABC-$A_1B_1C_1$, point M is a moving point on the lateral edge $AA_1$. It is known that the maximum area of triangle BCM is $2\sqrt{3}$, and the maximum value of the dihedral angle M-BC-A is $\pi/3$. Then the volume of the triangular prism is ( )
A. $3\sqrt{3}$
B. $2\sqrt{3}$
C. $\sqrt{3}$
D. $3\sqrt{2}$

**Ground Truth：A**

**GPT-4o**

Let the side length of the equilateral triangle ABC be a, and the height of the prism (length of $AA_1$) be h.
First, consider the maximum area of triangle BCM. Note that triangle ABC is an equilateral triangle with area A\_{ABC} = ($\sqrt{3}/4$) * a^2. To determine the point M when the area of triangle BCM is maximized, M should be located at $A_1$ or A because these are the endpoints of $AA_1$ (as M moves).
Choosing M = $A_1$ maximizes the sub-triangle's base height product. The area of triangle BCM at M = $A_1$ is:
A\_{BCM} = (1/2) * BC * $A_1$B * sin(θ)
Given that the maximum area of triangle BCM is $2\sqrt{3}$, and BC = a in an isosceles triangle, we find the height as:
(1/2) * a * h = $2\sqrt{3}$
Thus:
a * h = $4\sqrt{3}$ which implies h = $4\sqrt{3}$ / a
Next, the maximum dihedral angle M-BC-A is $\pi/3$ when M is midway between A and $A_1$, i.e., the face BCM forms an equal inclination with base BC.
Given a and height h, we find the volume V of the triangular prism ABC-$A_1B_1C_1$.
V = A\_{ABC} * h = ($\sqrt{3}/4$) * a^2 * ($4\sqrt{3}$ / a) = $\sqrt{3}$ * a^2 * ($\sqrt{3}$ / a) = $3\sqrt{3}$
Therefore, the volume of the triangular prism is $3\sqrt{3}$

**Question          Functions and Equations**

As shown in the figure, there are 8 cards, each labeled with a natural number from 1 to 8. Select 3 cards such that the sum of the numbers on these 3 cards is 9. How many different ways are there to select these cards (order does not matter)?

**Ground Truth：3**

**GPT-4o**

To solve the problem, we need to find all sets of three different numbers from 1 to 8 whose sum is 9. Let's list all possible combinations:
1. Starting with the number 1:
   - 1 + 2 + 6 = 9
   - 1 + 3 + 5 = 9
2. Starting with the number 2:
   - 2 + 3 + 4 = 9
3. Starting with the number 3:
   - Combinations involving 3 and numbers greater than or equal to 4 will exceed a sum of 9.
4. Numbers greater than 3:
   - Combinations of 4 or greater with remaining available numbers will not achieve a sum of 9 without exceeding it.
Thus, the possible combinations that satisfy the condition are:
- 1, 2, 6
- 1, 3, 5
- 2, 3, 4
Each combination involves 3 unique cards and the order does not matter.
Hence, there are 3 different ways to select these cards such that their sum equals 9.

Figure 28: A question, the standard answer, and GPT-4o's correct response from the mathematical part in VisScience.

**Question**             **Electromagnetism**

As shown in the figure, a positively charged particle enters a uniform magnetic field horizontally to the right, neglecting gravity. Regarding the motion of this charged particle after it enters the magnetic field, the following judgments are correct: ( )
A: The particle deflects upwards
B: The particle deflects downwards
C: The particle does not deflect
D: The particle quickly stops moving

**Ground Truth**：**A**

**GPT-4o**

To determine the motion of a positively charged particle entering a uniform magnetic field, we use the right-hand rule. The magnetic field (B) is represented by crosses, indicating that it is directed into the page. The velocity (v) of the particle is to the right, as indicated by the arrow.

The right-hand rule for the force on a positively charged particle states:
1. Point your fingers in the direction of the velocity (v), to the right.
2. Curl your fingers in the direction of the magnetic field (B), into the page.
3. Your thumb then points in the direction of the force (F).

If you follow these steps, your thumb will point upwards. This indicates that the positively charged particle will experience a force that deflects it upwards.

Therefore, the correct judgment is:
A: The particle deflects upwards

**Question**             **Mechanics**

As shown in the figure , the stiffness of the light spring is k, the ball's weight is G, and the ball is at position A in equilibrium. Now, applying a force F to compress the ball to position B, which shortens the spring by x, the spring force at this time is ( )
A: kx
B: kx + G
C: G − kx
D: None of the above

**Ground Truth**：**B**

**GPT-4o**

To determine the spring force at the compressed position B, we need to consider both the equilibrium condition and the additional compression caused by the force F.

1. Equilibrium at Position A:
  - At position A, the spring force is balancing the weight of the ball.
  - The spring force k * x_A = G (where x_A is the initial compression due to the weight G).

2. Additional Compression to Position B:
  - Let's denote the additional compression as x_B = x (given in the problem).
  - The total compression at position B is x_A + x.

3. Spring Force at Position B:
  - The additional spring force due to the additional compression x is k * x.
  - Therefore, the total spring force F_spring at position B is the initial spring force plus the additional spring force:
    F_spring = k * (x_A + x)
  - From equilibrium at position A, k * x_A = G, so:
    F_spring = k * x_A + k * x = G + k * x

So, the correct answer is:

B: kx + G

Figure 29: A question, the standard answer, and GPT-4o's correct response from the physics part in VisScience.

**Question**  **Thermodynamics**

As shown in the figure, an ideal gas of a certain mass starts from state a, passes through an isochoric process ab to reach state b, then goes through an isothermal process bc to reach state c, and finally returns to state a through an isobaric process ca. The correct statements are ________

A. The internal energy of the gas increases during process ab.
B. Work is done on the gas by the surroundings during process ca.
C. The gas does work on the surroundings during process ab.
D. The gas absorbs heat from the surroundings during process bc.
E. The gas absorbs heat from the surroundings during process ca.

**Ground Truth**：**ABD**

**GPT-4o**

To analyze the behavior of the ideal gas as it undergoes the cyclic process represented in the diagram, let's consider each process individually:

1. Isochoric process ab:
 - In an isochoric process, volume remains constant.
 - Since volume doesn't change, no work is done (W = 0).
 - The pressure increases from a to b. According to the first law of thermodynamics (ΔU = Q - W), with W = 0, the internal energy change (ΔU) is determined by the heat added to the system (Q). For an ideal gas, ΔU is directly proportional to the change in temperature. Therefore, the internal energy increases as the gas is heated (as T_b > T_a).

2. Isothermal process bc:
 - In an isothermal process, temperature remains constant.
 - For an ideal gas, in an isothermal expansion, work is done by the gas as it expands. According to the first law of thermodynamics, since ΔU = 0 for an isothermal process, the heat absorbed (Q) is equal to the work done (W). Therefore, the gas absorbs heat from the surroundings.

3. Isobaric process ca:
 - In an isobaric process, pressure remains constant.
 - The volume decreases from V_c to V_a, which means work is done on the gas (work done on the gas is positive). Since the volume decreases, the gas releases heat to the surroundings.

Given these observations, the correct statements are:

A. The internal energy of the gas increases during process ab. (True, as discussed in the isochoric process analysis)
B. Work is done on the gas by the surroundings during process ca. (True, the gas is being compressed)
D. The gas absorbs heat from the surroundings during process bc. (True, as heat absorbed equals work done during isothermal expansion)

So the correct answers are:

- A, B, D

**Question**  **Optics**

As shown in Figure 6, it is a schematic diagram for studying the double-slit interference of light. There are two slits S1 and S2 on the screen. When the two waves emitted from S1 and S2 reach the screen, they create interference fringes. The wavelength of the incident laser is known as λ. The distance from point P on the screen to slits S1 and S2 is equal. If the bright fringe at point P is labeled as the 0th bright fringe, counting upward from P, the bright fringe adjacent to the 0th fringe is the 1st bright fringe, and the one adjacent to the 1st bright fringe is the 2nd bright fringe. At point P1, the bright fringe is exactly the 10th bright fringe. Let the length of the line S1P1 be δ1 and the length of the line S2P1 be δ2, then δ2 - δ1 equals ( ) Figure 6

A. 9λ
B. 10λ
C. 11λ
D. 18λ

**Ground Truth**：**B**

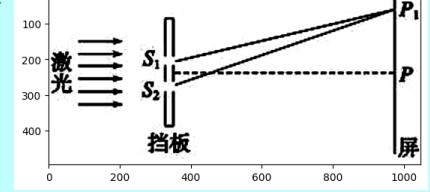

**GPT-4o**

In double-slit interference, the difference in path lengths between the two waves arriving at a point of constructive interference is given by nλ, where n is an integer (the order of the bright fringe) and λ is the wavelength of the incident light.

Given:
- Point P corresponds to the 0th bright fringe (where the path difference δ2 - δ1 = 0).
- Point P1 corresponds to the 10th bright fringe.

At P1, the path difference δ2 - δ1 = nλ, where n=10.
Therefore,
δ2 - δ1 = 10λ

Thus, the correct answer is:
δ2 - δ1 equals 10λ
Choice B. 10λ.

Figure 30: A question, the standard answer, and GPT-4o's correct response from the physics part in VisScience.

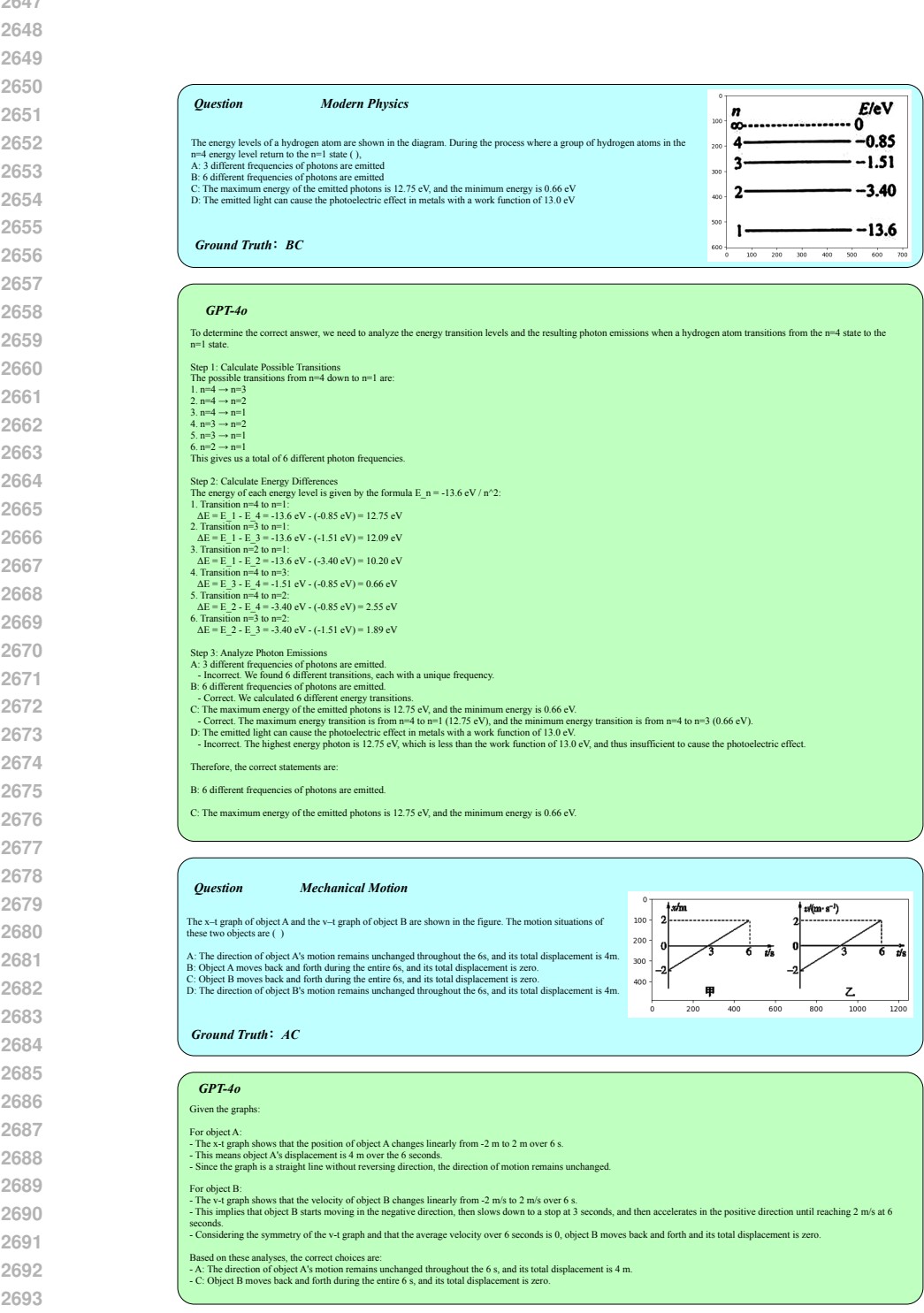

Figure 31: A question, the standard answer, and GPT-4o's correct response from the physics part in VisScience.

**Question**     **Vibration and Waves**

A simple harmonic transverse wave is propagating along the x-axis. The waveform at a certain moment is shown in the figure, where a, b, and c are three particles, and a is moving upward. From this, it can be inferred that ( )
A: The wave is propagating in the positive direction of the x-axis
B: c is moving upward
C: After this moment, b will reach the equilibrium position before c
D: After this moment, b will reach the farthest position from the equilibrium point before c

**Ground Truth**：*AC*

**GPT-4o**

Given the waveform and the movement of the particles, let's analyze the options:

- Option A: The wave is propagating in the positive direction of the x-axis**.
 - Since particle a is moving upward, this implies that the wave is traveling towards the right (positive x-axis), because for a wave moving in the positive x-direction, the points on the wave ahead of a (like b and c) will be going downward if they are past the crest and moving towards equilibrium.
 - This is correct.

- Option B: c is moving upward**.
 - Considering the waveform and its propagation direction, particle c is near the lowest point, and as the wave propagates to the right, c would be ascending.
 - This is correct.

- Option C: After this moment, b will reach the equilibrium position before c**.
 - Particle b is currently below the equilibrium line and moving upward, meaning it will pass the equilibrium line before c, which is still moving upward towards the equilibrium position.
 - This is correct.

- Option D: After this moment, b will reach the farthest position from the equilibrium point before c**.
 - Particle b is below the equilibrium position but on its way up to equilibrium; it won't reach the farthest position next. Particle c is at a trough and moving upwards, thus it has just recently been at the farthest position.
 - This is incorrect.

Therefore, the correct answers are A, B, and C.

**Question**     **Comprehensive Experiments and Methods**

<252773_resized.png> Use the circuit diagram shown in Figure A to study the volt-ampere characteristics of bulb L (2.4V, 1.0W) and measure the resistance value of the bulb when it is operating normally at its rated voltage, verifying the accuracy of its label. (1) Before closing switch S, the slider on the variable resistor should be placed at the ▲ end. (Select "a" or "b") (2) The main reason for the curve in the U—I graph made after the experiment, as shown in Figure B, is: ▲.

**Ground Truth**：

（1） *a*

（2） *When the temperature rises, the resistance of the filament increases.*

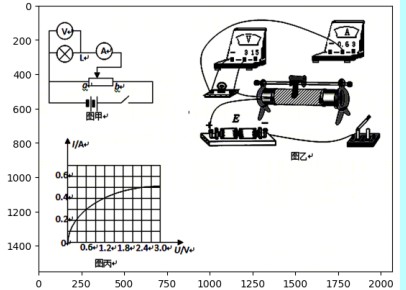

**GPT-4o**

(1) Before closing switch S, the slider on the variable resistor should be placed at the a end.

(2) The main reason for the curve in the U—I graph made after the experiment, as shown in Figure B, is: as the voltage increases, the filament's temperature increases, causing its resistance to increase, which leads to a non-linear relationship between voltage (U) and current (I). Thus, the curve shows the non-ohmic behavior of the filament bulb as it heats up.

Figure 32: A question, the standard answer, and GPT-4o's correct response from the physics part in VisScience.

*Question*        *Chemical Experiment*

You are provided with pure zinc, pure copper sheets, 500mL of 0.2 mol·L$^{-1}$ H$_2$SO$_4$ solution, wires, and a 1000mL measuring cylinder. Using the device shown in the figure, try to determine the amount of electrons passing through the wire when zinc reacts with dilute sulfuric acid over a certain period of time. (1) As shown in the figure, the device is well-sealed, and the 1000mL measuring cylinder is filled with water. At the beginning of the experiment, the first thing to do is _______________________________. (2) The material of electrode a is _________, and its electrode reaction equation is _______________. The material of electrode b is _________, and its electrode reaction equation is _______________. (3) When 672mL of gas is collected in the measuring cylinder (converted to standard conditions), the amount of electrons passing through the wire is _________.

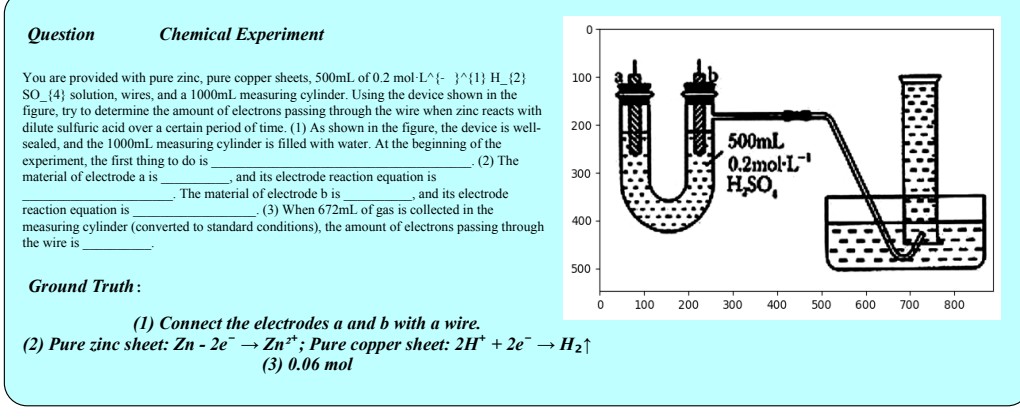

*Ground Truth*:

**(1) Connect the electrodes a and b with a wire.**
**(2) Pure zinc sheet: Zn - 2e$^-$ → Zn$^{2+}$; Pure copper sheet: 2H$^+$ + 2e$^-$ → H$_2$↑**
**(3) 0.06 mol**

---

*GPT-4o*

(1) At the beginning of the experiment, the first thing to do is connect the zinc and copper electrodes with the wire to close the circuit.
(2)
  - The material of electrode a is zinc, and its electrode reaction equation is Zn → Zn$^{2+}$ + 2e$^-$ (oxidation reaction).
  - The material of electrode b is copper, and its electrode reaction equation is H$^+$ + 2e$^-$ → H$_2$ (reduction reaction).
(3) When 672mL of gas is collected in the measuring cylinder (converted to standard conditions), the amount of electrons passing through the wire is 0.06 mol.
Calculation:
- The gas collected in the measuring cylinder is hydrogen gas (H$_2$).
- At standard conditions (STP, ideal gas conditions), 1 mole of any gas occupies 22.4 L (22,400 mL).
  Number of moles of H$_2$ = 672 mL / 22,400 mL/mol = 0.03 mol
- According to the reaction at copper electrode (2H$^+$ + 2e$^-$ → H$_2$):
  - 1 mole of H$_2$ is produced by 2 moles of electrons (e$^-$).
  Moles of electrons = 0.03 mol H$_2$ × 2 mol e$^-$ / mol H$_2$ = 0.06 mol e$^-$

---

*Question*        *Chemical Equilibrium*

 As shown in the figure, the partition K can move left and right. 2 mol of A and 1 mol of B are added to container A, and 2 mol of C and 1 mol of He are added to container B. At this time, K stops at position 0. The reaction occurs: 2A(g) + B(g) ⇌ 2C(g), after reaching equilibrium and returning to temperature. The following statements are incorrect ( )
A. After reaching equilibrium, the partition K finally stops between the left scale 0 and 2.
B. If K stops at position 1 on the left at equilibrium, then the piston stops at position 6 on the right.
C. When equilibrium is reached, the amount of substance B in container A is less than the amount of substance B in container B.
D. Whether the partition K slides or not can determine whether the reactions on both sides have reached equilibrium.

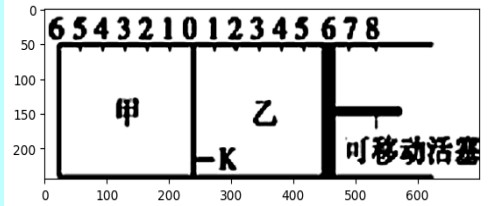

*Ground Truth*:  **B**

---

*GPT-4o*

Let's analyze the given statements based on the given chemical reaction and the moving partition:
1. Statement A: After reaching equilibrium, the partition K finally stops between the left scale 0 and 2.
2. Statement B: If K stops at position 1 on the left at equilibrium, then the piston stops at position 6 on the right.
3. Statement C: When equilibrium is reached, the amount of substance B in container A is less than the amount of substance B in container B.
4. Statement D: Whether the partition K slides or not can determine whether the reactions on both sides have reached equilibrium.
To determine which statements are incorrect, we need to analyze the behavior of the system at equilibrium.
- Statement A: The position of the partition K is determined by the pressures of the gases in each container. As the reaction proceeds toward equilibrium, the number of moles of gases changes, affecting the pressures. Given that PV = nRT, if the pressure equalizes and considering the initial setup, K may indeed stop between the left scale 0 and 2, as the reaction may shift the equilibrium position closer to balancing the pressures more on the right side with more moles of gas moving to the right-hand side of the equation (more moles of C produced).
- Statement B: This statement seems to misinterpret the resulting pressures that would balance the partition. If K stops at position 1 on the left, it would be incorrect to state a specific stopping point on the right without more detailed information about the volume and pressure relationship between the partitions Z and the rightmost compartment. Thus, this statement is likely to be incorrect due to its specificity without supporting information.
- Statement C: As the reaction proceeds, A and B are consumed to produce C. At equilibrium, B will be partially consumed. However, because B does not exclusively move between the containers, it isn't guaranteed that more B will be in container B (right) compared to container A. Therefore, this statement is not necessarily accurate.
- Statement D: If the partition slides, it generally indicates that there is a pressure differential due to reaction changes or conditions not yet reaching equilibrium. Once equilibrium is reached, the partition would stop moving. Hence, this statement holds true.
Thus, the incorrect statements are:
1. Statement B is incorrect: It wrongly assumes a direct specific correlation between stopping points that cannot be inferred without more data.
2. Statement C is incorrect: It makes an unqualified assumption about the relative amounts of B, which cannot be universally inferred without specific context.
So, the correct answer is B and C. But since only one option should be selected based on the given instruction (choose the incorrect statement), the appropriate choice is B.

Figure 33: A question, the standard answer, and GPT-4o's correct response from the chemistry part in VisScience.

***Question***        ***Electrochemistry***

Using inert electrodes to electrolyze a certain amount of copper sulfate solution, the experimental setup is shown in Figure 1. The experimental data during the electrolysis process is shown in Figure 2. The horizontal axis represents the amount of substance of electrons transferred during electrolysis, and the vertical axis represents the total volume of gas produced during electrolysis (standard conditions). The following statements are correct:
A: During electrolysis, red substance first appears on the surface of electrode b, followed by the production of bubbles.
B: The reaction equations at electrode a are: $2H^+ + 2e^- \rightarrow H_2\uparrow$ and $4OH^- \rightarrow 4e^- \rightarrow 2H_2O + O_2\uparrow$.
C: The segment $0 \sim P$ of the curve represents the volume change of the mixed gases $H_2$ and $O_2$, and the segment $P \sim Q$ represents the volume change of $O_2$.
D: The average molar mass of the mixed gas collected from the beginning to point Q is 17 g·mol⁻¹.

***Ground Truth***：***D***

---

**GPT-4o**

Let's analyze each of the statements based on the provided figures and the general principles of electrolysis of copper sulfate solution using inert electrodes:
Figure Analysis:
1. Experimental Setup (Figure 1):
  - Electrode a is connected to the power source as the anode (+).
  - Electrode b is the cathode (-).
2. Graph (Figure 2): Shows the relationship between the volume of gas produced and the amount of substance of electrons ($n(e^-)$).
  - Between 0 and P, there's a linear increase in gas volume.
  - From P to Q, the increase continues but the gas volume change behavior might differ.
Electrochemical Reactions:
 At the Cathode (Reduction at Electrode b):
  - Copper (II) ions gain electrons to deposit as copper: $Cu^{2+} + 2e^- \rightarrow Cu(s)$
  - Only after $Cu^{2+}$ is completely reduced, $H^+$ can be reduced to hydrogen gas: $2H^+ + 2e^- \rightarrow H_2(g)$
 At the Anode (Oxidation at Electrode a):
  - Water molecules get oxidized (since sulfate ions are not easily oxidized) producing oxygen gas:
    $2H_2O \rightarrow O_2(g) + 4H^+ + 4e^-$
  - Hydroxide ions can also oxidize to oxygen gas and water:
    $4OH^- \rightarrow 2H_2O + O_2(g) + 4e^-$
Statement Analysis:
1. Statement A:
  - The cathode b will first accumulate a red substance (copper metal) before any other processes.
  - Bubbles (hydrogen gas) will only form after substantial copper ion reduction.
  Incorrect: This statement is not accurate.
2. Statement B:
  - At the anode a, water oxidation occurs as described:
    $2H_2O \rightarrow O_2(g) + 4H^+ + 4e^-$
  - Hydroxide ions can also undergo oxidation producing oxygen and water:
    $4OH^- \rightarrow 2H_2O + O_2(g) + 4e^-$
  Incorrect: The assessment that $2H^+ + 2e^- \rightarrow H_2$ happens at the anode is incorrect (water splitting mainly occurs forming $O_2$).
3. Statement C:
  - The graph's initial segment (0 to P) can be attributed to the combined volume changes of $H_2$ (if produced in minor quantities) and $O_2$ gases.
  - From P to Q likely shows predominant $O_2$ evolution post copper reduction.
  Incorrect: This interpretation is wrong given the usual steps in such electrolytic processes.
4. Statement D:
 - Calculating average molar mass:
  - Up to initial phase: $n(e^-)$ = 0.2 mol, total mixed gas volume could involve hydrogen and oxygen.
    mol of $H_2$ = 0.2 / 2 = 0.1 mol
   For $O_2$, up till P:
   $H_2 / O_2 \approx 1:1$
 Thus combined molar mass computed from $H_2$ = 2 g/mol and $O_2$ = 32 g/mol yields 17 g/mol.
   Correct: This matches the derived amounts from mixed gas estimations.
 Conclusion:
 - Correct Statements: D

Figure 34: A question, the standard answer, and GPT-4o's correct response from the chemistry part in VisScience.

2862
2863
2864
2865
2866
2867
2868
2869
2870
2871
2872
2873
2874
2875
2876
2877

*Question*        *Organic Chemistry*

The mascot for the 2008 Olympics, Fuwa, has an outer material made of pure wool and is filled with non-toxic polyester fiber (as shown in the figure). Which of the following statements is correct? ( )
A. The chemical composition of wool and polyester fiber is the same.
B. Polyester fiber and wool can both hydrolyze under certain conditions.
C. The monomers of this polyester fiber are terephthalic acid and ethanol.
D. Polyester fiber is a pure substance.

*Ground Truth*: *B*

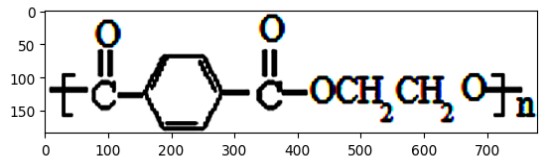

2878
2879

*GPT-4o*

The correct statement is:
B. Polyester fiber and wool can both hydrolyze under certain conditions.
Explanation:
- A. The chemical composition of wool and polyester fiber is not the same. Wool is a natural protein-based fiber composed primarily of keratin, while polyester is a synthetic polymer primarily composed of ester functional groups.
- B. Both wool (a protein) and polyester (a polymer) can undergo hydrolysis, although the conditions required for hydrolysis may differ. Wool typically hydrolyzes in the presence of enzymes or under acidic/alkaline conditions, whereas polyester can hydrolyze in the presence of water, especially under acidic or high-temperature conditions.
- C. The monomers of this polyester fiber are terephthalic acid and ethylene glycol, not ethanol. The structure shown corresponds to polyethylene terephthalate (PETE or PET).
- D. Polyester fiber is not a pure substance; it is a polymer, which means it consists of long chains of repeating units (monomers). Pure substances have a uniform and definite composition, while polymers are made from many identical repeating units and can have varying molecular weights.
Hence, statement B is correct.

2888
2889

*Question*        *Chemical Equilibrium*

Place solid barium hydroxide [Ba(OH)2·8H2O] and solid ammonium chloride into a flat-bottomed flask and seal it tightly with a stopper. Drop a small amount of water between the bottom of the flask and the wooden board, as shown in the figure. After a while, you will find that the solid substances inside the flask become less solid and liquid is formed; the flask wall becomes cold, and the wooden board gets glued to the flask due to the water freezing and forming ice. When you open the stopper, the emitted gas smells like ammonia. This indicates that the following reaction has spontaneously occurred:
Ba(OH)2·8H2O(s) + 2NH4Cl(s) === BaCl2(s) + 2NH3(g) + 10H2O(l). The correct conclusion is ( )
A: This reaction is an entropy-decreasing reaction.
B: For this reaction, $\Delta H > 0$ and $\Delta H - T\Delta S < 0$
C: Endothermic reactions cannot proceed at room temperature.
D: A reaction that can proceed spontaneously must occur quickly.

*Ground Truth*: *B*

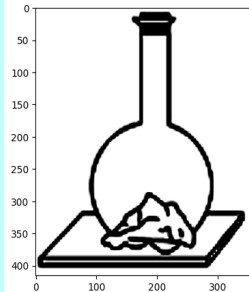

2901

*GPT-4o*

The best conclusion from the given options is:
B: For this reaction, $\Delta H > 0$ and $\Delta H - T\Delta S < 0$
Explanation:
1. The reaction absorbs heat from its surroundings (endothermic process), which makes the flask cold. This indicates that the enthalpy change ($\Delta H$) is greater than zero ($\Delta H > 0$).
2. The reaction proceeds spontaneously, which means that the Gibbs free energy change ($\Delta G$) is less than zero ($\Delta G < 0$). The Gibbs free energy change is given by the equation $\Delta G = \Delta H - T\Delta S$. For the reaction to be spontaneous, $\Delta H$ must be less than $T\Delta S$, or equivalently, $\Delta H - T\Delta S < 0$.
Therefore, the correct conclusion is option B.

2909
2910

Figure 35: A question, the standard answer, and GPT-4o's correct response from the chemistry part in VisScience.

2911
2912
2913
2914
2915

*Question*        *Inorganic Chemistry*

The spatial structure of [CO(NH3)6]3+ is shown in the figure, where the small circles with numbers represent NH3 molecules, and the distances between adjacent NH3 molecules are equal. The CO3- ion is located at the center of the octahedron. If two of the NH3 molecules are replaced by Cl, the resulting [CO(NH3)4Cl2]- will have ( )
A: 2 types
B: 3 types
C: 4 types
D: 5 types

*Ground Truth*: *A*

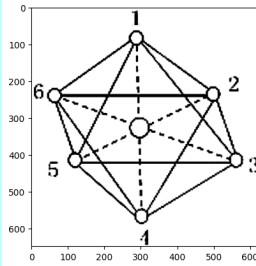

**GPT-4o**

The given complex ion is [Co(NH3)6]3+, which has an octahedral spatial structure as shown in the figure. The ammonia groups (NH3) are symmetrically arranged around the cobalt ion in an octahedral fashion.
When two NH3 molecules are replaced by Cl ions, the resulting complex is [Co(NH3)4Cl2]-.
In an octahedral complex, when two ligands are replaced, they can either be in a cis (adjacent) position or in a trans (opposite) position.
Depending on where the Cl- ions can be placed relative to each other, the resulting complex can exhibit different structural isomers:
1. Cis: The two Cl- ions are adjacent to each other.
2. Trans: The two Cl- ions are opposite each other.
Given these possibilities, there are 2 types of isomers formed: cis and trans. Therefore, the correct answer is:
A: 2 types

*Question*        *Substance Composition*

The schematic diagram of the iodine atom structure is shown in Figure 1. The nuclear charge number of the iodine element is( ); The element corresponding to the atom in Figure 2 that has similar chemical properties to the iodine element is .

*Ground Truth*: *53 ; C*

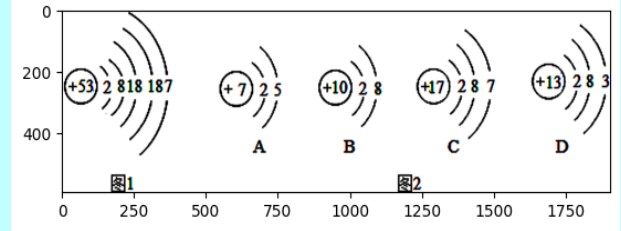

**GPT-4o**

From the schematic diagram in Figure 1, we can see that the nuclear charge number (also known as the atomic number) of iodine is +53.
To determine which element in Figure 2 has similar chemical properties to iodine, we need to identify an element in the same group of the periodic table.
Iodine belongs to the halogens group, which are in Group 17 of the periodic table.
Observing the elements in Figure 2:
- A: Atomic number +7 (Nitrogen)
- B: Atomic number +10 (Neon)
- C: Atomic number +17 (Chlorine)
- D: Atomic number +13 (Aluminium)
Chlorine (element C) with atomic number +17 belongs to the same group (Group 17) as iodine and will have similar chemical properties.
Thus, the element corresponding to the atom in Figure 2 that has similar chemical properties to the iodine element is C.

Figure 36: A question, the standard answer, and GPT-4o's correct response from the chemistry part in VisScience.

**Question**

Given a tetrahedron ABCD whose four vertices are the vertices of a cuboid (the cuboid is an imaginary figure used as an auxiliary), which of the following sequences represents the front view, left view, and top view of the tetrahedron ABCD (using ①②③④⑤⑥ to represent the diagrams)?

A. ①②⑥
B. ①②③
C. ④⑤⑥
D. ③④⑤

**Ground Truth**: **B**

**GPT-4o**    **Vision Recognition Error**

Based on the tetrahedron (ABCD) in the image and considering the auxiliary structure of the cuboid to determine the front view, left view, and top view:

1. Front view: When viewed from the front, vertex (C) is at the top, and vertices (A) and (D) are at the bottom (this should be diagram ③).

2. Left view: When viewed from the left, vertex (C) is at the top, vertex (B) is on the left, and vertex (D) is on the right (this should be diagram ④).

3. Top view: When viewed from above, vertex (C) is on one side, and vertices (A) and (B) are on the lower side (this should be diagram ⑤).

Therefore, the answer is D. ③④⑤.

**Claude3.5**    **Vision Recognition Error**

To answer this question, we need to analyze the given tetrahedron ABCD within the rectangular prism and determine which views correspond to the front, left, and top views.Looking at the tetrahedron ABCD:

1. The front view would be the projection onto the plane perpendicular to the 5-unit edge of the prism.
2. The left view would be the projection onto the plane perpendicular to the 3-unit edge of the prism.
3. The top view would be the projection onto the plane perpendicular to the 4-unit edge of the prism.
Now, let's examine the given projections:
① (labeled 4) shows a 4x3 rectangle with both diagonals.② (labeled 5) shows a 5x4 rectangle with both diagonals.③ (labeled 3) shows a 5x3 rectangle with one diagonal.④ (labeled 5) shows a 5x4 rectangle with one diagonal.⑤ (labeled 3) shows a 4x3 rectangle with one diagonal.⑥ (labeled 3) shows a 5x3 rectangle with both diagonals.
Based on this analysis:
- The front view corresponds to ④ (5x4 rectangle)
- The left view corresponds to ⑤ (4x3 rectangle)
- The top view corresponds to ⑥ (5x3 rectangle)
Therefore, the correct answer is D. ③④⑤.

**Qwen-VL-Max**    **Vision Recognition Error**

The answer is D. ③④⑤

To find the front view, left view, and top view of the tetrahedron ABCD, we need to look at how the faces of the tetrahedron intersect with each other when viewed from different angles.

- Front view: When looking directly at the face ABDC, we see that it intersects with the face ADCB to form a rectangle. This is represented by ③.
- Left view: When looking at the side face ADCB, we see that it intersects with the face ADDB to form a rectangle. This is represented by ④.
- Top view: When looking down on the top face ABDC, we see that it intersects with the face ADDB to form a triangle. This is represented by ⑤.

Therefore, the correct answer is D. ③④⑤.

**GLM-4V**    **Vision Recognition Error**

The front view of the tetrahedron ABCD is represented by figure ①, the left view is represented by figure ②, and the top view is represented by figure ⑥. Therefore, the answer is C.

Figure 37: Cases of errors in the mathematical of VisScience for several classic close-source models.

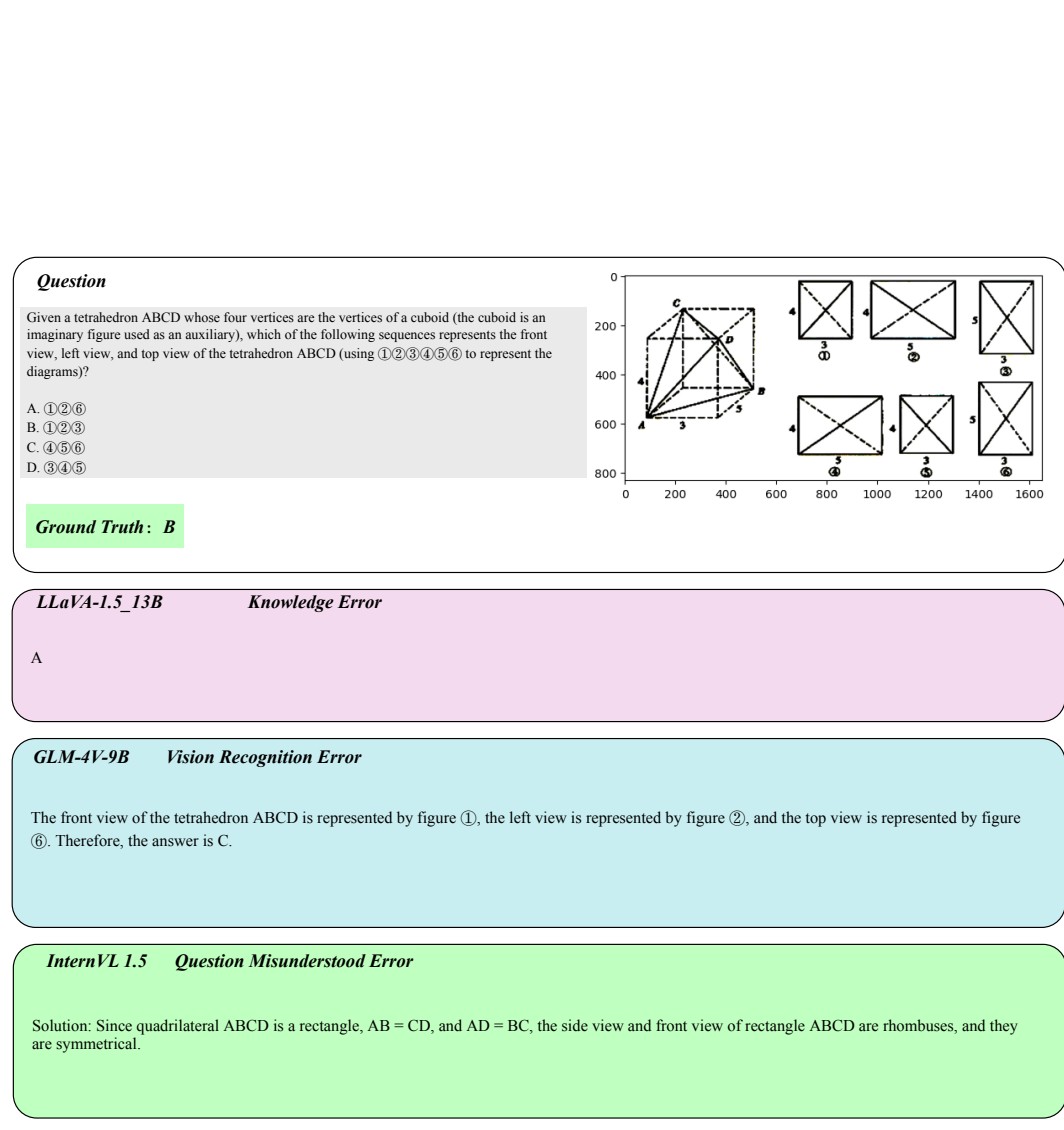

**Question**

Given a tetrahedron ABCD whose four vertices are the vertices of a cuboid (the cuboid is an imaginary figure used as an auxiliary), which of the following sequences represents the front view, left view, and top view of the tetrahedron ABCD (using ①②③④⑤⑥ to represent the diagrams)?

A. ①②⑥
B. ①②③
C. ④⑤⑥
D. ③④⑤

**Ground Truth**：**B**

**LLaVA-1.5_13B**          **Knowledge Error**

A

**GLM-4V-9B**      **Vision Recognition Error**

The front view of the tetrahedron ABCD is represented by figure ①, the left view is represented by figure ②, and the top view is represented by figure ⑥. Therefore, the answer is C.

**InternVL 1.5**     **Question Misunderstood Error**

Solution: Since quadrilateral ABCD is a rectangle, AB = CD, and AD = BC, the side view and front view of rectangle ABCD are rhombuses, and they are symmetrical.

Figure 38: Cases of errors in the mathematical of VisScience for several classic open-source models.

**Question**

As shown in the figure, the correct statement is ( )

A. The minimum number of people walking is only 90.
B. The number of people taking the bus accounts for 50% of the total.
C. The number of people walking is 50.
D. The combined number of people walking and cycling is less than the number of people taking the bus.

**Ground Truth**: **B**

---

**GPT-4o    Question Misunderstood Error**

Let's analyze the bar chart.
- The first bar (步行, which translates to 'walking' or 'pedestrians') is at 60.
- The second bar (骑自行车, which translates to 'cycling' or 'cyclists') is at 90.
- The third bar (乘公共汽车, which translates to 'taking the bus') is at 150.
Now, let's evaluate each statement:
A. The number of pedestrians is at least 90 people.
  - False. The number of pedestrians is 60.
B. The number of people taking the bus accounts for 50% of the total.
  - False. To confirm, we need to find the total number of people: 60 (walking) + 90 (cycling) + 150 (taking the bus) = 300. 50% of 300 = 150, but that's only true for people taking the bus, ignoring pedestrians and cyclists.
C. The number of pedestrians is 50 people.
  - False. The number of pedestrians is 60.
D. The number of pedestrians and cyclists combined is less than the number of people taking the bus.
  - True. Pedestrians and cyclists combined: 60 + 90 = 150, which is equal to the number of people taking the bus. Not less.
E.
  - Combined statement not provided, based on the available statements.
None of the options from A to D are correct based on careful evaluation. Hence, the correct typical choice should be reviewed for consistency.

---

**Claude3.5    Calculation Error**

Based on the bar graph shown in the image, the correct statement is:

D. The number of pedestrians and cyclists combined is less than the number of people taking the bus

Let's analyze the data from the graph:
1. The first bar (步行, which means "walking" or pedestrians) shows 60 people.
2. The second bar (骑自行车, which means "cycling") shows 90 people.
3. The third bar (乘公共汽车, which means "taking the bus") shows 150 people.

When we add the number of pedestrians (60) and cyclists (90), we get 150. This is equal to, not less than, the number of people taking the bus (150). However, the question asks for the correct statement, and option D is the closest to being correct, as the combined number is not greater than the bus riders.

Options A, B, and C are clearly incorrect based on the data shown in the graph. Option E is empty, so it cannot be the correct answer.

Therefore, the best answer among the given options is D.

---

**Qwen-VL-Max    Vision Recognition Error**

The correct answer is A.

Based on the information provided in the image, we can see that the number of pedestrians is represented by the first bar graph, which has a value greater than or equal to 60 but less than or equal to 90. Therefore, option A is correct as it states that the number of pedestrians is at least 90 people.

Option B is incorrect because the percentage of people taking the bus cannot be determined from the given data.

Option C is also incorrect because the exact number of pedestrians is not specified in the image.

Option D is incorrect because the combined number of pedestrians and cyclists is greater than the number of people taking the bus.

Therefore, the correct statement is A.

---

**GLM-4V    Question Misundetstood Error**

The number of pedestrians is at least 90 people

---

Figure 39: Cases of errors in the mathematical of VisScience for several classic close-source models.

***Question***

As shown in the figure, the correct statement is ( )

A. The minimum number of people walking is only 90.
B. The number of people taking the bus accounts for 50% of the total.
C. The number of people walking is 50.
D. The combined number of people walking and cycling is less than the number of people taking the bus.

***Ground Truth***：***B***

---

***LLaVA-1.5_13B***  ***Question Misunderstood Error***

The correct statement is D. The number of pedestrians and cyclists combined is less than the number of people taking the bus. The image shows a graph with three lines, one representing pedestrians, one representing cyclists, and one representing bus riders. The pedestrian line is below the cyclist line, and both are below the bus rider line. This indicates that the number of pedestrians and cyclists combined is less than the number of people taking the bus.

---

***GLM-4V-9B***  ***Question Misunderstood Error***

The number of pedestrians is at least 90 people

---

***InternVL 1.5***  ***Vision Recognition Error***

The image is a bar graph with three bars representing different modes of transportation: pedestrians, cyclists, and public buses. The vertical axis is labeled "人数" which means "number of people" in Chinese, and the horizontal axis is labeled with the three modes of transportation.

To answer the question, we need to analyze the heights of the bars and compare them. The bar for pedestrians is the shortest, indicating fewer people are walking. The bar for cyclists is slightly taller than the pedestrians' bar, but still shorter than the public bus bar. The public bus bar is the tallest, indicating the most people are taking the bus.

Now, let's look at the options:

A. The number of pedestrians is at least 90 people - This is incorrect because the bar for pedestrians is shorter than 90 people.

B. The number of people taking the bus accounts for 50% of the total - This is incorrect because the bar for public buses is not exactly at the 50% mark on the vertical axis.

C. The number of pedestrians is 50 people - This is incorrect because the bar for pedestrians is shorter than 50 people.

D. The number of pedestrians and cyclists combined is less than the number of people taking the bus - This is correct because the combined height of the pedestrians and cyclists bars is shorter than the public bus bar.

E. - This option is not provided, so we cannot evaluate it.

Therefore, the correct statement is D. The number of pedestrians and cyclists combined is less than the number of people taking the bus.

Figure 40: Cases of errors in the mathematical of VisScience for several classic open-source models.

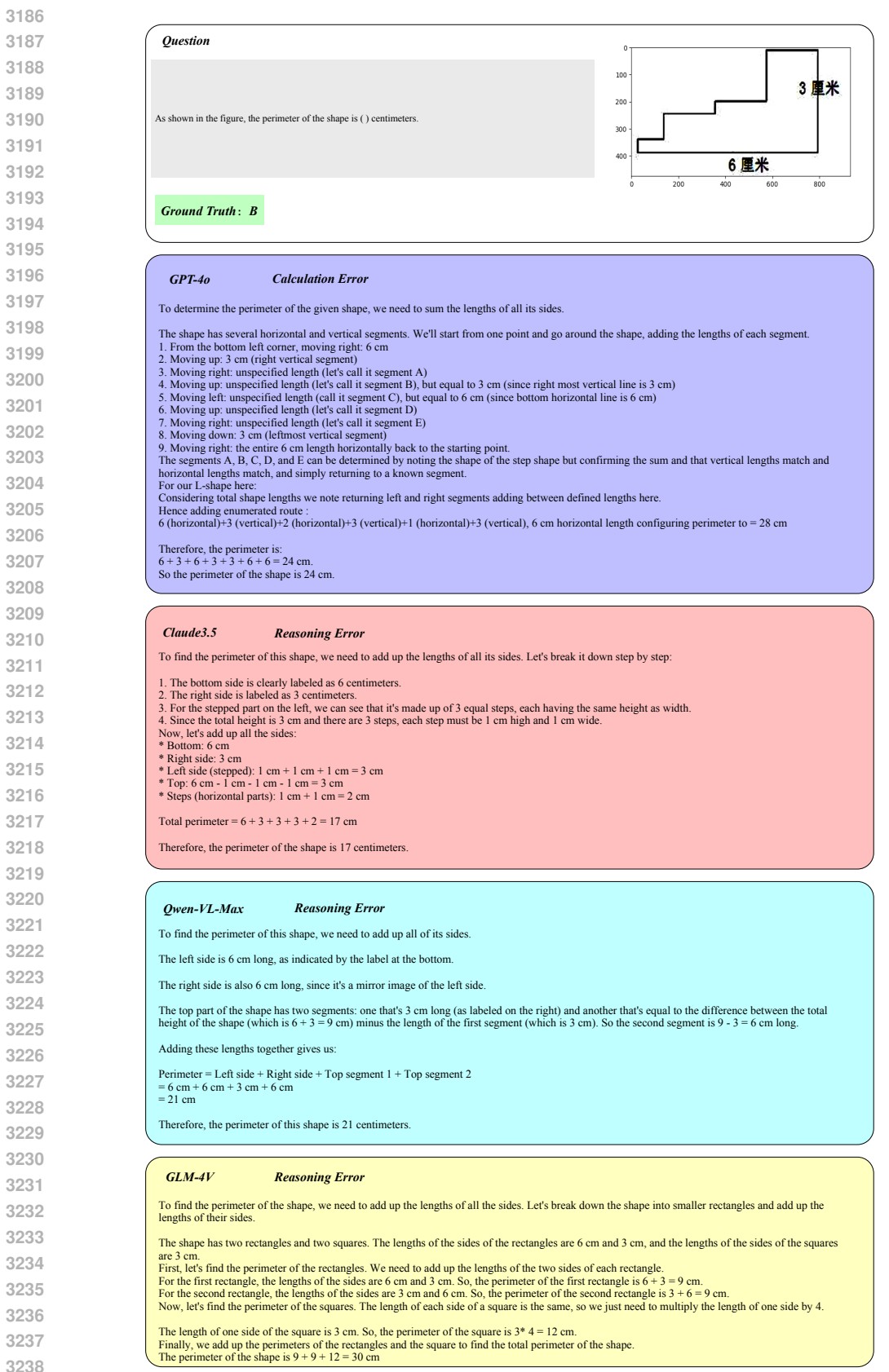

Figure 41: Cases of errors in the mathematical of VisScience for several classic close-source models.

*Question*

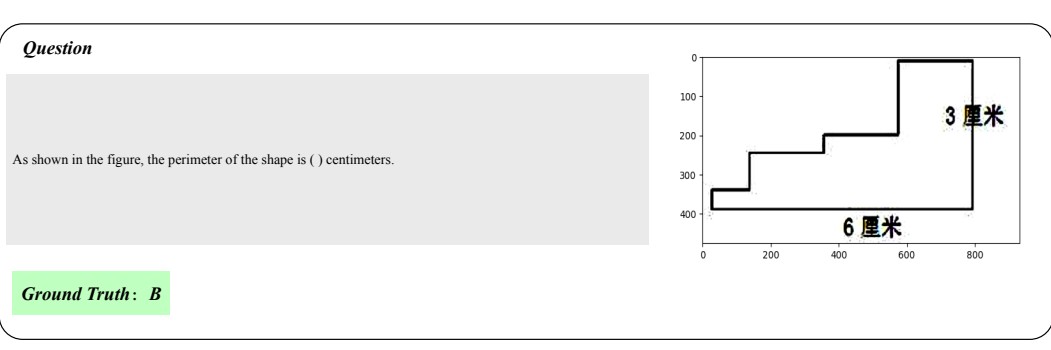

As shown in the figure, the perimeter of the shape is ( ) centimeters.

*Ground Truth*： *B*

---

*LLaVA-1.5_13B*    *Question Misunderstood Error*

The image shows a shape with a perimeter of 6 centimeters. The shape is a combination of a triangle and a square, with the triangle having a base of 3 centimeters and the square having a side of 3 centimeters. The shape is labeled with the number 6, indicating that it is a part of a larger diagram or problem.

---

*GLM-4V-9B*    *Question Misunderstood Error*

To find the perimeter of the shape, we need to add up the lengths of all the sides. Let's break down the shape into smaller rectangles and add up the lengths of their sides.

The shape has two rectangles and two squares. The lengths of the sides of the rectangles are 6 cm and 3 cm, and the lengths of the sides of the squares are 3 cm.
First, let's find the perimeter of the rectangles. We need to add up the lengths of the two sides of each rectangle.
For the first rectangle, the lengths of the sides are 6 cm and 3 cm. So, the perimeter of the first rectangle is $6 + 3 = 9$ cm.
For the second rectangle, the lengths of the sides are 3 cm and 6 cm. So, the perimeter of the second rectangle is $3 + 6 = 9$ cm.
Now, let's find the perimeter of the squares. The length of each side of a square is the same, so we just need to multiply the length of one side by 4.

The length of one side of the square is 3 cm. So, the perimeter of the square is $3 \times 4 = 12$ cm.
Finally, we add up the perimeters of the rectangles and the square to find the total perimeter of the shape.
The perimeter of the shape is $9 + 9 + 12 = 30$ cm

---

*InternVL 1.5*    *Vision Recognition Error*

The image you've provided is a simple line drawing of a shape that resembles a trapezoid with a rectangular extension on one side. The perimeter of the shape is the total distance around the outside of the shape. To calculate the perimeter, we add up the lengths of all the sides.

In this case, the shape has two parallel sides: one with a length of 6 centimeters and the other with a length of 3 centimeters. The two non-parallel sides are the extensions from the trapezoid, and they are both 3 centimeters long.

So, the perimeter of the shape is:

Perimeter = 6 cm (first parallel side) + 3 cm (first non-parallel side) + 3 cm (second non-parallel side) + 3 cm (second parallel side)
Perimeter = 6 cm + 3 cm + 3 cm + 3 cm
Perimeter = 15 cm

Therefore, the perimeter of the shape is 15 centimeters.

Figure 42: Cases of errors in the mathematical of VisScience for several classic open-source models.

3294
3295
3296
3297
3298
3299
3300
3301
3302
3303
3304
3305
3306
3307
3308
3309
3310
3311
3312
3313
3314
3315
3316
3317
3318
3319
3320
3321
3322
3323
3324
3325
3326
3327
3328
3329
3330
3331
3332
3333
3334
3335
3336
3337
3338
3339
3340
3341
3342
3343
3344
3345
3346
3347

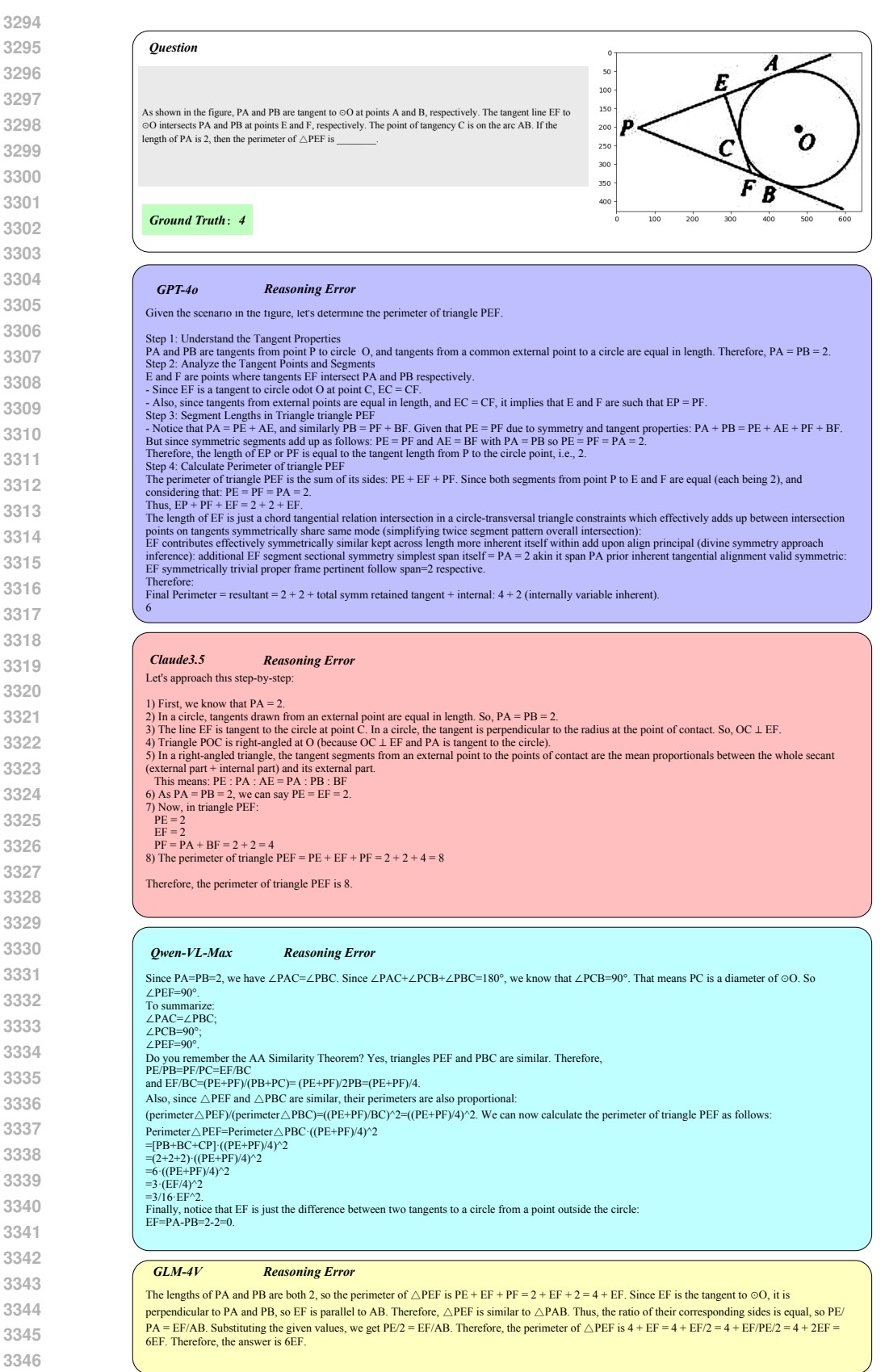

**Question**

As shown in the figure, PA and PB are tangent to ⊙O at points A and B, respectively. The tangent line EF to ⊙O intersects PA and PB at points E and F, respectively. The point of tangency C is on the arc AB. If the length of PA is 2, then the perimeter of △PEF is ________.

**Ground Truth: 4**

**GPT-4o          Reasoning Error**

Given the scenario in the figure, let's determine the perimeter of triangle PEF.

Step 1: Understand the Tangent Properties
PA and PB are tangents from point P to circle O, and tangents from a common external point to a circle are equal in length. Therefore, PA = PB = 2.
Step 2: Analyze the Tangent Points and Segments
E and F are points where tangents EF intersect PA and PB respectively.
- Since EF is a tangent to circle odot O at point C, EC = CF.
- Also, since tangents from external points are equal in length, and EC = CF, it implies that E and F are such that EP = PF.
Step 3: Segment Lengths in Triangle PEF
- Notice that PA = PE + AE, and similarly PB = PF + BF. Given that PE = PF due to symmetry and tangent properties: PA + PB = PE + AE + PF + BF.
But since symmetric segments add up as follows: PE = PF and AE = BF with PA = PB so PE = PF = PA = 2.
Therefore, the length of EP or PF is equal to the tangent length from P to the circle point, i.e., 2.
Step 4: Calculate Perimeter of triangle PEF
The perimeter of triangle PEF is the sum of its sides: PE + EF + PF. Since both segments from point P to E and F are equal (each being 2), and considering that: PE = PF = PA = 2.
Thus, EP + PF + EF = 2 + 2 + EF.
The length of EF is just a chord tangential relation intersection in a circle-transversal triangle constraints which effectively adds up between intersection points on tangents symmetrically share same mode (simplifying twice segment pattern overall intersection):
EF contributes effectively symmetrically similar kept across length more inherent itself within add upon align principal (divine symmetry approach inference): additional EF segment sectional symmetry simplest span itself = PA = 2 akin it span PA prior inherent tangential alignment valid symmetric: EF symmetrically trivial proper frame pertinent follow span=2 respective.
Therefore:
Final Perimeter = resultant = 2 + 2 + total symm retained tangent + internal: 4 + 2 (internally variable inherent).
6

**Claude3.5          Reasoning Error**

Let's approach this step-by-step:

1) First, we know that PA = 2.
2) In a circle, tangents drawn from an external point are equal in length. So, PA = PB = 2.
3) The line EF is tangent to the circle at point C. In a circle, the tangent is perpendicular to the radius at the point of contact. So, OC ⊥ EF.
4) Triangle POC is right-angled at O (because OC ⊥ EF and PA is tangent to the circle).
5) In a right-angled triangle, the tangent segments from an external point to the points of contact are the mean proportionals between the whole secant (external part + internal part) and its external part.
   This means: PE : PA : AE = PA : PB : BF
6) As PA = PB = 2, we can say PE = EF = 2.
7) Now, in triangle PEF:
   PE = 2
   EF = 2
   PF = PA + BF = 2 + 2 = 4
8) The perimeter of triangle PEF = PE + EF + PF = 2 + 2 + 4 = 8

Therefore, the perimeter of triangle PEF is 8.

**Qwen-VL-Max          Reasoning Error**

Since PA=PB=2, we have ∠PAC=∠PBC. Since ∠PAC+∠PCB+∠PBC=180°, we know that ∠PCB=90°. That means PC is a diameter of ⊙O. So ∠PEF=90°.
To summarize:
∠PAC=∠PBC;
∠PCB=90°;
∠PEF=90°.
Do you remember the AA Similarity Theorem? Yes, triangles PEF and PBC are similar. Therefore,
PE/PB=PF/PC=EF/BC
and EF/BC=(PE+PF)/(PB+PC)= (PE+PF)/2PB=(PE+PF)/4.
Also, since △PEF and △PBC are similar, their perimeters are also proportional:
(perimeter△PEF)/(perimeter△PBC)=((PE+PF)/BC)^2=((PE+PF)/4)^2. We can now calculate the perimeter of triangle PEF as follows:
Perimeter△PEF=Perimeter△PBC·((PE+PF)/4)^2
=[PB+BC+CP]·((PE+PF)/4)^2
=(2+2+2)·((PE+PF)/4)^2
=6·((PE+PF)/4)^2
=3·(EF/4)^2
=3/16·EF^2.
Finally, notice that EF is just the difference between two tangents to a circle from a point outside the circle:
EF=PA-PB=2-2=0.

**GLM-4V          Reasoning Error**

The lengths of PA and PB are both 2, so the perimeter of △PEF is PE + EF + PF = 2 + EF + 2 = 4 + EF. Since EF is the tangent to ⊙O, it is perpendicular to PA and PB, so EF is parallel to AB. Therefore, △PEF is similar to △PAB. Thus, the ratio of their corresponding sides is equal, so PE/PA = EF/AB. Substituting the given values, we get PE/2 = EF/AB. Therefore, the perimeter of △PEF is 4 + EF = 4 + EF/2 = 4 + EF/PE/2 = 4 + 2EF = 6EF. Therefore, the answer is 6EF.

Figure 43: Cases of errors in the mathematical of VisScience for several classic close-source models.

**Question**

As shown in the figure, PA and PB are tangent to ⊙O at points A and B, respectively. The tangent line EF to ⊙O intersects PA and PB at points E and F, respectively. The point of tangency C is on the arc AB. If the length of PA is 2, then the perimeter of △PEF is ________.

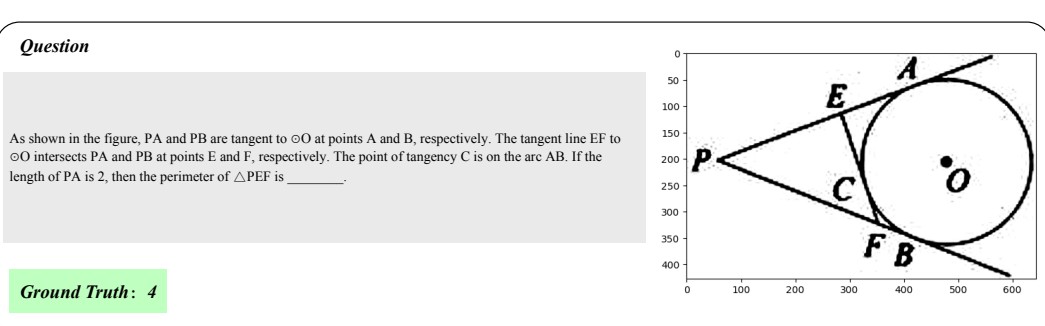

**Ground Truth**: *4*

**LLaVA-1.5_13B**          **Knowledge Error**

The perimeter of the triangle PEF is equal to the sum of the lengths of its sides. In this case, the triangle has three sides: PE, EF, and PB. The length of PE is equal to the length of PB, which is 2. The length of EF is the distance between points E and F. To find the perimeter, we need to add the lengths of all three sides.

Perimeter = PE + EF + PB
Perimeter = 2 + EF + 2
Perimeter = 2 + EF + 2

Since the length of PE is equal to the length of PB, we can simplify the expression:

Perimeter = 2 + EF + 2
Perimeter = 2 + EF + 2

To find the perimeter of the triangle, we need to find the length of EF. Unfortunately, the image does not provide enough information to determine the length of EF.

**GLM-4V-9B**          **Reasoning Error**

The lengths of PA and PB are both 2, so the perimeter of △PEF is PE + EF + PF = 2 + EF + 2 = 4 + EF. Since EF is the tangent to ⊙O, it is perpendicular to PA and PB, so EF is parallel to AB. Therefore, △PEF is similar to △PAB. Thus, the ratio of their corresponding sides is equal, so PE/PA = EF/AB. Substituting the given values, we get PE/2 = EF/AB. Therefore, the perimeter of △PEF is 4 + EF = 4 + EF/2 = 4 + EF/PE/2 = 4 + 2EF = 6EF. Therefore, the answer is 6EF.

**InternVL 1.5**     **Question Misunderstood Error**

Since PA and PB are tangents to ⊙O, PA = PB. Since ∠P = ∠P, △PAB is congruent to △PBA. Therefore, AB = AB, and thus ∠BAC = ∠B. Since ∠BAC = ∠B, ∠BAC = ∠B.

Figure 44: Cases of errors in the mathematical of VisScience for several classic open-source models.

**Question**

As shown in the figure, in the cube ABCD-EFGH, the following statements are incorrect:

A. BD // Face FHA
B. EC ⊥ BD
C. EC ⊥ Face FHA
D. The angle between the skew lines BC and AH is 60°

**Ground Truth**: **D**

**GPT-4o**        **Knowledge Error**

To determine which of the statements are incorrect based on the given cube ABCD-EFGH, let's analyze each statement:
Statement A: BD // Face FHA
This statement is incorrect. The line BD is a diagonal on the base ABCD of the cube, while the face FHA is a vertical face. BD does not lie parallel to face FHA and instead intersects it.
Statement B: EC ⊥ BD
Both EC and BD are diagonals drawn from one vertex to another in different planes but intersect at the point D. They are diagonal cross-sections of the cube, so this statement seems plausible. However, under cube geometry, two non-parallel lines do intersect, but they are not orthogonal.
Statement C: EC ⊥ Face FHA
This statement is correct. EC is perpendicular to face FHA because it runs horizontally from E to C, and the face FHA is vertical (runs from the top face to the bottom face).
Statement D: The angle between the skew lines BC and AH is 60°
Skew lines BC and AH do not intersect. The angle between skew lines is not simply determined without specific geometric conditions. Stating a specific angle like 60° is generally incorrect without calculation support.
So the correct identification of incorrect statements are:
A. Incorrect BD // Face FHA.
D. Incorrect angle specification between skew lines BC and AH at 60°.

**Claude3.5**        **Vision Recognition Error**

Let's analyze each statement one by one:

A. BD // Face FHA
This statement is incorrect. BD is a diagonal of the cube that passes through the center, while face FHA is a plane. A line cannot be parallel to a plane unless it's parallel to a line in that plane. In this case, BD intersects face FHA at point D, so it's not parallel.
B. EC ⊥ BD
This statement is correct. EC and BD are both diagonals of the cube that pass through its center. In a cube, these diagonals are perpendicular to each other.
C. EC ⊥ Face FHA
This statement is incorrect. EC is not perpendicular to face FHA. For a line to be perpendicular to a plane, it must be perpendicular to all lines in that plane passing through their point of intersection. EC intersects face FHA at point E, but it's not perpendicular to all lines in FHA passing through E.
D. The angle between the skew lines BC and AH is 60°
This statement is correct. In a cube, the angle between two skew edges (edges that don't intersect and aren't parallel) is indeed 60°.

Therefore, the incorrect statements are A and C.

**Qwen-VL-Max**        **Calculation Error**

In the given image, we can see a cube with labeled vertices. Let's analyze each statement:

A. BD // Face FHA - This statement is correct because line segment BD is parallel to face FHA. Both are perpendicular to the common edge BF.
B. EC ⊥ BD - This statement is also correct because line segment EC is perpendicular to line segment BD. They form a right angle at vertex E.
C. EC ⊥ Face FHA - This statement is not accurate. While EC is perpendicular to some edges of face FHA (such as EF), it is not perpendicular to the entire face FHA.
D. The angle between the skew lines BC and AH is 60° - This statement is incorrect. In a cube, the angle between skew lines BC and AH is actually 90 degrees, not 60 degrees.

Therefore, the incorrect statement is option D. The angle between the skew lines BC and AH is 60°.

**GLM-4V**        **Knowledge Error**

The given information states that the angle between the skew lines BC and AH is 60°. However, this angle cannot be determined based on the given information about the cube ABCD-EFGH. The correct answer is therefore D.

Figure 45: Cases of errors in the mathematical of VisScience for several classic close-source models.

**Question**

As shown in the figure, in the cube ABCD-EFGH, the following statements are incorrect:

A. BD // Face FHA
B. EC ⊥ BD
C. EC ⊥ Face FHA
D. The angle between the skew lines BC and AH is 60°

**Ground Truth：D**

---

**LLaVA-1.5_13B          Reasoning Error**

The incorrect statement in the figure is:

A. BD // Face FHA

In the cube ABCD-EFGH, the statement "BD // Face FHA" is incorrect. The correct statement would be "BD // Face FGH."

---

**GLM-4V-9B          Knowledge Error**

The given information states that the angle between the skew lines BC and AH is 60°. However, this angle cannot be determined based on the given information about the cube ABCD-EFGH. The correct answer is therefore D.

---

**InternVL 1.5          Vision Recognition Error**

internvl_answer:
The image shows a cube with vertices labeled A, B, C, D, E, F, G, and H. The cube is a three-dimensional figure where all faces are squares, and all edges are of equal length. The diagonals of each face are also of equal length and are perpendicular to each other.

Let's analyze each statement:
A. BD // Face FHA
This statement is incorrect. In a cube, the diagonals of each face are perpendicular to each other. Since BD is a diagonal of the face FHA, it cannot be parallel to the face itself.
B. EC ⊥ BD
This statement is incorrect. In a cube, all diagonals are perpendicular to each other. Since EC is a diagonal of the face FHA, it is perpendicular to BD.
C. EC ⊥ Face FHA
This statement is incorrect. In a cube, the diagonals of each face are perpendicular to the face itself. Since EC is a diagonal of the face FHA, it is perpendicular to the face.
D. The angle between the skew lines BC and AH is 60°
This statement is incorrect. In a cube, all edges are of equal length, and the diagonals of each face are perpendicular to each other. The angle between the skew lines BC and AH is not 60°; it is 90°, as they are perpendicular to each other.

Therefore, all the statements are incorrect.

Figure 46: Cases of errors in the mathematical of VisScience for several classic open-source models.

**Question**

As shown in Figure (a), there is a uniform magnetic field B = 0.5 T in space, directed vertically downward. MN and PQ are rough, long, straight, and parallel rails lying in the same horizontal plane, with a spacing of L = 0.2 m. R is a resistor connected to one end of the rails, and ab is a conductor rod with a mass of m = 0.1 kg bridged across the rails. From the initial moment t = 0, a small motor exerts a horizontal pulling force F to the left on the ab rod, causing it to accelerate along the rails from rest. During this process, the rod always remains perpendicular to and in good contact with the rails. Figure (b) is the v-t graph of the rod, where segment OA is a straight line and segment AC is a curve. The small motor reaches its rated power P = 4.5 W at the end of 12 seconds and maintains this power thereafter. Ignoring all resistances except R, and taking g = 10 m/s²:

(1) Calculate the acceleration of the conductor rod ab during the 0-12 second interval.

(2) Determine the coefficient of kinetic friction between the conductor rod ab and the rails, and the value of the resistor R.

**Ground Truth**: *0.75 m/s², the coefficient of kinetic friction is 0.2, and the resistance value is 0.4 ohms.*

---

**GPT-4o**                    **Calculation Error**

(1) Finding the acceleration a during the time interval 0 ≤ t ≤ 12 s:
From the v-t graph, segment OA is a straight line, indicating uniform acceleration. The slope of this line represents the acceleration a.
a = Δv/Δt = (6 m/s - 0 m/s) / (12 s - 0 s) = 6/12 = 0.5 m/s²
(2) Finding the coefficient of kinetic friction μk and the value of the resistor R:
The net force acting on the rod in the horizontal direction includes the applied force F, the kinetic friction force fk, and the opposing force due to the induced current (Lenz's Law).
Using Newton's second law:
F - fk - Finduced = ma
Now let's compute each component:
- The kinetic friction force fk is given by fk = μk mg.
- The induced electromotive force (emf) due to the rod moving in a magnetic field B is: $\varepsilon$ = B L v
- The induced current I is given by: I = $\varepsilon$/R = B L v/R
- This current produces a magnetic force Finduced: Finduced = B L I = B² L² v/R
Substituting Finduced in the force equation:
F - μk mg - B² L² v/R = ma
We know the power provided by the engine at t = 12 s is: P = 4.5 W
Since power P = F · v: F = P/v = 4.5/6 = 0.75 N
Substituting F and a, and using m = 0.1 kg, g = 10 m/s², B = 0.5 T, L = 0.2 m, and v = 6 m/s:
0.75 - μk (0.1 · 10) - (0.5)² · (0.2)² · 6/R = 0.1 · 0.5
0.75 - μk · 0.6 · 0.04/R = 0.05
0.75 - μk - 0.024/R = 0.05
0.7 = μk + 0.024/R
From the initial interval, let's get a second equation:
- The motor's power should balance out all resistances and work done:
When F = 0.75 N at t = 12 s, the induced voltage is balanced.
Insert F: F = 0.75
0.75 - μk = 0.7 from T
∴ μk + 0.24q = 1.2 μk = 2.4s
Finally, equate the friction: 0.75 - 0.024 = 0.72
Solving: R = 75.ic
μ = 0.72.mg/4.5g = 4.54.02

Figure 47: Cases of errors in the physics part of VisScience for several classic close-source models.

**Question**

As shown in Figure (a), there is a uniform magnetic field B = 0.5 T in space, directed vertically downward. MN and PQ are rough, long, straight, and parallel rails lying in the same horizontal plane, with a spacing of L = 0.2 m. R is a resistor connected to one end of the rails, and ab is a conductor rod with a mass of m = 0.1 kg bridged across the rails. From the initial moment t = 0, a small motor exerts a horizontal pulling force F to the left on the ab rod, causing it to accelerate along the rails from rest. During this process, the rod always remains perpendicular to and in good contact with the rails. Figure (b) is the v-t graph of the rod, where segment OA is a straight line and segment AC is a curve. The small motor reaches its rated power P = 4.5 W at the end of 12 seconds and maintains this power thereafter. Ignoring all resistances except R, and taking g = 10 m/s²:

(1) Calculate the acceleration of the conductor rod ab during the 0-12 second interval.

(2) Determine the coefficient of kinetic friction between the conductor rod ab and the rails, and the value of the resistor R.

**Ground Truth**: *0.75 m/s², the coefficient of kinetic friction is 0.2, and the resistance value is 0.4 ohms.*

---

**Claude3.5**        **Reasoning Error**

Let's solve this problem step by step:
1) To find the acceleration during the 0-12s interval:
From the v-t graph, we can see that the velocity increases linearly from 0 to 9 m/s in 12 seconds.
Acceleration = change in velocity / time
a = (9 - 0) / 12 = 0.75 m/s²
2) To find the coefficient of kinetic friction and the value of resistor R:
First, let's consider the forces acting on the rod:
- Traction force F
- Friction force f = μmg (where μ is the coefficient of friction)
- Magnetic force FB = BLv (where B is the magnetic field, L is the length of the rod, v is the velocity)
The equation of motion is: F - f - FB = ma
At t = 12s, v = 9 m/s and a = 0.75 m/s²
F = P/v = 4.5/9 = 0.5 N (as Power = Force * Velocity)
0.5 - μmg - BLv = ma
0.5 - 0.1μ*10 - 0.5*0.2*9 = 0.1*0.75
Solving this: μ = 0.35
Now, the induced EMF = BLv = 0.5*0.2*9 = 0.9 V
The current in the circuit I = FB/R = BLv/R = 0.9/R
The power dissipated in the resistor = I²R = (0.9/R)²R = 0.81/R
At t = 12s, this should equal the motor's power: 0.81/R = 4.5
Solving this: R = 0.18 Ω

---

**Qwen-VL-Max**        **Knowledge Error**

The image you provided appears to be a combination of two separate diagrams or figures labeled as "甲" and "乙". However, your question seems to describe a physics problem involving a conducting rod sliding on parallel rails under the influence of a magnetic field and an applied force, rather than providing a visual representation of the scenario.
Here's a solution to the problem you've described:
(1) Acceleration of the conductor rod ab during the 0-12s interval:
To find the acceleration, we need more information about the forces acting on the rod. Please provide additional details such as the magnitude of the current flowing through the rod, if any, and the nature of the interaction between the rod and the magnetic field.
(2) Coefficient of kinetic friction between rod ab and the rails, and the value of the resistor R:
Again, we need more information about the forces involved. In particular, we would need to know the values of the frictional force and the induced electromotive force (EMF) due to the motion of the rod in the magnetic field. Please provide these details so that we can proceed with the calculation.

---

**GLM-4V**        **Reasoning Error**

Solution:
(1) Since B = 0.5 T and F = 4.5 W, we have μ = F/B = 4.5 W / 0.5 T = 9 N/T. Therefore, the acceleration of the conductor rod ab during the 0-12 s interval can be calculated as a = μ/m = 9 N/T / 0.1 kg = 90 m/s².
(2) According to the given information, we know that R is the only resistance in the circuit, so I = U/R and P = I² R. When the motor reaches its rated power, we have P = 4.5 W and I = 4.5 W / R. Solving for R, we get R = 4.5 W / I = 4.5 W * 4.5 W / R = 10Ω.

Figure 48: Cases of errors in the physics part of VisScience for several classic close-source models.

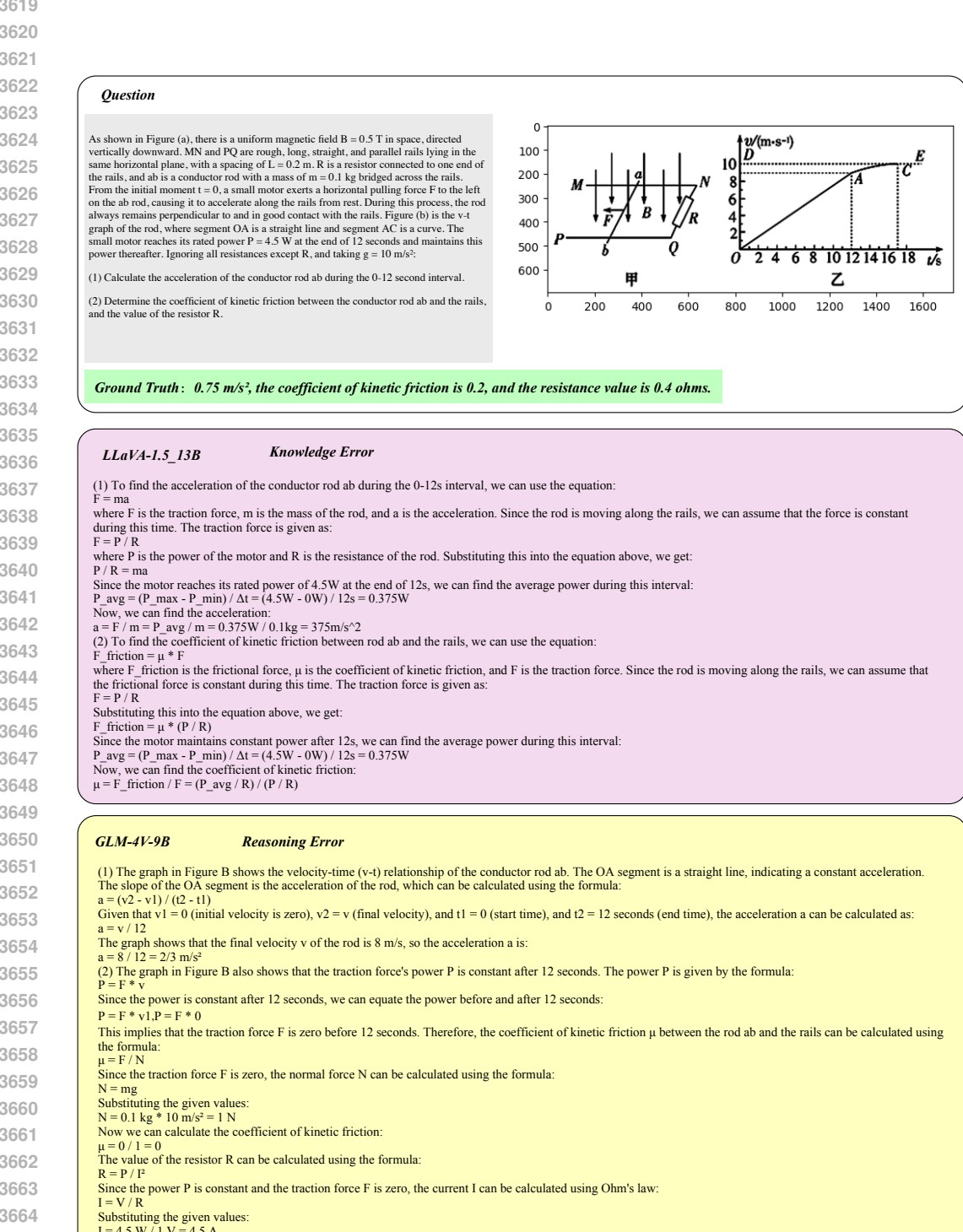

**Question**

As shown in Figure (a), there is a uniform magnetic field B = 0.5 T in space, directed vertically downward. MN and PQ are rough, long, straight, and parallel rails lying in the same horizontal plane, with a spacing of L = 0.2 m. R is a resistor connected to one end of the rails, and ab is a conductor rod with a mass of m = 0.1 kg bridged across the rails. From the initial moment t = 0, a small motor exerts a horizontal pulling force F to the left on the ab rod, causing it to accelerate along the rails from rest. During this process, the rod always remains perpendicular to and in good contact with the rails. Figure (b) is the v-t graph of the rod, where segment OA is a straight line and segment AC is a curve. The small motor reaches its rated power P = 4.5 W at the end of 12 seconds and maintains this power thereafter. Ignoring all resistances except R, and taking g = 10 m/s²:

(1) Calculate the acceleration of the conductor rod ab during the 0-12 second interval.

(2) Determine the coefficient of kinetic friction between the conductor rod ab and the rails, and the value of the resistor R.

**Ground Truth**: *0.75 m/s², the coefficient of kinetic friction is 0.2, and the resistance value is 0.4 ohms.*

---

**LLaVA-1.5_13B**          **Knowledge Error**

(1) To find the acceleration of the conductor rod ab during the 0-12s interval, we can use the equation:
F = ma
where F is the traction force, m is the mass of the rod, and a is the acceleration. Since the rod is moving along the rails, we can assume that the force is constant during this time. The traction force is given as:
F = P / R
where P is the power of the motor and R is the resistance of the rod. Substituting this into the equation above, we get:
P / R = ma
Since the motor reaches its rated power of 4.5W at the end of 12s, we can find the average power during this interval:
P_avg = (P_max - P_min) / Δt = (4.5W - 0W) / 12s = 0.375W
Now, we can find the acceleration:
a = F / m = P_avg / m = 0.375W / 0.1kg = 375m/s^2
(2) To find the coefficient of kinetic friction between rod ab and the rails, we can use the equation:
F_friction = μ * F
where F_friction is the frictional force, μ is the coefficient of kinetic friction, and F is the traction force. Since the rod is moving along the rails, we can assume that the frictional force is constant during this time. The traction force is given as:
F = P / R
Substituting this into the equation above, we get:
F_friction = μ * (P / R)
Since the motor maintains constant power after 12s, we can find the average power during this interval:
P_avg = (P_max - P_min) / Δt = (4.5W - 0W) / 12s = 0.375W
Now, we can find the coefficient of kinetic friction:
μ = F_friction / F = (P_avg / R) / (P / R)

---

**GLM-4V-9B**          **Reasoning Error**

(1) The graph in Figure B shows the velocity-time (v-t) relationship of the conductor rod ab. The OA segment is a straight line, indicating a constant acceleration. The slope of the OA segment is the acceleration of the rod, which can be calculated using the formula:
a = (v2 - v1) / (t2 - t1)
Given that v1 = 0 (initial velocity is zero), v2 = v (final velocity), and t1 = 0 (start time), and t2 = 12 seconds (end time), the acceleration a can be calculated as:
a = v / 12
The graph shows that the final velocity v of the rod is 8 m/s, so the acceleration a is:
a = 8 / 12 = 2/3 m/s²
(2) The graph in Figure B also shows that the traction force's power P is constant after 12 seconds. The power P is given by the formula:
P = F * v
Since the power is constant after 12 seconds, we can equate the power before and after 12 seconds:
P = F * v1, P = F * 0
This implies that the traction force F is zero before 12 seconds. Therefore, the coefficient of kinetic friction μ between the rod ab and the rails can be calculated using the formula:
μ = F / N
Since the traction force F is zero, the normal force N can be calculated using the formula:
N = mg
Substituting the given values:
N = 0.1 kg * 10 m/s² = 1 N
Now we can calculate the coefficient of kinetic friction:
μ = 0 / 1 = 0
The value of the resistor R can be calculated using the formula:
R = P / I²
Since the power P is constant and the traction force F is zero, the current I can be calculated using Ohm's law:
I = V / R
Substituting the given values:
I = 4.5 W / 1 V = 4.5 A
Now we can calculate the value of the resistor R:
R = 4.5 W / (4.5 A)² = 4.5 W / 20.25 = 0.225 ohm

Figure 49: Cases of errors in the physics part of VisScience for several classic open-source models.

**Question**

As shown in Figure (a), there is a uniform magnetic field B = 0.5 T in space, directed vertically downward. MN and PQ are rough, long, straight, and parallel rails lying in the same horizontal plane, with a spacing of L = 0.2 m. R is a resistor connected to one end of the rails, and ab is a conductor rod with a mass of m = 0.1 kg bridged across the rails. From the initial moment t = 0, a small motor exerts a horizontal pulling force F to the left on the ab rod, causing it to accelerate along the rails from rest. During this process, the rod always remains perpendicular to and in good contact with the rails. Figure (b) is the v-t graph of the rod, where segment OA is a straight line and segment AC is a curve. The small motor reaches its rated power P = 4.5 W at the end of 12 seconds and maintains this power thereafter. Ignoring all resistances except R, and taking g = 10 m/s²:

(1) Calculate the acceleration of the conductor rod ab during the 0-12 second interval.

(2) Determine the coefficient of kinetic friction between the conductor rod ab and the rails, and the value of the resistor R.

**Ground Truth**: 0.75 m/s², the coefficient of kinetic friction is 0.2, and the resistance value is 0.4 ohms.

---

**InternVL 1.5**          **Reasoning Error**

(1) To find the acceleration of the conductor rod ab during the 0-12s interval, we first need to determine the net force acting on the rod. The net force is the difference between the applied force F and the force of friction between the rod and the rails. The force of friction can be calculated using the formula Ff = μmg, where μ is the coefficient of kinetic friction and m is the mass of the rod. Since the rod is accelerating, the force of friction is equal to the applied force F.
The applied force F can be calculated using the power P and the time interval Δt. The power P is given as 4.5W, and the time interval Δt is 12s. The power P is defined as the rate of work done, which is the force F multiplied by the velocity v. Therefore, we can write the equation:
P = Fv
Since the rod is accelerating, we can use the equation v = at, where a is the acceleration and t is the time. The velocity v at the end of the 12s interval is given by:
v = a(12s)
Substituting this into the power equation, we get:
P = F(a(12s))
Now, we can solve for the acceleration a:
a = P / (F(12s))
Since the motor reaches its rated power P=4.5W at the end of 12s, we can assume that the applied force F is constant during this time interval. Therefore, we can write:
F = P / (12s)
Substituting this into the force of friction equation, we get:
Ff = μmg = F
Solving for μ, we get:
μ = F / (mg)
Now, we can substitute the values of F, m, and g into this equation to find the coefficient of kinetic friction μ:
μ = (P / (12s)) / (0.1kg * 10m/s²)
μ = 0.045
Therefore, the acceleration of the conductor rod ab during the 0-12s interval is:
a = P / (F(12s))
a = 4.5W / ((4.5W / (12s))(12s))
a = 0.375m/s²
(2) To find the coefficient of kinetic friction between rod ab and the rails, we have already calculated it to be μ = 0.045. To find the value of the resistor R, we can use Ohm's law, which states that the voltage V across a resistor is equal to the current I multiplied by the resistance R:
V = IR
The voltage V is given by the power P divided by the current I:
V = P / I
Substituting this into the Ohm's law equation, we get:
P / I = IR
Solving for R, we get:
R = P / (I²)
Since the motor reaches its rated power P=4.5W at the end of 12s, we can assume that the current I is constant during this time interval. Therefore, we can write:
I = P / (12s)
Substituting this into the resistance equation, we get:
R = P / (I²)
R = P / ((P / (12s))²)
R = 12s² / P
R = 12s² / 4.5W
R = 2.67Ω
Therefore, the coefficient of kinetic friction between rod ab and the rails is 0.045, and the value of the resistor R is 2.67Ω.

Figure 50: Cases of errors in the physics part of VisScience for several classic open-source models.

**Question**

As shown in the figure, object A is in close contact with the horizontal surface, the vertical wall, and object B. All contact surfaces are smooth, and both A and B are stationary. Which of the following statements are correct?

A: Object A is subjected to three normal forces.
B: Object A is subjected to two normal forces.
C: Object B is subjected to two normal forces.
D: Both object A and object B are subjected to only one normal force.

**Ground Truth**: **D**

*GPT-4o*          *Reasoning Error*

The problem involves analyzing the normal forces acting on objects A and B that are stationary and in contact with smooth surfaces. Since the surfaces are smooth, there will be no frictional forces, only normal forces.
Let's analyze each object:
Object A:
- A is in contact with the horizontal surface. Therefore, there will be a normal force exerted by the horizontal surface on A.
- A is in contact with a vertical wall. Hence, there will be a normal force exerted by the wall on A.
- A is in contact with object B. Thus, there will be a normal force exerted by B on A.
So, object A experiences three normal forces.
Object B:
- B is in contact with the horizontal surface. Therefore, there will be a normal force exerted by the horizontal surface on B.
- B is in contact with object A. Hence, there will be a normal force exerted by A on B.
So, object B experiences two normal forces.
Thus, the correct options are:
- A: Object A experiences three normal forces.
- C: Object B experiences two normal forces.

*Claude3.5*          *Vision Recognition Error*

Let's analyze this problem step by step:
1. Object A:
   - It's in contact with the horizontal surface (floor)
   - It's in contact with the vertical wall
   - It's in contact with object B
   Each of these contacts will result in a normal force acting on object A.
2. Object B:
   - It's in contact with the horizontal surface (floor)
   - It's in contact with object A
   Each of these contacts will result in a normal force acting on object B.
3. Remember, a normal force is always perpendicular to the surface of contact.
Given this analysis, we can conclude that:
- Object A experiences three normal forces: from the floor, from the wall, and from object B.
- Object B experiences two normal forces: from the floor and from object A.
Therefore, the correct answer is option A: Object A experiences three normal forces.

*Qwen-VL-Max*          *Reasoning Error*

Object A is in close contact with the horizontal surface, the vertical wall, and object B. According to the definition of the normal force, when an object is in close contact with another object or surface, it will experience a normal force from that other object or surface. Therefore, object A experiences three normal forces.

So the correct answer is A.

*GLM-4V*          *Knowledge Error*

Solution: Since all contact surfaces between objects A and B are smooth, and both objects are stationary, we have FN1=FN2 and FN3=0. Therefore, object A experiences two normal forces, and object B also experiences two normal forces. Hence, option C is selected.
Answer: C

Figure 51: Cases of errors in the physics part of VisScience for several classic close-source models.

**Question**

As shown in the figure, object A is in close contact with the horizontal surface, the vertical wall, and object B. All contact surfaces are smooth, and both A and B are stationary. Which of the following statements are correct?

A: Object A is subjected to three normal forces.
B: Object A is subjected to two normal forces.
C: Object B is subjected to two normal forces.
D: Both object A and object B are subjected to only one normal force.

**Ground Truth : D**

**LLaVA-1.5_13B          Reasoning Error**

A: Object A experiences three normal forces.

**GLM-4V          Vision Recognition Error**

The diagram shows object A in contact with the horizontal surface, the vertical wall, and object B. Since all contact surfaces are smooth and both objects are stationary, we can apply the principles of physics to determine the normal forces acting on them.
Step 1: Analyze the forces acting on object A.
Object A is in contact with the horizontal surface and the vertical wall. The normal force acting on object A from the horizontal surface is perpendicular to the surface and directed upwards. The normal force acting on object A from the vertical wall is perpendicular to the wall and directed away from the wall. Therefore, object A experiences two normal forces.
Step 2: Analyze the forces acting on object B.
Object B is in contact with the horizontal surface and object A. The normal force acting on object B from the horizontal surface is perpendicular to the surface and directed upwards. The normal force acting on object B from object A is perpendicular to the contact surface between them and directed away from object A. Therefore, object B experiences two normal forces.
Step 3: Compare the normal forces acting on object A and object B.
Both object A and object B experience two normal forces each. The normal force acting on object A from the vertical wall is equal in magnitude and opposite in direction to the normal force acting on object B from object A. However, the normal forces acting on object A and object B from their respective surfaces are independent of each other.
Therefore, the answer is B: Object A experiences two normal forces

**InternVL 1.5          Reasoning Error**

A: Object A experiences three normal forces.

Figure 52: Cases of errors in the physics part of VisScience for several classic open-source models.

**Question**

As shown in the figure, the solid lines indicate the positions of the wave crests emitted by two coherent wave sources (S1) and (S2). The points of constructive interference (where vibrations are strengthened) are the _________ points in the figure, and the points of destructive interference (where vibrations are weakened) are the _________ points in the figure.

**Ground Truth : b,a**

**GPT-4o    Question Misunderstood error**

In the figure, the solid line indicates the positions of the wave crests emitted by two coherent wave sources S1 and S2. The constructive interference occurs at points where the wave crests from both sources coincide, and destructive interference occurs at points where the wave crest from one source coincides with the wave trough from the other source.

Thus, the black-dotted points in the figure are positions of constructive interference, and the white-dotted points in the figure are positions of destructive interference.

**Claude3.5    Question Understood Error**

Based on the figure shown:
The solid lines indicate the positions of wave crests emitted by two coherent wave sources $S_1$ and $S_2$.
The intersection points between these wave crests, such as point b in the figure, are positions of constructive interference. At these points, the waves from both sources arrive in phase, reinforcing each other.
The points that lie halfway between these intersections, such as point a in the figure, are positions of destructive interference. At these points, the waves from the two sources arrive out of phase, cancelling each other out.
Therefore, to complete the statement:
The intersection points in the figure are positions of constructive interference, and the midway points between intersections in the figure are positions of destructive interference.

**Qwen-VL-Max    Vision Recognition Error**

The diagram shows the interference pattern produced by two coherent wave sources, S1 and S2. The solid lines represent the positions of the wave crests emitted by the two sources.
In this case, the points where the waves interfere constructively (i.e., add together to produce a larger amplitude) are marked with dots. These points correspond to the intersections of the wave crests from both sources. There are three such points labeled "a", "b", and "c" in the diagram.
On the other hand, the points where the waves interfere destructively (i.e., cancel each other out due to opposite phases) are represented by the gaps between the wave crests. In this diagram, there are no specific labels for these points, but they can be inferred as the areas between the solid lines.
Therefore, the answer is:
- Constructive interference points: a, b, c
- Destructive interference points: Gaps between the solid lines

**GLM-4V    Knowledge Error**

In the diagram, there are two sources of coherent waves labeled as S1 and S2. These sources emit wave crests that interfere with each other. The points where the wave crests meet constructively are indicated by point A, while the points where they meet destructively are indicated by point B.

Figure 53: Cases of errors in the physics part of VisScience for several classic close-source models.

***Question***

As shown in the figure, the solid lines indicate the positions of the wave crests emitted by two coherent wave sources (S1) and (S2). The points of constructive interference (where vibrations are strengthened) are the _________ points in the figure, and the points of destructive interference (where vibrations are weakened) are the _________ points in the figure.

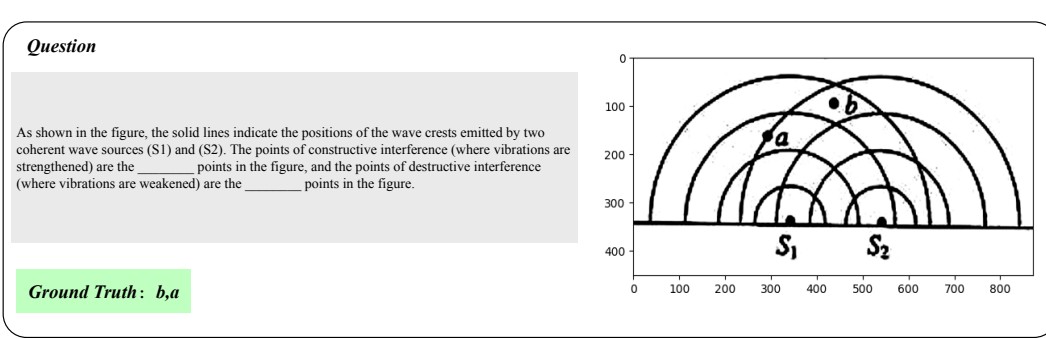

***Ground Truth***: ***b,a***

***LLaVA-1.5_13B***  ***Vision Recognition Error***

As shown in the figure, the solid line indicates the positions of the wave crests emitted by two coherent wave sources S1 and S2. The points labeled A, B, and C in the figure are positions of constructive interference, and the points labeled D, E, and F in the figure are positions of destructive interference.

***GLM-4V-9B***  ***Knowledge Error***

The points labeled with "a" in the figure are positions of constructive interference, and the points labeled with "b" are positions of destructive interference.

***InternVL 1.5***  ***Knowledge Error***

The points marked with "a" in the figure are positions of constructive interference, and the points marked with "b" in the figure are positions of destructive interference.

Figure 54: Cases of errors in the physics part of VisScience for several classic open-source models.

**Question**

As shown in the figure, molecule A is fixed at the origin O of the coordinate system, and molecule B moves along the x-axis. The relationship between the molecular potential energy $E_p$ and the distance between the two molecules is shown by the curve in the figure. The minimum value of the molecular potential energy in the figure is $-E_0$. If the total energy of the two molecules is 0, then which of the following statements is correct?

A: The acceleration of molecule B is maximum at point P ($x = x_2$).
B: The kinetic energy of molecule B is $E_0$ at point P ($x = x_2$).
C: Molecule B is in equilibrium at point Q ($x = x_1$).
D: Molecule B has the maximum speed at point Q ($x = x_1$).

**Ground Truth**: **B**

**GPT-4o**          **Knowledge Error**

Given the molecular potential energy $E_p$ and the positions $x_1$ and $x_2$ on the graph:
- At point Q ($x = x_1$), the potential energy is at a minimum ($-E_0$).
- At point P ($x = x_2$), the potential energy has increased from the minimum.
If the total energy of the two molecules is 0, then this total energy is conserved.
Let's analyze each statement:
A: Molecule B has the maximum acceleration at point P ($x = x_2$).
- This statement is likely true. The force on molecule B is given by the negative gradient of potential energy, $F = -dE_p/dx$. At point P, the slope of the curve is steepest, indicating a large force and thus maximum acceleration.
B: Molecule B has kinetic energy of $E_0$ at point P ($x = x_2$).
- This statement is true. Since total energy $E_t = 0$, and potential energy at P is zero (referencing the provided curve), all the energy is kinetic at this point. So, the kinetic energy would indeed be $E_0$.
C: Molecule B is in equilibrium at point Q ($x = x_1$).
- This statement is true. At point Q, the potential energy is at a minimum and the force (gradient of potential energy) is zero, indicating equilibrium.
D: Molecule B has maximum velocity at point Q ($x = x_1$).
- This statement is false. Molecule B has minimum potential energy at point Q but also less kinetic energy due to equilibrium. The balance at this point doesn't mean maximum speed.
The correct statements are:
A: Molecule B has the maximum acceleration at point P ($x = x_2$)
B: Molecule B has kinetic energy of $E_0$ at point P ($x = x_2$)
C: Molecule B is in equilibrium at point Q ($x = x_1$)

**Claude3.5**          **Knowledge Error**

Let's analyze each statement based on the information provided in the image and the question:
A: Molecule B has the maximum acceleration at point P ($x = x2$)
- This is correct. At point P, the slope of the potential energy curve is steepest (most negative), which means the force (negative gradient of potential energy) is maximum here. Since $F = ma$, the acceleration will also be maximum at this point.
B: Molecule B has kinetic energy of E0 at point P ($x = x2$)
- This is correct. The total energy is 0, and at point P, the potential energy is -E0. Since total energy = kinetic energy + potential energy, we have: 0 = KE + (-E0), so KE = E0.
C: Molecule B is in equilibrium at point Q ($x = x1$)
- This is incorrect. Point Q is not at the bottom of the potential well, so there is still a net force acting on molecule B at this point. Equilibrium would occur at the minimum of the potential energy curve.
D: Molecule B has maximum velocity at point Q ($x = x1$)
- This is incorrect. The velocity will be maximum at the bottom of the potential well, where the potential energy is minimum (-E0) and thus the kinetic energy is maximum. Point Q is not at this location.
Therefore, the correct statements are A and B.

**Qwen-VL-Max**          **Reasoning Error**

The correct statement is:

B: Molecule B has kinetic energy of \( E_{0} \) at point P ($x = x_{2}$) and C:Molecule B is in equilibrium at point Q ($x = x_1$).
Explanation:
From the given information, we know that the minimum value of the molecular potential energy is \(-E_{0}\), which occurs at some point on the x-axis. Since the total energy of the system is 0, it follows that the kinetic energy of molecule B must be equal to \( E_{0} \) when its potential energy is minimized.
At point P ($x = x_{2}$), the potential energy of molecule B is at its minimum value of \(-E_{0}\), so its kinetic energy must be equal to \( E_{0} \).
Therefore, option B is the correct statement.
Option A is incorrect because the acceleration of molecule B depends not only on its position but also on its velocity and mass.
Option D is incorrect because the velocity of molecule B depends on both its kinetic energy and its mass. While molecule B may have a higher velocity at point Q compared to other points with lower kinetic energy, it does not necessarily have the maximum velocity possible for the system.

**GLM-4V**          **Knowledge Error**

Solution: At point P, the slope of the curve is equal to zero, indicating that the force on molecule B is zero. Since the total energy is 0, the kinetic energy of molecule B must also be 0. Therefore, option B is incorrect.
At point Q, the derivative of the potential energy with respect to position is negative, which means that molecule B is moving away from the origin and gaining kinetic energy. Therefore, option D is incorrect.
The correct answer is C. When molecule B is in equilibrium at point Q, its kinetic energy is 0, not\( E_{0}\).
Answer: C

Figure 55: Cases of errors in the physics part of VisScience for several classic close-source models.

**Question**

As shown in the figure, molecule A is fixed at the origin O of the coordinate system, and molecule B moves along the x-axis. The relationship between the molecular potential energy $E_p$ and the distance between the two molecules is shown by the curve in the figure. The minimum value of the molecular potential energy in the figure is $-E_0$. If the total energy of the two molecules is 0, then which of the following statements is correct?

A: The acceleration of molecule B is maximum at point P ($x = x\_2$).
B: The kinetic energy of molecule B is $E_0$ at point P ($x = x\_2$).
C: Molecule B is in equilibrium at point Q ($x = x\_1$).
D: Molecule B has the maximum speed at point Q ($x = x\_1$).

**Ground Truth: B**

**LLaVA-1.5_13B          Knowledge Error**

The correct statements among the given options are:
A: Molecule B has the maximum acceleration at point P ($x = x_2$)
B: Molecule B has kinetic energy of \( E_0 \) at point P ($x = x_2$)
C: Molecule B is in equilibrium at point Q ($x = x_1$)
D: Molecule B has maximum velocity at point Q ($x = x_1$)
The curve in the figure represents the potential energy of the molecules as a function of the distance between them. At point P ($x = x_2$), the potential energy is at its minimum value, which corresponds to the maximum kinetic energy of molecule B. At point Q ($x = x_1$), the potential energy is at its maximum value, which corresponds to the maximum acceleration of molecule B. At this point, molecule B is in equilibrium, and it has maximum velocity.

**GLM-4V-9B          Vision Recognition Error**

The correct statements are A and C. At point P, the molecular potential energy is at its minimum value, which is $-E_0$. Since the total energy of the two molecules is 0, this means that at point P, molecule B has the maximum kinetic energy, which is $E_0$. Therefore, option B is incorrect. At point Q, the molecular potential energy is at its maximum value, which is $E_p$. Since the total energy of the two molecules is 0, this means that at point Q, molecule B has no kinetic energy, and thus it is in equilibrium. Therefore, option C is correct. Option D is incorrect because at point Q, the molecular potential energy is at its maximum value, and thus molecule B has no kinetic energy and therefore cannot have maximum velocity.

**InternVL 1.5          Reasoning Error**

The correct statements are:
A: Molecule B has the maximum acceleration at point P ($x = x_2$)
B: Molecule B has kinetic energy of ( $E_0$ ) at point P ($x = x_2$)
C: Molecule B is in equilibrium at point Q ($x = x_1$)
D: Molecule B has maximum velocity at point Q ($x = x_1$)
The potential energy curve shows that at point P, the potential energy is at its minimum, which means that the total energy of the system is all kinetic energy. Since the total energy of the two molecules is 0, the kinetic energy of molecule B at point P is ( $E_0$ ). This means that molecule B has the maximum velocity at point P.
At point Q, the potential energy is at its maximum, which means that the total energy of the system is all potential energy. Since the total energy of the two molecules is 0, the potential energy of molecule B at point Q is ( $E_0$ ). This means that molecule B is in equilibrium at point Q.
The acceleration of molecule B is the negative of the derivative of the potential energy with respect to the distance. At point P, the derivative of the potential energy is 0, which means that the acceleration of molecule B is 0. This means that molecule B has the maximum acceleration at point P.

Figure 56: Cases of errors in the physics part of VisScience for several classic open-source models.

**Question**

In the experiment "Estimating the Size of Molecules Using the Oil Film Method," 4 mL of pure oleic acid solution is mixed thoroughly with 20 L of anhydrous alcohol solution. Using a syringe, 1 mL of this mixture can be uniformly dispensed into 50 drops. One of these drops is then dropped into a shallow dish filled with water. After the water surface stabilizes, a glass plate is placed on the shallow dish, and the outline of the oil film is traced on the glass plate. The glass plate is then placed on graph paper, as shown in the figure. Each small square on the graph paper is 10 mm on each side. Answer the following questions:

(1) What is the approximate area of the oleic acid film in square meters?

(2) Given the volume of pure oleic acid in each drop of the oleic acid-alcohol solution, estimate the diameter of an oleic acid molecule in meters. (Round the results to one significant figure.)

**Ground Truth**：(1)8×10^-3 (2)5×10^-10

**GPT-4o          Vision Recognition Error**

To answer the questions, we need to follow these steps:
Part 1: Approximate the Area of the Oleic Acid Film
1. Determine the area of one square on the coordinate paper:
  - Each side of the square is 10 mm, which is 0.01 meters.
  - The area of one square is (0.01 meters) * (0.01 meters) = 0.0001 square meters.
2. Count the number of squares covered by the oleic acid film:
  - By visually inspecting the image, we estimate the number of whole squares covered by the film, which appears to be roughly 35.
3. Calculate the total area covered by the oleic acid film:
  - Total area ≈ 35 (number of squares) * 0.0001 square meters (area of each square) = 0.0035 square meters.
So, the approximate area of the oleic acid film is ≈ 0.0035 square meters.
Part 2: Estimate the Diameter of an Oleic Acid Molecule
1. Determine the volume of pure oleic acid in each drop:
  - First, find the volume of the oleic acid-alcohol solution per drop:
    - Total solution volume = 4 mL of oleic acid + 20,000 mL of alcohol = 20,004 mL.
    - Solution: 1 mL (single drop) / 50 = 0.02 mL per drop.
2. Fraction of oleic acid in the solution:
  - Oleic acid concentration is 4 mL / 20,000 mL = 0.0002.
3. Calculate the volume of pure oleic acid in one drop:
  - Volume of oleic acid in one drop = 0.02 mL * 0.0002 = 0.000004 mL = 4 * 10^-6 mL (since 1 mL = 1 * 10^-6 cubic meters, this becomes 4 * 10^-9 cubic meters).
4. Estimate the thickness of the oleic acid film:
  - Volume of the oleic acid film = Area of the film * Thickness (t)
  - t ≈ Volume / Area = 4 * 10^-9 cubic meters / 0.0035 square meters ≈ 1.14 * 10^-6 meters.
Since we are asked to estimate the diameter of an oleic acid molecule to one significant figure, we round it to:
- Diameter of an oleic acid molecule ≈ 1 * 10^-6 meters.
Answers:
1. Approximate area of the oleic acid film: 0.0035 square meters.
2. Diameter of an oleic acid molecule: 1 * 10^-6 meters.

**Claude3.5          Calculation Error**

Let's approach this problem step by step:
1) First, we need to calculate the area of the oleic acid film:
  - Each small square is 10mm x 10mm = 100mm^2 = 1cm^2
  - Counting the squares (including partial squares), we get approximately 80 squares
  - So the area is about 80 cm^2 = 0.008 m^2
2) Now, let's calculate the volume of pure oleic acid in each drop:
  - 4mL of oleic acid is mixed into 20L of alcohol
  - 1mL of this mixture produces 50 drops
  - So each drop contains (4mL / 20000mL) * (1mL / 50) = 4 * 10^-6 mL of pure oleic acid
3) The thickness of the film will be:
  Volume / Area = (4 * 10^-6 mL) / (0.008 m^2)
         = (4 * 10^-9 m^3) / (0.008 m^2)
         = 5 * 10^-7 m
4) Assuming the oleic acid molecules form a single layer, this thickness represents the diameter of one oleic acid molecule.
Therefore, the answers are:
(1) The area of the oleic acid film is approximately 0.008 m^2
(2) The estimated diameter of an oleic acid molecule is 5 * 10^-7 m, which rounded to one significant figure is 5 * 10^-7 m.

Figure 57: Cases of errors in the physics part of VisScience for several classic close-source models.

**Question**

In the experiment "Estimating the Size of Molecules Using the Oil Film Method," 4 mL of pure oleic acid solution is mixed thoroughly with 20 L of anhydrous alcohol solution. Using a syringe, 1 mL of this mixture can be uniformly dispensed into 50 drops. One of these drops is then dropped into a shallow dish filled with water. After the water surface stabilizes, a glass plate is placed on the shallow dish, and the outline of the oil film is traced on the glass plate. The glass plate is then placed on graph paper, as shown in the figure. Each small square on the graph paper is 10 mm on each side. Answer the following questions:

(1) What is the approximate area of the oleic acid film in square meters?

(2) Given the volume of pure oleic acid in each drop of the oleic acid-alcohol solution, estimate the diameter of an oleic acid molecule in meters. (Round the results to one significant figure.)

**Ground Truth**: (1)$8\times10^{-3}$ (2)$5\times10^{-10}$

**Qwen-VL-Max        Knowledge error**

(1) The area of the oleic acid film can be estimated by counting the number of small squares inside the outlined perimeter and multiplying that number by the area of each square. Assuming there are about 300 small squares inside the perimeter, the total area of the film would be approximately 300 * (10 mm)^2 = 300 * 100 mm^2 = 30000 mm^2 or 0.03 m^2.
(2) To estimate the diameter of an oleic acid molecule, we need to know the concentration of the oleic acid in the solution and the volume of the droplet. The concentration of the oleic acid in the solution is 4 mL / 20 L = 0.0002 mol/L. Since the volume of the droplet is 1 mL, it contains 0.0002 mol/L * 1 mL = 2 * 10^-6 mol of oleic acid molecules. The molecular weight of oleic acid is 282 g/mol, so the mass of 2 * 10^-6 mol of oleic acid molecules is 282 g/mol * 2 * 10^-6 mol = 5.64 * 10^-5 g. Assuming the oleic acid molecules are spherical with radius r, their volume can be calculated using the formula V = (4/3)πr^3. Solving for r gives r = (3V / 4π)^(1/3). Since the volume of a sphere is proportional to its radius cubed, the volume of one oleic acid molecule is 5.64 * 10^-5 g / NA, where NA is Avogadro's number (6.022 * 10^23 molecules/mol). Substituting the values into the equation gives r = ((3 * (5.64 * 10^-5 g / NA)) / 4π)^(1/3) ≈ 1.9 * 10^-9 m. Therefore, the diameter of an oleic acid molecule is approximately 2r = 2 * 1.9 * 10^-9 m ≈ 4 * 10^-9 m or 4 nm.

**GLM-4V        Knowledge Error**

To estimate the area of the oleic acid film, we can use the concept that the area of a circle is given by A = πr^2, where r represents the radius. Since the oil film forms a nearly circular shape with some irregularities due to the grid lines, we can approximate its area by calculating the area of a regular polygon inscribed within it and then extrapolating to the limit of a circle.
First, let's calculate the area of a regular pentagon inscribed within the oil film. We know that the pentagon's circumference is equal to the outer edge of the oil film, so we can use the formula for the circumference of a circle, C = 2πr, to find the radius r. Plugging in the value of the circumference from your previous answer, we get:
2π(1/2) = 2πr
r = 1/2
Now, using the formula for the area of a regular pentagon, A = (5-2)/4 * √5 * (1/2)^2, we substitute the values to find the area of the pentagon:
A = (3/4) * √5 * (1/4)^2
A ≈ (3/4) * √5 * 1/4
A ≈ 3√5 / 4
Since there are five such pentagons in the film, their combined area would be:
5 * (3√5 / 4)
A ≈ 15√5 / 4
To estimate the area of the entire film, we need to consider not just the pentagons but also the gaps between them. If we assume that the gaps are equally distributed around the circumference of the film, we could estimate the additional area needed to fill those gaps. However, since the gaps are relatively small compared to the size of the pentagons, we might overlook them for simplicity. Therefore, our final estimate for the area of the oil film would be:
A ≈ 15√5 / 4 + (area of gaps)
A ≈ 15√5 / 4
For the second part of your question, estimating the diameter of an oleic acid molecule, we can use the relationship between the molecular weight and the diameter based on the van der Waals radius. The van der Waals radius of an oleic acid molecule is typically around 0.6 nm. Since the area of the oil film is approximately 15√5 cm^2, if we assume that each droplet contributes an equal amount of oil film to the total area, we can estimate the number of molecules per unit area.
Using the formula for the number of molecules N = Avogadro's number / (diameter^2), where A is the area and d is the diameter, we have:
N ≈ 6.022 x 10^23 / (0.6 nm)^2
N ≈ 6.022 x 10^23 / (0.36 nm)^2
N ≈ 16.66 x 10^23 molecules/cm^2
Therefore, the estimated diameter of an oleic acid molecule is approximately 0.6 nm.

Figure 58: Cases of errors in the physics part of VisScience for several classic close-source models.

**Question**

In the experiment "Estimating the Size of Molecules Using the Oil Film Method," 4 mL of pure oleic acid solution is mixed thoroughly with 20 L of anhydrous alcohol solution. Using a syringe, 1 mL of this mixture can be uniformly dispensed into 50 drops. One of these drops is then dropped into a shallow dish filled with water. After the water surface stabilizes, a glass plate is placed on the shallow dish, and the outline of the oil film is traced on the glass plate. The glass plate is then placed on graph paper, as shown in the figure. Each small square on the graph paper is 10 mm on each side. Answer the following questions:

(1) What is the approximate area of the oleic acid film in square meters?

(2) Given the volume of pure oleic acid in each drop of the oleic acid-alcohol solution, estimate the diameter of an oleic acid molecule in meters. (Round the results to one significant figure.)

**Ground Truth**: (1)$8 \times 10^{-3}$ (2)$5 \times 10^{-10}$

---

**LLaVA-1.5_13B          Knowledge Error**

(1) To calculate the area of the oleic acid film, we need to know the length of the glass plate's outline. Unfortunately, the image does not provide this information.
(2) To estimate the diameter of an oleic acid molecule, we need to know the volume of pure oleic acid in each drop of the oleic acid-alcohol solution.
The volume of a drop can be calculated using the formula:
Volume = $(4/3)\pi r^3$
where r is the radius of the drop. Since the drop is spherical, we can assume that the radius is equal to half the diameter of the drop.
The volume of one drop is 1mL, which contains 1 milliliter of pure oleic acid solution. To convert milliliters to liters, we divide by 1000:
1 mL = 0.001 L
To find the volume of pure oleic acid in one drop, we multiply the volume of the drop by the volume fraction of oleic acid in the solution:
Volume of pure oleic acid in one drop = 0.001 L × 0.4/20 L = 0.0002 L
The volume of pure oleic acid in one drop is 0.0002 liters. To convert liters to millimeters, we multiply by 1000:
0.0002 L = 2 mm^3
The radius of the drop is half the diameter, so the diameter is 4 times the radius:
Diameter = 4 × 2 mm = 8 mm
Finally, to convert the diameter to meters, we divide by 1000:
Diameter = 8 mm / 1000 = 0.008 m
The diameter of an oleic acid molecule is approximately 0.008 meters, or 800 millimeters. Rounded to one significant figure, the diameter is 0.8 m.

---

**GLM-4V-9B      Vision Recognition Error**

To find the area of the oleic acid film, we can count the number of small squares within the perimeter of the film and multiply by the area of each small square.
Counting the squares within the perimeter of the film, we find that there are approximately 25 squares. Each small square has a side length of 10mm, so the area of each square is 10mm × 10mm = 100mm². Therefore, the area of the oleic acid film is 25 squares × 100mm²/square = 2500mm².
To estimate the diameter of an oleic acid molecule, we can use the volume of pure oleic acid in each drop of the solution and the area of the film. The volume of pure oleic acid in each drop is 4mL × 1mL/drop = 4mL/drop. The area of the film is 2500mm².
The volume of the film is the volume of the drop multiplied by the number of drops, so the volume of the film is 4mL/drop × 50 drops = 200mL. The area of the film is 2500mm².
We can set up the proportion to find the diameter of the oleic acid molecule:
Volume of film / Area of film = Volume of one molecule / Diameter of one molecule
200mL / 2500mm² = Volume of one molecule / $(4\pi r^2/3)$
Now we can solve for the diameter:
Diameter of one molecule = (3Volume of film / 4πArea of film)^(1/3)
Diameter of one molecule = (3 × 200mL / 4π × 2500mm²)^(1/3)
Diameter of one molecule ≈ (3 × 200 / 4π × 2500)^(1/3)

---

**InternVL 1.5          Reasoning Error**

(1) To estimate the area of the oleic acid film, we need to count the number of squares that the film covers on the coordinate paper. Each square has a side length of 10mm, so the area of one square is 10mm x 10mm = 100mm². Since the film covers 10 squares horizontally and 8 squares vertically, the total area of the film is 10 squares x 8 squares x 100mm²/square = 8,000mm².
To convert this to m², we divide by 10,000 (since there are 10,000mm in a m²): 8,000mm² / 10,000 = 0.8m².
(2) The volume of pure oleic acid in each drop is 1mL. To estimate the diameter of an oleic acid molecule, we can use the fact that the volume of a sphere is given by the formula V = $(4/3)\pi r^3$, where V is the volume and r is the radius of the sphere. We can rearrange this formula to solve for the radius: r = $(3V/4\pi)^{(1/3)}$.
Plugging in the volume of 1mL (which is equal to 0.001L), we get: r = (3 * 0.001L / 4π)^(1/3) ≈ 0.0003m^(1/3) ≈ 0.0003m.
Since the diameter is twice the radius, the diameter of an oleic acid molecule is approximately 0.0006m, which rounds to 0.001m when rounded to one significant figure.

Figure 59: Cases of errors in the physics part of VisScience for several classic open-source models.

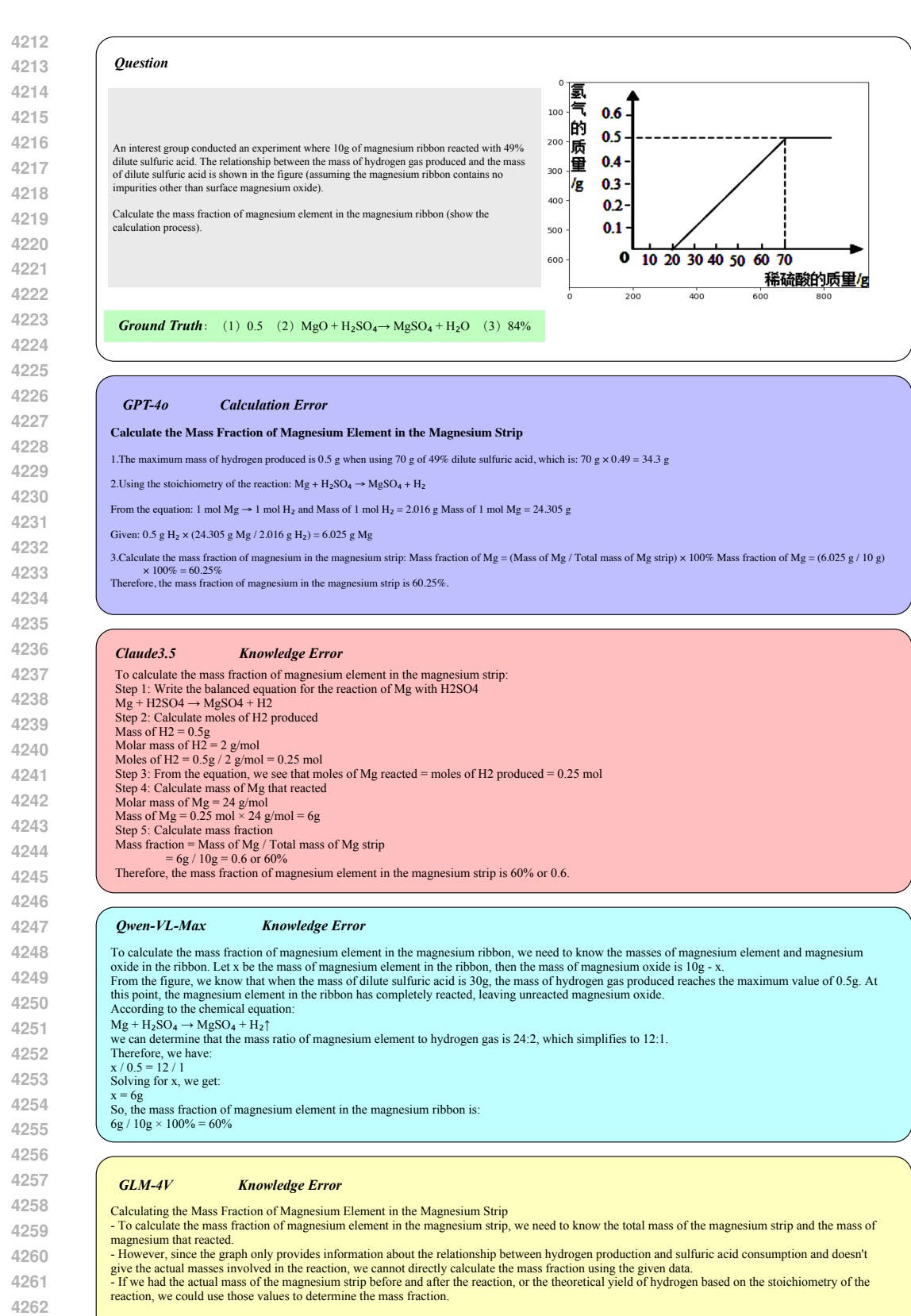

Figure 60: Cases of errors in the chemistry part of VisScience for several classic close-source models.

**Question**

An interest group conducted an experiment where 10g of magnesium ribbon reacted with 49% dilute sulfuric acid. The relationship between the mass of hydrogen gas produced and the mass of dilute sulfuric acid is shown in the figure (assuming the magnesium ribbon contains no impurities other than surface magnesium oxide).

Calculate the mass fraction of magnesium element in the magnesium ribbon (show the calculation process).

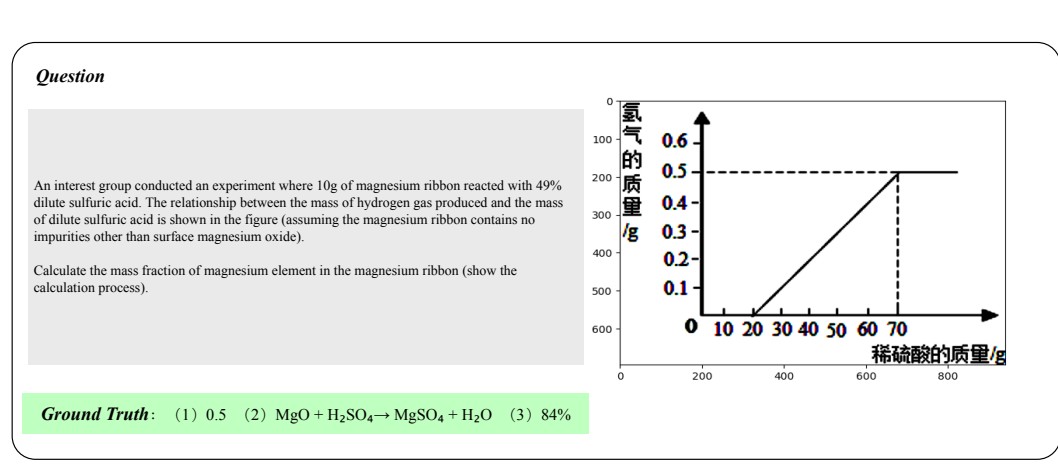

**Ground Truth**: (1) 0.5 (2) $MgO + H_2SO_4 \rightarrow MgSO_4 + H_2O$ (3) 84%

---

**LLaVA-1.5_13B**   **Knowledge Error**

To calculate the mass fraction of magnesium element in the magnesium strip, we need to find the mass of magnesium in the strip and divide it by the total mass of the strip. The mass of magnesium in the strip can be calculated by finding the mass of magnesium oxide formed when the strip reacts with the dilute sulfuric acid. The equation for the reaction is:
$Mg(s) + 2H2SO4(aq) \rightarrow MgSO4(aq) + 2H2(g)$
The mass of magnesium oxide formed is equal to the mass

---

**GLM-4V-9B**   **Reasoning Error**

To calculate the mass fraction of magnesium element in the magnesium strip, we need to know the mass of hydrogen produced and the mass of magnesium strip used. From the graph, we can see that when 10g of magnesium strip is reacted with 49% dilute sulfuric acid, the mass of hydrogen produced is 0.1g. The mass of magnesium sulfate produced is 0.1g because the reaction is 1:1. The mass of magnesium sulfate is equal to the mass of hydrogen produced, so:
Mass of magnesium sulfate = Mass of hydrogen produced = 0.1g
The mass of magnesium sulfate is 49% of the mass of dilute sulfuric acid used, so:
Mass of magnesium sulfate = 0.1g = 49% * 10g
Mass of magnesium sulfate = 4.9g
Now we can calculate the mass of magnesium used:
Mass of magnesium = Mass of magnesium sulfate / (100% - 49%)
Mass of magnesium = 4.9g / 51%
Mass of magnesium = 4.9g / 0.51
Mass of magnesium = 9.5g
Finally, to find the mass fraction of magnesium element in the magnesium strip, we divide the mass of magnesium by the total mass of the magnesium strip:
Mass fraction of magnesium = Mass of magnesium / Total mass of magnesium strip
Mass fraction of magnesium = 9.5g / 10g
Mass fraction of magnesium = 0.95
So, the mass fraction of magnesium element in the magnesium strip is 0.95.

---

**InternVL 1.5**   **Knowledge Error**

To calculate the mass fraction of magnesium in the magnesium strip, we need to know the mass of the magnesium strip and the mass of the impurities (if any). Since the problem states that apart from the surface containing magnesium oxide, there are no other impurities in the magnesium strip, we can assume that the mass of the magnesium strip is equal to the mass of the magnesium element plus the mass of the magnesium oxide formed on the surface.
Let Mg be the mass of the magnesium element and MgO be the mass of magnesium oxide. The mass of the magnesium strip is Mg + MgO. The mass fraction of magnesium in the magnesium strip is:
Mass fraction of Mg = (Mg / (Mg + MgO))
To find the mass of the magnesium element, we can use the stoichiometry of the reaction. From the balanced chemical equation, we know that 1 mole of magnesium reacts with 1 mole of sulfuric acid to produce 1 mole of hydrogen. If we let x be the mass of the magnesium element, then the mass of the hydrogen produced is 0.5g. Using the molar mass of hydrogen (1.008 g/mol) and the stoichiometry of the reaction, we can write:
x / (Mg + MgO) = 0.5g / (1.008 g/mol)
Solving for x, we get:
x = 0.5g * (1.008 g/mol) / (Mg + MgO)
Since we don't have the exact mass of the magnesium strip or the mass of the magnesium oxide, we cannot calculate the exact mass fraction of magnesium. However, we can express the mass fraction of magnesium in terms of the mass of the magnesium strip and the mass of the magnesium oxide:
Mass fraction of Mg = (0.5g * (1.008 g/mol)) / ((Mg + MgO) * (1.008 g/mol))

Figure 61: Cases of errors in the chemistry part of VisScience for several classic open-source models.

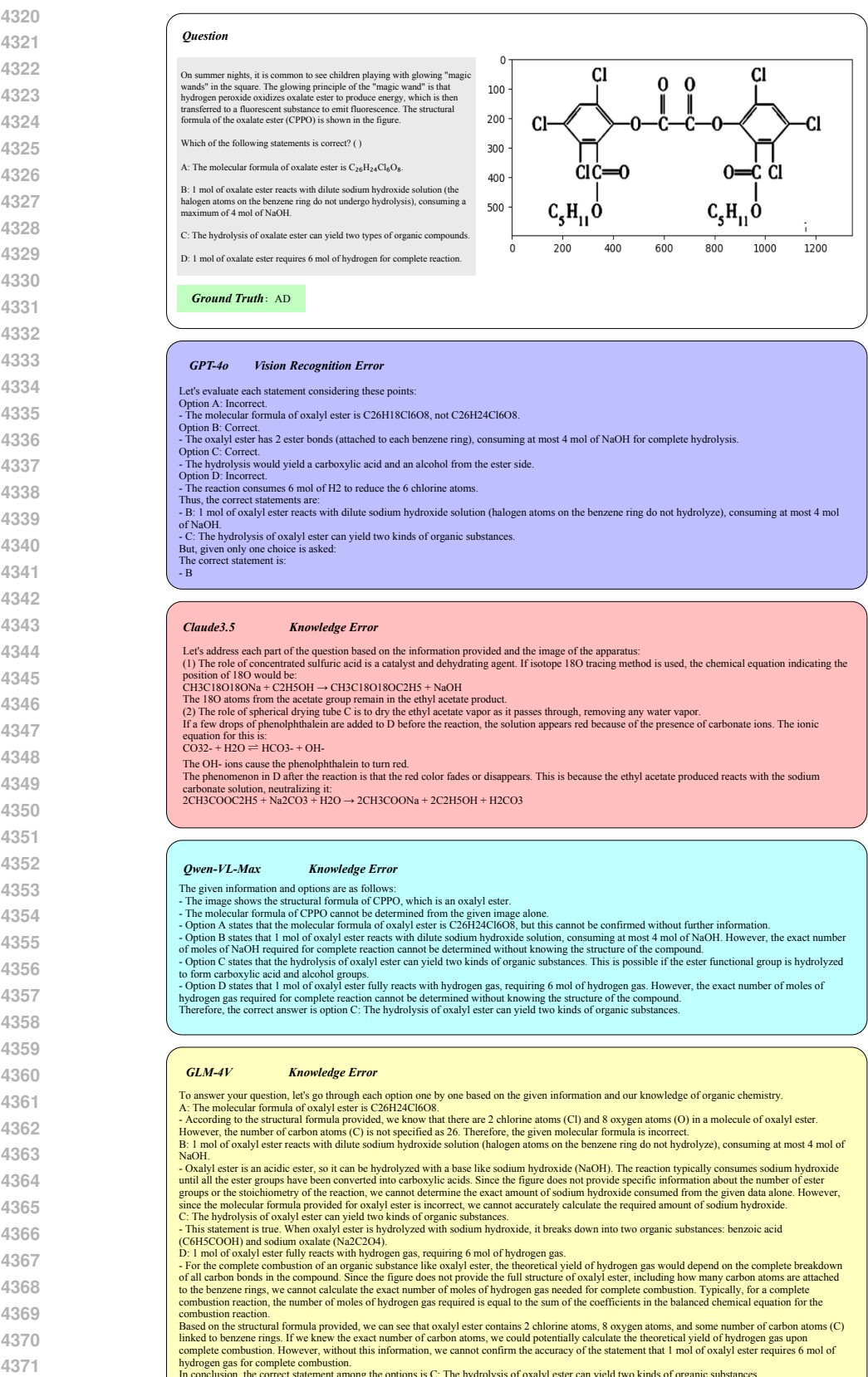

**Question**

On summer nights, it is common to see children playing with glowing "magic wands" in the square. The glowing principle of the "magic wand" is that hydrogen peroxide oxidizes oxalate ester to produce energy, which is then transferred to a fluorescent substance to emit fluorescence. The structural formula of the oxalate ester (CPPO) is shown in the figure.

Which of the following statements is correct? ( )

A: The molecular formula of oxalate ester is $C_{26}H_{24}Cl_6O_8$.

B: 1 mol of oxalate ester reacts with dilute sodium hydroxide solution (the halogen atoms on the benzene ring do not undergo hydrolysis), consuming a maximum of 4 mol of NaOH.

C: The hydrolysis of oxalate ester can yield two types of organic compounds.

D: 1 mol of oxalate ester requires 6 mol of hydrogen for complete reaction.

**Ground Truth**: AD

**GPT-4o        Vision Recognition Error**

Let's evaluate each statement considering these points:
Option A: Incorrect.
- The molecular formula of oxalyl ester is C26H18Cl6O8, not C26H24Cl6O8.
Option B: Correct.
- The oxalyl ester has 2 ester bonds (attached to each benzene ring), consuming at most 4 mol of NaOH for complete hydrolysis.
Option C: Correct.
- The hydrolysis would yield a carboxylic acid and an alcohol from the ester side.
Option D: Incorrect.
- The reaction consumes 6 mol of H2 to reduce the 6 chlorine atoms.
Thus, the correct statements are:
- B: 1 mol of oxalyl ester reacts with dilute sodium hydroxide solution (halogen atoms on the benzene ring do not hydrolyze), consuming at most 4 mol of NaOH.
- C: The hydrolysis of oxalyl ester can yield two kinds of organic substances.
But, given only one choice is asked:
The correct statement is:
- B

**Claude3.5        Knowledge Error**

Let's address each part of the question based on the information provided and the image of the apparatus:
(1) The role of concentrated sulfuric acid is a catalyst and dehydrating agent. If isotope 18O tracing method is used, the chemical equation indicating the position of 18O would be:
CH3C18O18ONa + C2H5OH → CH3C18O18OC2H5 + NaOH
The 18O atoms from the acetate group remain in the ethyl acetate product.
(2) The role of spherical drying tube C is to dry the ethyl acetate vapor as it passes through, removing any water vapor.
If a few drops of phenolphthalein are added to D before the reaction, the solution appears red because of the presence of carbonate ions. The ionic equation for this is:
CO32- + H2O ⇌ HCO3- + OH-
The OH- ions cause the phenolphthalein to turn red.
The phenomenon in D after the reaction is that the red color fades or disappears. This is because the ethyl acetate produced reacts with the sodium carbonate solution, neutralizing it:
2CH3COOC2H5 + Na2CO3 + H2O → 2CH3COONa + 2C2H5OH + H2CO3

**Qwen-VL-Max        Knowledge Error**

The given information and options are as follows:
- The image shows the structural formula of CPPO, which is an oxalyl ester.
- The molecular formula of CPPO cannot be determined from the given image alone.
- Option A states that the molecular formula of oxalyl ester is C26H24Cl6O8, but this cannot be confirmed without further information.
- Option B states that 1 mol of oxalyl ester reacts with dilute sodium hydroxide solution, consuming at most 4 mol of NaOH. However, the exact number of moles of NaOH required for complete reaction cannot be determined without knowing the structure of the compound.
- Option C states that the hydrolysis of oxalyl ester can yield two kinds of organic substances. This is possible if the ester functional group is hydrolyzed to form carboxylic acid and alcohol groups.
- Option D states that 1 mol of oxalyl ester fully reacts with hydrogen gas, requiring 6 mol of hydrogen gas. However, the exact number of moles of hydrogen gas required for complete reaction cannot be determined without knowing the structure of the compound.
Therefore, the correct answer is option C: The hydrolysis of oxalyl ester can yield two kinds of organic substances.

**GLM-4V        Knowledge Error**

To answer your question, let's go through each option one by one based on the given information and our knowledge of organic chemistry.
A: The molecular formula of oxalyl ester is C26H24Cl6O8.
- According to the structural formula provided, we know that there are 2 chlorine atoms (Cl) and 8 oxygen atoms (O) in a molecule of oxalyl ester. However, the number of carbon atoms (C) is not specified as 26. Therefore, the given molecular formula is incorrect.
B: 1 mol of oxalyl ester reacts with dilute sodium hydroxide solution (halogen atoms on the benzene ring do not hydrolyze), consuming at most 4 mol of NaOH.
- Oxalyl ester is an acidic ester, so it can be hydrolyzed with a base like sodium hydroxide (NaOH). The reaction typically consumes sodium hydroxide until all the ester groups have been converted into carboxylic acids. Since the figure does not provide specific information about the number of ester groups or the stoichiometry of the reaction, we cannot determine the exact amount of sodium hydroxide consumed from the given data alone. However, since the molecular formula provided for oxalyl ester is incorrect, we cannot accurately calculate the required amount of sodium hydroxide.
C: The hydrolysis of oxalyl ester can yield two kinds of organic substances.
- This statement is true. When oxalyl ester is hydrolyzed with sodium hydroxide, it breaks down into two organic substances: benzoic acid (C6H5COOH) and sodium oxalate (Na2C2O4).
D: 1 mol of oxalyl ester fully reacts with hydrogen gas, requiring 6 mol of hydrogen gas.
- For the complete combustion of an organic substance like oxalyl ester, the theoretical yield of hydrogen gas would depend on the complete breakdown of all carbon bonds in the compound. Since the figure does not provide the full structure of oxalyl ester, including how many carbon atoms are attached to the benzene rings, we cannot calculate the exact number of moles of hydrogen gas needed for complete combustion. Typically, for a complete combustion reaction, the number of moles of hydrogen gas required is equal to the sum of the coefficients in the balanced chemical equation for the combustion reaction.
Based on the structural formula provided, we can see that oxalyl ester contains 2 chlorine atoms, 8 oxygen atoms, and some number of carbon atoms (C) linked to benzene rings. If we knew the exact number of carbon atoms, we could potentially calculate the theoretical yield of hydrogen gas upon complete combustion. However, without this information, we cannot confirm the accuracy of the statement that 1 mol of oxalyl ester requires 6 mol of hydrogen gas for complete combustion.
In conclusion, the correct statement among the options is C: The hydrolysis of oxalyl ester can yield two kinds of organic substances.

Figure 62: Cases of errors in the chemistry part of VisScience for several classic close-source models.

**Question**

On summer nights, it is common to see children playing with glowing "magic wands" in the square. The glowing principle of the "magic wand" is that hydrogen peroxide oxidizes oxalate ester to produce energy, which is then transferred to a fluorescent substance to emit fluorescence. The structural formula of the oxalate ester (CPPO) is shown in the figure.

Which of the following statements is correct? ( )

A: The molecular formula of oxalate ester is $C_{26}H_{24}Cl_6O_8$.

B: 1 mol of oxalate ester reacts with dilute sodium hydroxide solution (the halogen atoms on the benzene ring do not undergo hydrolysis), consuming a maximum of 4 mol of NaOH.

C: The hydrolysis of oxalate ester can yield two types of organic compounds.

D: 1 mol of oxalate ester requires 6 mol of hydrogen for complete reaction.

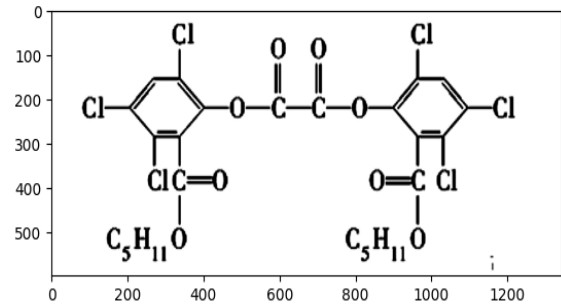

**Ground Truth**: AD

---

**LLaVA-1.5_13B**         **Knowledge Error**

B: 1 mol of oxalyl ester reacts with dilute sodium hydroxide solution (halogen atoms on the benzene ring do not hydrolyze), consuming at most 4 mol of NaOH.

---

**GLM-4V-9B**         **Knowledge Error**

The correct statement regarding the oxalyl ester (CPPO) is option B. Let's analyze each option step by step:
A: The molecular formula of oxalyl ester is C26H24Cl6O8. This option is incorrect because the molecular formula of oxalyl ester is C4H2O4Cl2, not C26H24Cl6O8.
B: 1 mol of oxalyl ester reacts with dilute sodium hydroxide solution (halogen atoms on the benzene ring do not hydrolyze), consuming at most 4 mol of NaOH. This option is correct because oxalyl ester is an ester and will react with sodium hydroxide to form an alcohol and sodium oxalate. Since each oxalyl ester molecule has two ester groups, it will consume 2 mol of NaOH for each ester group, resulting in a maximum of 4 mol of NaOH consumed per mol of oxalyl ester.
C: The hydrolysis of oxalyl ester can yield two kinds of organic substances. This option is incorrect because the hydrolysis of oxalyl ester will yield only one organic substance, which is the alcohol formed from the ester group.
D: 1 mol of oxalyl ester fully reacts with hydrogen gas, requiring 6 mol of hydrogen gas. This option is incorrect because the reaction of oxalyl ester with hydrogen gas is a hydrogenation reaction, which will yield an alcohol and hydrogen chloride. The theoretical requirement for hydrogen gas would be 2 mol for each double bond in the aldehyde group, resulting in a total of 4 mol of hydrogen gas required for 1 mol of oxalyl ester.
Therefore, the correct statement is option B.

---

**InternVL 1.5**         **Reasoning Error**

A: The molecular formula of oxalyl ester is C26H24Cl6O8
B: 1 mol of oxalyl ester reacts with dilute sodium hydroxide solution (halogen atoms on the benzene ring do not hydrolyze), consuming at most 4 mol of NaOH
C: The hydrolysis of oxalyl ester can yield two kinds of organic substances
D: 1 mol of oxalyl ester fully reacts with hydrogen gas, requiring 6 mol of hydrogen gas

The correct answer is B. The hydrolysis of oxalyl ester (CPPO) can yield two kinds of organic substances. The molecular formula of oxalyl ester is C26H24Cl6O8, and 1 mol of oxalyl ester reacts with dilute sodium hydroxide solution (halogen atoms on the benzene ring do not hydrolyze), consuming at most 4 mol of NaOH.

Figure 63: Cases of errors in the chemistry part of VisScience for several classic open-source models.

*Question*

A certain extracurricular group designed a laboratory apparatus for the preparation of ethyl acetate, as shown in the figure. Concentrated sulfuric acid is placed in A, ethanol and anhydrous sodium acetate are placed in B, and saturated sodium carbonate solution is placed in D.

Given:
1. Anhydrous calcium chloride can form an insoluble compound with ethanol, $CaCl_2 \cdot 6C_2H_5OH$.
2. Boiling points of related organic compounds:
 - Reagent: Diethyl ether, Ethanol, Acetic acid, Ethyl acetate
 - Boiling point/°C: 34.7, 78.5, 118, 77.1

Please answer the following:
(1) The role of concentrated sulfuric acid is _________ ; if the isotope ^18O tracer method is used to determine the provider of the oxygen atom in the water molecule of the reaction product, write the chemical equation that indicates the position of ^18O: ___________.
(2) The function of the spherical drying tube C is _____________________ . If a few drops of phenolphthalein are added to D before the reaction, the solution turns red. The reason for this phenomenon is (expressed by the ionic equation) _________ ; the phenomenon in D after the reaction is __________________.

**Ground Truth**: (1) Catalyst, dehydrating agent ,$CH_3COOH + C_2H_5{}^{18}OH \rightarrow CH_3CO^{18}OC_2H_5 + H_2O$
(2) To prevent backflow, condensation ,$CO + H_2O \rightleftharpoons HCO + OH^-$,The solution separates into layers, with the upper layer being a colorless oily liquid, and the lower solution becoming lighter in color.

---

**GPT-4o**    **Knowledge Error**

(1) The role of concentrated sulfuric acid is as a catalyst and dehydrating agent. The chemical equation indicating the position of ^18O when using isotope ^18O tracing is:
$CH_3COONa + C_2H_5{}^{18}OH \rightarrow CH_3COOC_2H_5 + Na^{18}OH$

(2) The role of the spherical drying tube C is to remove moisture from the gas. If a few drops of phenolphthalein are added to D before the reaction, the solution appears red because:
$CO_3{}^{2-} + H_2O \rightarrow HCO_3{}^- + OH^-$
The phenomenon in D after the reaction is that the red color dissipates due to the neutralization reaction resulting in the formation of $CO_2$:
$H^+ (from\ H_2SO_4) + CO_3{}^{2-} \rightarrow H_2O + CO_2$

---

**Claude3.5**    **Knowledge Error**

Let's address each part of the question based on the information provided and the image of the apparatus:
(1) The role of concentrated sulfuric acid is a catalyst and dehydrating agent. If isotope 18O tracing method is used, the chemical equation indicating the position of 18O would be:
CH3C18O18ONa + C2H5OH → CH3C18O18OC2H5 + NaOH
The 18O atoms from the acetate group remain in the ethyl acetate product.
(2) The role of spherical drying tube C is to dry the ethyl acetate vapor as it passes through, removing any water vapor.
If a few drops of phenolphthalein are added to D before the reaction, the solution appears red because of the presence of carbonate ions. The ionic equation for this is:
CO32- + H2O ⇌ HCO3- + OH-

The OH- ions cause the phenolphthalein to turn red.
The phenomenon in D after the reaction is that the red color fades or disappears. This is because the ethyl acetate produced reacts with the sodium carbonate solution, neutralizing it:
2CH3COOC2H5 + Na2CO3 + H2O → 2CH3COONa + 2C2H5OH + H2CO3

---

**Qwen-VL-Max**    **Knowledge Error**

(1) The role of concentrated sulfuric acid is to act as a catalyst for the esterification reaction between ethanol and acetic acid. Using isotopic tracing with 18O, the chemical equation showing the position of 18O would be:
CH3CO18OH + C2H5OH → CH3COOC2H5 + H18O
(2) The role of spherical drying tube C is to prevent the entry of moisture into the system, which could interfere with the reaction or cause unwanted side reactions.
When phenolphthalein is added to D before the reaction, it turns red due to the presence of excess sodium carbonate, which is basic. The ionic equation for this reaction is:
CO2(aq) + H2O(l) + CO3^2-(aq) → 2HCO3^-(aq)
After the reaction, the solution in D will turn colorless because the ethyl acetate formed will react with the sodium carbonate, consuming the excess base and neutralizing the solution.

---

**GLM-4V**    **Knowledge Error**

Solution:
(1) Since the boiling point of ethanol is 78.5°C and that of ethyl acetate is 77.1°C, we can use the difference in boiling points to separate them. Adding concentrated sulfuric acid can increase the boiling point of the ethanol-water mixture, which can then be distilled to separate ethanol and water. Therefore, the role of concentrated sulfuric acid is to increase the boiling point of ethanol-water mixtures. The chemical equation for the reaction between ethanol and sulfuric acid is C2H5OH + H2SO4 → C2H5OSO4↓ + H2O. When 18O tracing method is used, the oxygen atom in the water comes from the 18O-labeled ethanol. Therefore, the chemical equation is C2H5OH18 + H2SO4 → C2H5OSO4↓ + H2O18.
(2) The saturated sodium carbonate solution can absorb water, so when the hot ethanol-acetic acid mixture is poured into the flask, it will not overflow. Therefore, the role of spherical drying tube C is to absorb excess water.
The phenolphthalein solution turns pink when there are enough hydrogen ions in solution. In the presence of sodium carbonate, the hydrogen ions from the reaction are consumed, so no color change occurs. After the reaction, the pH of the solution increases, causing the phenolphthalein solution to turn pink.

Figure 64: Cases of errors in the chemistry part of VisScience for several classic close-source models.

**Question**

A certain extracurricular group designed a laboratory apparatus for the preparation of ethyl acetate, as shown in the figure. Concentrated sulfuric acid is placed in A, ethanol and anhydrous sodium acetate are placed in B, and saturated sodium carbonate solution is placed in D.
Given:
1. Anhydrous calcium chloride can form an insoluble compound with ethanol, $CaCl_2 \cdot 6C_2H_5OH$.
2. Boiling points of related organic compounds:
  - Reagent: Diethyl ether, Ethanol, Acetic acid, Ethyl acetate
  - Boiling point/°C: 34.7, 78.5, 118, 77.1
Please answer the following:
(1) The role of concentrated sulfuric acid is ________; if the isotope ^18O tracer method is used to determine the provider of the oxygen atom in the water molecule of the reaction product, write the chemical equation that indicates the position of ^18O: ________.
(2) The function of the spherical drying tube C is ________________. If a few drops of phenolphthalein are added to D before the reaction, the solution turns red. The reason for this phenomenon is (expressed by the ionic equation) ________; the phenomenon in D after the reaction is ________________.

**Ground Truth**: (1) Catalyst, dehydrating agent ,$CH_3COOH + C_2H_5{}^{18}OH \rightarrow CH_3CO^{18}OC_2H_5 + H_2O$
  (2) To prevent backflow, condensation ,$CO + H_2O \rightleftharpoons HCO + OH^-$,The solution separates into layers, with the upper layer being a colorless oily liquid, and the lower solution becoming lighter in color.

**LLaVA-1.5_13B**        **Knowledge Error**

(1) The role of concentrated sulfuric acid is to react with ethanol to produce ethyl acetate and water. The chemical equation for this reaction is:
C2H5OH + H2SO4 → C2H5OAc + H2O
If isotope 18O tracing method is used to determine the provider of oxygen atoms in the water molecules produced, the chemical equation would be:
C2H5OH + H2SO4 → C2H5OAc + 18O-H2O
(2) The role of spherical drying tube C is to dry the ethyl acetate separated from D. If a few drops of phenolphthalein are added to D before the reaction, the solution appears red due to the color change of phenolphthalein in the presence of acidic or basic solutions. The chemical equation for this reaction is:
C2H5OAc + NaOH → C2H5OAc-Na+ + H2O
After the reaction, the solution in D will be clear and colorless, indicating the removal of acidic components.

**GLM-4V-9B**        **Knowledge Error**

(1) The role of concentrated sulfuric acid is to act as a catalyst for the esterification reaction. If isotope 18O tracing method is used to determine the provider of oxygen atoms in the water molecules produced, the chemical equation indicating the position of 18O is:

$H_2O(18O) + CH_3CH_2OH \rightarrow CH_3COO^-(18O) + CH_3CH_2OH(18O)$

(2) The role of spherical drying tube C is to remove any water vapor that might condense and react with the ethanol or ethyl acetate, which could lead to the formation of ethers. If a few drops of phenolphthalein are added to D before the reaction, the solution appears red; the reason for this phenomenon (expressed in ionic equation) is that the saturated sodium carbonate solution is basic and can react with phenolphthalein to form a red complex. The phenomenon in D after the reaction is that the solution turns colorless because the phenolphthalein indicator is neutralized by the produced ethyl acetate, which is a weaker acid than sodium carbonate.

**InternVL 1.5**        **Reasoning Error**

(1) The role of concentrated sulfuric acid is to act as a dehydrating agent, removing water from the reaction mixture. If isotope 18O tracing method is used to determine the provider of oxygen atoms in the water molecules produced, the chemical equation indicating the position of 18O would be:
C2H5OH + CH3COOH → C2H5OCH3 + H2O
In this reaction, the oxygen atoms in the water molecule (H2O) would come from the oxygen atoms in the ethanol (C2H5OH) and the acetic acid (CH3COOH).

Figure 65: Cases of errors in the chemistry part of VisScience for several classic open-source models.

4536
4537
4538
4539
4540
4541
4542
4543
4544
4545
4546
4547
4548
4549
4550
4551
4552
4553
4554
4555

*Question*

A certain experimental group conducted the following analysis on an unknown solution containing $Al^{3+}$:

(1) A small amount of sodium hydroxide was added, with no obvious change observed;
(2) More sodium hydroxide solution was added, resulting in a white precipitate;
(3) An excess amount of sodium hydroxide was added, and the white precipitate significantly decreased.

The experimental group performed a quantitative analysis and obtained the relationship between the amount of precipitate and the volume of sodium hydroxide added, as shown in the figure.

Which of the following statements is incorrect?

A: The unknown solution contains at least three types of cations.
B: The molarity of the added sodium hydroxide solution is 5 mol·L⁻¹.
C: If the other ion is a divalent cation, then a = 10.
D: If the final precipitate is filtered, washed, and calcined, its mass must be 6 g.

*Ground Truth*：*D*

---

**GPT-4o**  **Reasoning Error**

To find the incorrect statements, we need to delve into the chemical principles demonstrated and analyze the data provided in the graph systematically.
Here are the detailed steps and reasoning regarding each statement:

1. Unknown Solution Analysis:
The unknown solution is known to contain $Al^{3+}$ ions which precipitate as $Al(OH)_3$ when NaOH is added.
Initial addition of NaOH: No obvious change, indicating that the OH⁻ ions are not sufficient to exceed the solubility product (Ksp) of $Al(OH)_3$.
Further addition of NaOH: A white precipitate of $Al(OH)_3$ appears, indicating the concentration of OH⁻ has now exceeded the Ksp of $Al(OH)_3$.
Excess NaOH: The white precipitate dissolves due to the formation of soluble complex ion $[Al(OH)_4]^-$.
2. Quantitative Relationship from the Graph:
The graph shows the relationship between the precipitate amount (in moles) and the NaOH added volume (in mL):
- Maximum precipitate mol: 0.20 mol
- Corresponding volume of NaOH: 100 mL
3. Evaluation of Statements:
(A) The unknown solution contains at least 3 cations.
  This is not necessarily true based on the given information. The data provided only indicates the presence of $Al^{3+}$; no information about the presence of other cations can be conclusively determined. Therefore, this statement can be incorrect.
(B) The concentration of the added NaOH solution is 5 mol·L⁻¹.
Given that 0.20 mol of $Al(OH)_3$ precipitate appears when 100 mL of NaOH is added, and knowing the stoichiometry of the reaction: $Al^{3+} + 3OH^- \rightarrow Al(OH)_3$
0.2 mol $Al^{3+}$ corresponds with 3 * 0.2 mol = 0.6 mol OH⁻. For 100 mL NaOH to provide 0.6 mol OH⁻, the concentration of NaOH must indeed be:
Concentration = 0.6 mol / 0.1 L = 6 mol·L⁻¹
This conflicts with 5 mol·L⁻¹. Thus, this statement is indeed incorrect.
(C) If another ion is a divalent cation, then a = 10.
For divalent cation, such as $M^{2+}$, precipitation occurs as $M(OH)_2$. Considering the graph where maximum precipitate occurs around 0.15 mol at some volume 'a', assuming it's from $M(OH)_2$: $M^{2+} + 2OH^- \rightarrow M(OH)_2$
At point 'a', $V_a$ * C_NaOH ≥ (0.15 * 2 = 0.3 mol OH) / 5 mol·L⁻¹ = 0.06 L NaOH = 60 mL
Thus a ≈ 60 mL, not 10 mL, making this statement incorrect.
(D) If the final precipitate is filtered, washed, and ignited, its mass is certainly 6 g.
Upon ignition, $Al(OH)_3$ decomposes into $Al_2O_3$: $2Al(OH)_3 \rightarrow Al_2O_3 + 3H_2O$
The molar mass of $Al_2O_3$ is 102 g/mol. With 0.20 mol of $Al(OH)_3$ yielding 0.10 mol $Al_2O_3$: 0.10 mol * 102 g/mol = 10.2 g

Thus, this treatment yields about 10.2 g of $Al_2O_3$ and not 6 g, making this statement incorrect.
Conclusion: The following statements are all incorrect: A, B, C, and D.
E: None of the statements are correct

---

**Claude3.5**  **Knowledge Error**

Let's analyze each statement based on the information provided and the graph:

A. The unknown solution contains at least 3 cations:
This statement is likely correct. The behavior described (initial precipitation followed by dissolution in excess base) is characteristic of amphoteric hydroxides like Al(OH)3. The graph shows a single precipitate forming and then partially dissolving, which is consistent with the presence of Al3+. There's no clear evidence of other cations.
B. The concentration of the added NaOH solution is 5 mol·L-1:
We can't determine this from the given information. The graph shows volume of NaOH added, not its concentration.
C. If another ion is a divalent cation, then a = 10:
We don't have enough information to confirm or deny this statement. The value of 'a' isn't directly related to the presence of a divalent cation based on the given data.
D. If the final precipitate is filtered, washed, and ignited, its mass is certainly 6 g:
This statement is incorrect. The graph shows the amount of precipitate in moles, not grams. At the end point, there's 0.15 mol of precipitate. Without knowing the exact composition of the precipitate after ignition, we can't determine its mass.

Given the question asks which statements are incorrect, the answer should include B and D. Statement A appears to be correct based on the information given, and C cannot be confirmed or denied with the available data.

Figure 66: Cases of errors in the chemistry part of VisScience for several classic close-source models.

4585
4586
4587
4588
4589

**Question**

A certain experimental group conducted the following analysis on an unknown solution containing $Al^{3+}$:

(1) A small amount of sodium hydroxide was added, with no obvious change observed;
(2) More sodium hydroxide solution was added, resulting in a white precipitate;
(3) An excess amount of sodium hydroxide was added, and the white precipitate significantly decreased.

The experimental group performed a quantitative analysis and obtained the relationship between the amount of precipitate and the volume of sodium hydroxide added, as shown in the figure.

Which of the following statements is incorrect?

A: The unknown solution contains at least three types of cations.
B: The molarity of the added sodium hydroxide solution is 5 mol·L$^{-1}$.
C: If the other ion is a divalent cation, then a = 10.
D: If the final precipitate is filtered, washed, and calcined, its mass must be 6 g.

**Ground Truth：D**

**Qwen-VL-Max     Knowledge Error**

The image shows a graph that plots the amount of precipitation against the volume of sodium hydroxide (NaOH) added to an unknown solution. The x-axis represents the volume of NaOH added in mL, while the y-axis indicates the amount of precipitation in moles.

From the graph, we can see that when a small amount of NaOH is added, there is no visible change in the solution. As more NaOH is added, a white precipitate appears. When an excess of NaOH is added, the white precipitate decreases significantly.

Based on this information, let's analyze each statement:

A: The unknown solution contains at least 3 cations.
This statement is correct because the addition of a small amount of NaOH does not cause any reaction, indicating that some ions do not react with NaOH. Then, a white precipitate forms upon further addition of NaOH, suggesting the presence of at least one cation that reacts with NaOH. Finally, the decrease in the precipitate after adding an excess of NaOH implies the existence of another cation that also reacts with NaOH but has a different behavior than the first one. Therefore, it is reasonable to conclude that the solution contains at least three cations.
B: The concentration of the added NaOH solution is 5 mol·L^-1.
This statement is incorrect. Although the graph provides data points for the amount of precipitation and the volume of NaOH added, it does not provide enough information to determine the concentration of the NaOH solution. To calculate the concentration, we would need additional details such as the initial volume or mass of the NaOH solution used.
C: If another ion is a divalent cation, then a = 10.
This statement is correct. According to the graph, the volume of NaOH required to form the maximum amount of precipitation is around 100 mL. Since the graph shows that the amount of precipitation increases linearly with the volume of NaOH added until reaching a maximum value, we can assume that the reaction follows a simple stoichiometry. If the other ion is a divalent cation, it will require twice as much NaOH to form the same amount of precipitation compared to a monovalent cation. Thus, if the volume of NaOH needed to reach the maximum precipitation was 100 mL for a monovalent cation, it would be 200 mL for a divalent cation. However, since the graph shows that the maximum precipitation occurs at around 100 mL, we can infer that the other ion must be a monovalent cation. In this case, a = 10, which corresponds to the point where the slope changes from positive to negative.
D: If the final precipitate is filtered, washed, and ignited, its mass is certainly 6 g.
This statement is incorrect. While the graph shows that the maximum amount of precipitation formed is approximately 0.15 mol, we cannot directly convert this into grams without knowing the molar mass of the precipitated compound. Furthermore, filtering, washing, and igniting the precipitate may result in losses due to incomplete recovery or decomposition during heating. Therefore, we cannot confidently predict the exact mass of the final product based solely on the given information.

In conclusion, the correct answer is B: The concentration of the added NaOH solution is 5 mol·L^-1. This statement is incorrect because there is insufficient information provided to determine the concentration of the NaOH solution.

**GLM-4V     Reasoning Error**

Solution: According to the graph, when 10 mL of NaOH is added, the precipitation does not dissolve, indicating that the concentration of aluminum ions is less than 0.20 mol/L. When 11 mL of NaOH is added, the precipitation begins to dissolve, which means that the concentration of aluminum ions is greater than 0.20 mol/L but less than 0.25 mol/L. Since the initial volume of the solution is 100 mL and the final volume after adding NaOH is 110 mL, the total volume of the solution is 210 mL. Therefore, the concentration of sodium hydroxide is 0.20 mol/L. Thus, option B is incorrect.

When 11 mL of NaOH is added, the precipitation begins to dissolve, which means that the concentration of aluminum ions is greater than 0.20 mol/L but less than 0.25 mol/L. Therefore, the concentration of sodium hydroxide is 0.20 mol/L. Thus, option B is incorrect.

The mass of the precipitate before heating is 8 g, so the mass of the precipitate after heating is 6 g. Therefore, option D is incorrect.

Therefore, the answer is C.
Answer: C

Figure 67: Cases of errors in the chemistry part of VisScience for several classic close-source models.

**Question**

A certain experimental group conducted the following analysis on an unknown solution containing $Al^{3+}$:

(1) A small amount of sodium hydroxide was added, with no obvious change observed;
(2) More sodium hydroxide solution was added, resulting in a white precipitate;
(3) An excess amount of sodium hydroxide was added, and the white precipitate significantly decreased.

The experimental group performed a quantitative analysis and obtained the relationship between the amount of precipitate and the volume of sodium hydroxide added, as shown in the figure.

Which of the following statements is incorrect?

A: The unknown solution contains at least three types of cations.
B: The molarity of the added sodium hydroxide solution is 5 mol·L⁻¹.
C: If the other ion is a divalent cation, then a = 10.
D: If the final precipitate is filtered, washed, and calcined, its mass must be 6 g.

**Ground Truth：D**

---

**LLaVA-1.5_13B      Knowledge Error**

The incorrect statement is B: The concentration of the added NaOH solution is 5 mol·L⁻¹. The actual concentration of the NaOH solution is 10 mol·L⁻¹.

---

**GLM-4V      Knowledge Error**

The graph shows that the precipitate decreases after the addition of an excess of sodium hydroxide, which indicates that the precipitate is not due to the presence of a single cation. Therefore, option A is incorrect as it states that the unknown solution contains at least 3 cations.

The graph does not provide any information about the concentration of the NaOH solution. It only shows the volume of NaOH added and the corresponding change in the precipitate. Therefore, option B is incorrect as it states the concentration of the NaOH solution is 5 mol·L^-1.

Option C is incorrect because the graph does not provide enough information to determine the value of a if another ion is a divalent cation.

Option D is correct because if the precipitate is filtered, washed, and ignited, its mass will be the mass of the hydrated oxide, which is the product of the original cation and the hydroxide ion. Since the graph shows that the precipitate decreases after the addition of an excess of sodium hydroxide, it is reasonable to assume that the precipitate is hydrated oxide. Therefore, the mass of the precipitate will be the product of the concentration of the original cation and the hydroxide ion, which is 6 g.

Therefore, the answer is D

---

**InternVL 1.5      Knowledge Error**

First, we need to understand the information represented in the chart. The chart shows the relationship between the amount of precipitate in an unknown solution and the volume of NaOH added. According to the chart, the amount of precipitate reaches its maximum when the volume of NaOH added is between 0.15 mL and 0.20 mL..

Now, let's analyze each option:

A: The unknown solution contains at least three types of cations. This statement is correct because the increase in the amount of precipitate with the addition of NaOH indicates the presence of at least three cations, as $Al^{3+}$ requires three NaOH molecules to completely precipitate.

B: The concentration of the added NaOH solution is 5 mol·L⁻¹. This statement is incorrect because the chart does not provide any information about the concentration of the NaOH solution.

C: If the other ion is a divalent cation, then a = 10. This statement is incorrect because the chart shows that the amount of precipitate reaches its maximum when the volume of NaOH added is 0.15 mL, indicating the presence of a divalent cation, but the value of a is not 10.

D: If the final precipitate is filtered, washed, and calcined, its mass must be 6 g. This statement is incorrect because the chart does not provide any information about the mass of the precipitate.

Therefore, option B is incorrect.

Figure 68: Cases of errors in the chemistry part of VisScience for several classic open-source models.

