# OpenReview forum: "VisScience: An Extensive Benchmark for Evaluating K12 Educational Multi-modal Scientific Reasoning"
_ICLR.cc/2025/Conference — ICLR 2025 Conference Withdrawn Submission_

### Official Review · Reviewer_8GXX · 2024-11-01

**Soundness:** 3
**Presentation:** 3
**Contribution:** 4
**Rating:** 8
**Confidence:** 5

**Summary:**

The paper introduces a comprehensive benchmark designed to assess the capabilities of multi-modal large language models (MLLMs) across various scientific disciplines, including mathematics, physics, and chemistry. The benchmark, named VisScience, comprises 3,000 questions drawn from K12 education, evenly distributed across the three disciplines, and covers 21 distinct subjects categorized into five difficulty levels. The authors conducted extensive experiments with 25 representative MLLMs, revealing that closed-source MLLMs generally outperform open-source models in scientific reasoning tasks. Notable performances include Claude3.5-Sonnet with 53.4% accuracy in mathematics, GPT-4o with 38.2% in physics, and Gemini-1.5-Pro with 47.0% in chemistry. The paper highlights the strengths and limitations of MLLMs and underscores the importance of developing models that can effectively handle multi-modal scientific reasoning.

**Strengths:**

The paper presents the VisScience benchmark, which is an original contribution to the field of multi-modal large language models (MLLMs). It addresses a significant gap in the evaluation of MLLMs by extending the scope beyond mathematics to include physics and chemistry, covering a comprehensive range of scientific disciplines. The benchmark's design, integrating questions from K12 education and classifying them into detailed subjects and difficulty levels, is a creative approach that provides a more nuanced assessment of MLLM capabilities.

The quality of the VisScience benchmark is evident in its thorough construction and rigorous evaluation process. The dataset is meticulously curated, with 3,000 high-quality questions selected to represent a breadth of knowledge points and difficulty levels. The authors' commitment to ensuring the benchmark's reliability through multiple checks and annotations reflects a high standard of quality in dataset curation.

**Weaknesses:**

While the paper provides a general categorization of errors, a more in-depth analysis of the types of errors made by MLLMs across different subjects could offer more targeted insights for model improvement. Suggestions for specific model enhancements or training techniques based on error analysis would be valuable additions.

**Questions:**

Will the authors be making the VisScience benchmark and the associated evaluation tools publicly available? If so, what are the plans for supporting the community in using these resources?

The paper categorizes errors into several types, but does not provide specific recommendations for model improvement. Could the authors provide more detailed insights or suggestions on how to address the most common error types observed?

---

### Official Review · Reviewer_PLH9 · 2024-11-01

**Soundness:** 2
**Presentation:** 2
**Contribution:** 2
**Rating:** 3
**Confidence:** 4

**Summary:**

This paper proposes VisScience, a benchmark for evaluating MLLMs, covering multiple disciplines such as mathematics, physics, and chemistry, with a total of 3000 questions. The authors aim to distinguish VisScience from existing benchmarks through "difficulty level", "bilingual" and ultimately present a more comprehensive set of experimental results.

**Strengths:**

1. The paper presents the VisScience benchmark, encompassing multiple disciplines to provide a more comprehensive evaluation guide for evaluating MLLMs.
2. The paper conducts extensive experiments and performs error analysis across different disciplines.

**Weaknesses:**

1. **Bilingual Dimension:** As a benchmark for evaluating MLLMs, the language of the text within images is highly significant. The authors present two languages in VisScience. However, through the Appendix, it can be observed that in English questions, most images still contain Chinese text.
2. **Difficulty Level Classification:** The authors describe the use of LLM for classifying the difficulty levels, while VisScience is a multimodal benchamrk. I have some concerns about using text alone for classification without considering the images. For instance, in a math problem asking about an area, the complexity presented by the image can vary greatly, which cannot be detected by simply using LLM for classification.

**Questions:**

Please refer to the weaknesses section. The two points mentioned will impact the practical contributions of this paper, and I hope the authors can provide a detailed explanation. Moreover, VisScience includes the disciplines of physics, chemistry, and mathematics. Mathematics is already widely recognized as an essential foundational capability. However, is the significance of physics and chemistry limited to specific application scenarios for the model? I also hope the authors can provide a more detailed interpretation of the motivation and significance of these two disciplines from the perspective of model evaluation.

---

### Official Review · Reviewer_NKyR · 2024-11-02

**Soundness:** 2
**Presentation:** 3
**Contribution:** 3
**Rating:** 5
**Confidence:** 4

**Summary:**

This submission introduces VisScience, an extensive benchmark for evaluating K12 educational multi-modal scientific reasoning across mathematics, physics, and chemistry using large language models.​ The authors highlight the strengths and limitations of current MLLMs and suggest areas for future improvement.

**Strengths:**

There are several noticeable strengths in this paper:
* This paper presents a well-justified motivation for creating a benchmark focusing on K12 math, physics, and chemistry knowledge.
* This submission conducts extensive experiments ranging from open-source and closed-source LLM and MLLMs.
* The manuscript is commendable for its clarity and structured writing style, which greatly facilitates reader comprehension.
* Additionally, the inclusion of clear and illustrative figures and tables is a notable strength, as it significantly aids in conveying the main claims of the paper to the audience.

**Weaknesses:**

This submission creates a benchmark, which is a notable contribution without a doubt. However, there are some unignorable weaknesses that, if addressed, could greatly enhance the clarity and usability of the benchmark.

---

**Weakness #1: Ambiguity in Dataset Annotation Process**

The dataset annotation process described in Section 2.2 is ambiguous, with multiple steps left unclear. Specific questions regarding this process include:

(1-1) Source of Translations: How were the English and Chinese versions of each question obtained? Which version served as the source, and how were questions and answers translated—via large language models (LLMs) or human annotators?

(1-2) Quality Control and Verification: What steps were taken to ensure quality and correctness in the annotations? The authors mention “we check xxx” and “we screen out xxx” (Lines 202-211), but it is unclear who "we" refers to. Was this verification conducted solely by the authors or by additional human annotators? Additionally, was the check single-pass, or did multiple judges perform repeated rounds of verification?

(1-3) Presence of Bilingual Images: The K12 questions listed in the Appendix indicate that many images contain Chinese characters. Are English versions of these images also available?

(1-4) Question-Image Pairing: In Table 1, the authors report 3k questions and 3k images. Does each question have an associated image, or are there cases where a single question is linked to multiple images?

---

**Weakness #2: Lack of Clarity in Experimental Setup and Analysis**

Certain aspects of the experimental setup are unclear, and additional analysis is needed for some experimental results. Specific questions include:

(2-1) Input for English-specialized MLLMs: In Lines 1052-1053, the authors state that “The English version of VisScience is designed to facilitate the evaluation of MLLMs that specialize in English, assessing their capabilities in scientific reasoning.” Were English versions of the images provided as inputs to these MLLMs during evaluation?

(2-2) Unexpected Performance of LLMs vs. MLLMs: In Table 2 and Table 3, text-only LLMs that accept text questions but no image inputs sometimes outperform MLLMs with access to both text questions and images. Why might this discrepancy occur?

(2-3) Questionable Necessity of Image Inputs: A followup question of (2-2) -- the accuracies on some text-only LLMs are higher than random guess (25% for multiple choices with 4-choice), is it because the image inputs are not necessary, or is it because the LLMs have already been posted to these questions during training that they don't even need to look at the paired image input to answer?


---


**Weakness #3: Benchmark Score Evaluation Methodology**

The evaluation method for benchmark scores is unclear, particularly in Section 3.1, Lines 316-317, where the authors mention GPT-4o as the judge. Questions related to score evaluation include:

(3-1) Handling Multiple Correct Answers: For example, in the first example in Figure 11 on page 30, a multiple-choice question has multiple correct answers (ground truth C & D). If a model selects only C, is this considered correct, partially correct, or incorrect? How is accuracy calculated in such cases?

(3-2) Accuracy for Multi-part Free-form Questions: In Figure 19 on page 38, a free-form question contains multiple sub-questions. How is accuracy determined here? Do all sub-questions carry equal weight, and how many must a model answer correctly for GPT-4o to deem the entire question correct?

(3-3) Weighting Free-form vs. Multiple-choice Questions: Free-form questions often contain multiple sub-questions. When reporting overall accuracy, do these free-form questions carry the same weight as single-answer multiple-choice questions? For instance, if a free-form question has five sub-questions, how is the overall score calculated?

**Questions:**

Please refer to the Weaknesses Section above.

---

### Official Review · Reviewer_cxoP · 2024-11-04

**Soundness:** 2
**Presentation:** 3
**Contribution:** 2
**Rating:** 3
**Confidence:** 4

**Summary:**

The paper introduces the VisScience benchmark, which is designed to evaluate the performance of multimodal large language models (MLLMs) in scientific reasoning tasks. This benchmark covers three major subjects—mathematics, physics, and chemistry，consists of 3,000 questions, spanning both English and Chinese language tasks. It is further divided into 21 sub-disciplines and five difficulty levels. Through experiments, the authors tested 25 MLLMs, and the results showed that closed-source models performed better across all subjects.

**Strengths:**

- The paper introduces VisScience, a multimodal scientific reasoning benchmark covering mathematics, physics, and chemistry, with bilingual (Chinese and English) support. This benchmark addresses the shortcomings of existing benchmarks in multidisciplinary and multilingual evaluations. The dataset has undergone rigorous screening and verification by both LLMs and human experts to ensure its quality. Comprehensive evaluations were conducted on 25 closed-source and open-source MLLMs, revealing the strengths and limitations of different models in scientific reasoning.

**Weaknesses:**

- VisScience includes bilingual tasks in Chinese and English; however, in the case studies presented in the appendix, it is observed that the text within images corresponding to English questions might be in Chinese. This bilingual mixing situation could lead to different reasoning capabilities of the models when processing information in different languages, especially during the integration of visual and linguistic information. Therefore, it is suggested that the authors further investigate and address this issue in future research to ensure that the models can perform stably in true multilingual environments.
- The paper mentions that the data underwent multiple checks using manual review and LLM assessment, along with some screening principles. However, the details of this process are not elaborated upon in the paper. This process is crucial for ensuring the quality of the dataset and the accuracy of model evaluation. It is recommended that the authors supplement the paper with a detailed description of this process.

**Questions:**

- In the experimental section of this paper, a text-only setting was adopted, but it is not explicitly stated whether the problems used are the same as those in the multimodal setting. If the two are indeed consistent, for questions that rely on graphical information, posing them through text alone may not constitute a complete question. If this design is intended to validate the effectiveness of graphical information, it is recommended that the authors provide clearer explanations in the text to elucidate the purpose of the experimental design and the interpretation of the results.
- The paper categorizes the inference results of the model but does not provide detailed definitions and classification criteria for each type of error. To enhance the transparency and reproducibility of the paper, it is recommended that the authors clearly define each type of error in the text and describe the specific methods used for error classification.
- The source of the data is not clearly specified in the paper. It is suggested that the authors provide additional explanations regarding the data sources in the paper.
- In Fig. 26, Fig. 32, and Fig. 33, the titles indicate that they display cases of standard answers and correct GPT-4o answers. However, it can be observed that in some cases, the answers provided by the model are actually incorrect. It is recommended that the authors carefully review all figures and cases in the paper to ensure that the displayed results match the annotations and to avoid such errors, thereby improving the accuracy and reliability of the paper.

---

### Note · Authors · 2024-12-13

I have read and agree with the venue's withdrawal policy on behalf of myself and my co-authors.